# Enhance the Safety in Reinforcement Learning by ADRC Lagrangian Methods

## Abstract

Safe reinforcement learning (Safe RL) seeks to maximize rewards while satisfying safety constraints, typically addressed through Lagrangian-based methods. However, existing approaches, including PID and classical Lagrangian methods, suffer from oscillations and frequent safety violations due to parameter sensitivity and inherent phase lag. To address these limitations, we propose ADRC-Lagrangian methods that leverage Active Disturbance Rejection Control (ADRC) for enhanced robustness and reduced oscillations. Our unified framework subsumes a broad class of PID Lagrangian updates as frozen-parameter special cases, with the classical update arising as the degenerate integral-only member of the PID family, while significantly improving safety performance. Extensive experiments demonstrate that our approach reduces violation rates by 42–50%, constraint violation magnitudes by 83–86%, and average costs by 47–52% relative to Lagrangian baselines, establishing superior effectiveness for Safe RL in complex environments.

## 1 Introduction

Reinforcement Learning (RL) aims to maximize rewards as agents interact with environments, finding applications in fields such as robotics (Sun et al., 2023; Luo et al., 2024; Li et al., 2024) and the post-training of Large Language Models (LLMs) (Bai et al., 2022; Lee et al., 2023; Rafailov et al., 2023). However, in real-world scenarios like autonomous driving (Muhammad et al., 2020), safety requirements are often of paramount importance. It is essential not only to maximize rewards but also to ensure compliance with safety constraints. To address this challenge, Safe RL (Garcıa & Fernández, 2015) has emerged as a paradigm dedicated to reliable and robust policy learning in complex and dynamic environments. Safe RL is typically formulated as a Constrained Markov Decision Process (CMDP). Among the various approaches for solving CMDPs, Lagrangian methods play a pivotal role by transforming constrained optimization problems into unconstrained ones through the introduction of a dual variable, the Lagrange multiplier. This transformation enables the adaptation of any RL algorithm into a Safe RL framework (Achiam et al., 2017; Chow et al., 2018b; 2019), leading to the development of numerous novel Safe RL algorithms (Liu et al., 2024; Chen et al., 2024).

Classical Lagrangian updates can be interpreted as pure integral controllers on the constraint violation signal. While simple, this mechanism reacts slowly to the rapid distributional shifts caused by policy updates and the stochasticity of cost estimates, leading to lag, overshoot, and persistent oscillations in safety performance. Attempts to mitigate these issues with PID-based extensions (Stooke et al., 2020) reduce oscillations by adding proportional and derivative terms, but their behavior remains fragile: performance is highly sensitive to the chosen gains and rarely transfers robustly across tasks (Panda, 2012; Åström & Hägglund, 1995). These limitations point to a deeper challenge: existing methods lack a way to explicitly counteract the drifting disturbances that underlie oscillatory training.

To overcome this challenge, we turn to the broader toolbox of *adaptive control*, whose central goal is to maintain stability and performance under unknown and time-varying dynamics. Among the many adaptive strategies, Active Disturbance Rejection Control (ADRC) (Han, 1998; Zhong et al., 2020a) is particularly well suited to the Safe RL setting. Unlike classical adaptive methods that often require a parametric model

or extensive gain tuning, ADRC treats all uncertainty, including model error, noise, and nonstationarity, as a lumped disturbance, and employs a lightweight observer to estimate and cancel it online. This observer-based design makes ADRC both model-free and robust to changing dynamics, while its reliance only on observable signals (such as cost returns) makes it a natural match for reinforcement learning. In contrast, alternatives such as MRAC (Nguyen & Nguyen, 2018; Singh & Kumar, 2015) or Lyapunov-based adaptive schemes (Parks, 1966) typically assume access to richer state information or stronger structural knowledge, which is impractical in high-dimensional RL environments.

Building on this idea, we introduce ADRC Lagrangian methods, which augment the classical dual update with an observer that estimates and cancels the disturbance acting on the constraint return. By combining this observer with a smooth reference trajectory for the cost threshold, our update suppresses transient overshoot while directly compensating for nonstationarity and noise. The resulting method is simple to implement, model-free, and optimizer-agnostic, yet it fundamentally changes the dynamics of constraint regulation: we derive a theoretical lower bound on the observer gain that provides safe defaults across diverse environments, we show that freezing the adaptive components of our update yields exactly a PID Lagrangian rule and characterize precisely which PID gains arise this way, with the classical integral update as the degenerate member of the PID family, and we establish through frequency-domain analysis that our approach achieves smaller disturbance-estimation error and reduced phase lag. Empirically, these properties translate into substantial improvements in safety throughout training: on OmniSafe benchmarks our method reduces violation rates by 42–50% relative to classical and PID Lagrangian baselines (70.47%/80.94% → 40.62%), lowers violation magnitudes by 83–86% (17.00/20.68 → 2.85), and decreases average costs by 47–52% (38.90/43.43 → 20.65), all demonstrated by TD3-ADRC on CarGoal in Table 1, while maintaining competitive reward performance. These results demonstrate that bringing the ADRC perspective into Safe RL yields both principled theoretical guarantees and significant empirical gains.

In conclusion, our main contributions are:

- We are the first to introduce ADRC into Safe RL, dynamically adjusting the Lagrange multiplier to improve constraint satisfaction and training stability.

- We theoretically establish that, with frozen observer gain and no transient shaping, our ADRC update reduces exactly to a PID Lagrangian rule, and we characterize exactly which PID gains arise in this way; the classical integral update is the degenerate member of the PID family. Moreover, through frequency-domain analysis, we demonstrate that our method significantly reduces phase lag compared to traditional approaches, leading to faster and more stable constraint satisfaction.

- Comprehensive experiments validate the effectiveness of our method, showing significant improvements in reducing oscillations during training across diverse benchmarks.

## 2  Related Work

**Safe RL**  Safe reinforcement learning (RL) aims to find optimal policies that maximize rewards while satisfying safety constraints (Garcıa & Fernández, 2015; Achiam et al., 2017; Wachi & Sui, 2020; Yang et al., 2020). Common approaches include safe exploration techniques to ensure safety during training (Sui et al., 2015; Wang et al., 2023) and the primal-dual framework, which employs Lagrangian multipliers to address constrained optimization (Ray et al., 2019; Ding et al., 2020; Chow et al., 2018b; 2019). Recent advances have improved the tuning of these multipliers through gradient-based methods (Lillicrap et al., 2019; Tessler et al., 2018; Zhang et al., 2020), PID-based updates (Stooke et al., 2020), adaptive primal-dual methods (Chen et al., 2024), and variational inference approaches (Liu et al., 2022; Huang et al., 2022), enhancing algorithm stability and performance (Yao et al., 2024). On-policy algorithms can be broadly categorized into Lagrangian methods, such as PDO (Chow et al., 2018b), and convex optimization methods like CPO (Achiam et al., 2017) and CVPO (Liu et al., 2022). Recent developments, including APPO (Dai et al., 2023) and CUP (Yang et al., 2022), specifically address oscillatory behaviors and improve constraint feasibility. Furthermore, accurate estimation of objective and constraint functions is crucial, as it significantly influences the efficiency and reliability of policy updates (Altman, 2021). Additionally, methods targeting training stability, such

as policy inertia learning (Chen et al., 2021) and soft-switching gradient manipulation (Gu et al., 2024), have effectively reduced oscillations, highlighting their importance for Safe RL. Complementary to the online CMDP setting considered here, Safe Policy Improvement approaches (Laroche et al., 2019; Scholl et al., 2022) learn from offline data with guarantees of not underperforming a behavior baseline, without requiring a CMDP formulation.

**Control Theory**  Control theory includes traditional controllers such as Proportional-Integral-Derivative (PID) controllers (Åström & Hägglund, 2006; Ang et al., 2005), which, despite their widespread use, are sensitive to parameter variations and external disturbances (Panda, 2012; Åström & Hägglund, 1995). To overcome these limitations, Model Reference Adaptive Control (MRAC), which is a milestone of adaptive control, has been developed, enabling dynamic parameter adjustment to maintain performance under system uncertainties (Nguyen & Nguyen, 2018; Parks, 1966). By incorporating the MIT rule (Mareels et al., 1987), MRAC-PID controllers leverage backpropagation-inspired methods (Rumelhart et al., 1986) to adapt PID parameters in real time, addressing control gain sensitivity (Singh & Kumar, 2015; Kungwalrut et al., 2011). Furthermore, Zhang & Guo (2019) demonstrated that PID parameters could be selected within a specific manifold to ensure global system stabilization with exponential error convergence. In parallel, Active Disturbance Rejection Control (ADRC) has emerged as a robust alternative for managing uncertainties and disturbances. First introduced by Han (1998), ADRC has been further developed to enhance its applicability and theoretical underpinnings (Han, 2009). Recent advancements include applications in nonlinear systems (Guo & Zhao, 2017) and rigorous stability analysis using Lyapunov functions (Zhong et al., 2020b), solidifying ADRC as an effective and versatile control strategy.

## 3  Preliminaries

**Safe Reinforcement Learning**  A Markov Decision Process (MDP) $\mathcal{M}$ (Puterman, 2014) is defined by the tuple $(\mathcal{S}, \mathcal{A}, R, \mathbb{P}, \mu, \gamma)$, where $\mathcal{S}$ and $\mathcal{A}$ denote state and action spaces, $R$ is the reward function, $\mathbb{P}(s' \mid s, a)$ is the state transition probability, $\mu$ is the initial state distribution, and $\gamma \in (0, 1)$ is the discount factor. A parameterized stationary policy $\pi_\theta(a \mid s)$ specifies action probabilities given state $s$. The goal of reinforcement learning (RL) is to maximize the expected return:

$$J(\pi_\theta) = \mathbb{E}_{s \sim \mu} \left[ \sum_{t=0}^{\infty} \gamma^t r_{t+1} \right]. \tag{1}$$

Safe RL is typically formulated as a constrained MDP (CMDP) (Altman, 1999), which extends MDPs with constraints defined by cost functions $c_i : \mathcal{S} \times \mathcal{A} \to \mathbb{R}$ and thresholds $d_i$. The cost return under policy $\pi_\theta$ is:

$$J_{c_i}(\pi_\theta) = \mathbb{E}_{\pi_\theta} \left[ \sum_{t=0}^{\infty} \gamma^t c_i(s_t, a_t) \right]. \tag{2}$$

Safe RL aims to find an optimal policy:

$$\pi^* = \arg\max_{\pi_\theta} J(\pi_\theta) \quad \text{s.t.} \quad J_{c_i}(\pi_\theta) \leq d_i, \forall i. \tag{3}$$

**Lagrangian Methods**  In constrained optimization problems such as those in Safe RL, the goal is to maximize the objective function while satisfying constraints. A common approach is to apply the Lagrangian method. Specifically, for a CMDP with a single cost constraint, denote $d$ as the cost threshold, and $\lambda \geq 0$ as the Lagrangian multiplier, we define the Lagrangian function as:

$$\mathcal{L}(\theta, \lambda) = J(\pi_\theta) - \lambda(J_c(\pi_\theta) - d), \tag{4}$$

The optimal solution aims to maximize the Lagrangian with respect to $\theta$ while minimizing it with respect to the multiplier $\lambda$. To achieve this, we apply a gradient-based approach to update $\lambda$ iteratively. Specifically, we define the constraint violation as $g(\pi_\theta) := J_c(\pi_\theta) - d$. Denote $\alpha > 0$ as the learning rate controlling the update step size; we perform gradient descent on $\lambda$ to minimize $\mathcal{L}$:

$$\dot{\lambda} = \alpha g(\pi_\theta), \tag{5}$$

since $\frac{\partial \mathcal{L}}{\partial \lambda} = -g(\pi_\theta)$, this is equivalently:

$$\dot{\lambda} = -\alpha \frac{\partial \mathcal{L}(\theta, \lambda)}{\partial \lambda} = \alpha g(\pi_\theta), \tag{6}$$

Discretizing over time, the Lagrangian multiplier is updated iteratively by:

$$\lambda_t = \lambda_{t-1} + \alpha g(\pi_{\theta_{t-1}}), \tag{7}$$

or equivalently, by summing constraint violations over time:

$$\lambda_t = \lambda_0 + \alpha \sum_{\tau=0}^{t-1} g(\pi_{\theta_\tau}) \approx \alpha \int_0^t g(\pi_{\theta_\tau}) d\tau. \tag{8}$$

This shows that classical Lagrangian methods implement a pure Integral (I) controller on the constraint violation signal $g(\pi_\theta)$. With this view, to reduce oscillations during training, PID Lagrangian methods (Stooke et al., 2020) generalize the integral control by adding proportional (P) and derivative (D) terms into the dynamics of $\lambda$:

$$\dot{\lambda} = \alpha g(\pi_\theta) + \beta \dot{g}(\pi_\theta) + \gamma \ddot{g}(\pi_\theta), \tag{9}$$

where $\alpha$, $\beta$, and $\gamma$ are positive coefficients controlling the strength of the I, P, and D terms respectively.

Similarly, integrating this equation over time, the resulting PID update law for the multiplier is:

$$\lambda_t = \left( K_p g(\pi_{\theta_t}) + K_i \int_0^t g(\pi_{\theta_\tau}) d\tau + K_d \dot{g}(\pi_{\theta_t}) \right)_+, \tag{10}$$

where $K_p$, $K_i$, $K_d$ are proportional, integral, and derivative gains that need to be tuned carefully, with $(K_i, K_p, K_d) = (\alpha, \beta, \gamma)$ under this integration, and $(\cdot)_+ = \max(0, \cdot)$ keeps the multiplier nonnegative.

**Active Disturbance Rejection Control** Compared with PID controller, Active Disturbance Rejection Control (ADRC) (Han, 1998; Zhong et al., 2020a) provides a more adaptive and resilient alternative. Unlike traditional PID control, ADRC explicitly estimates and compensates for unknown disturbances through an observer-based framework, reducing reliance on precise model knowledge and hyperparameter sensitivity. The core component of ADRC is the Extended State Observer (ESO), which is designed to simultaneously estimate both the internal system states and the total disturbance affecting the system dynamics. By accurately reconstructing the disturbance in real time, the control input can proactively reject its influence, significantly enhancing system stability and performance. In practical Safe RL scenarios, where exact system dynamics are unknown and only observable quantities like costs are available, a reduced-order ESO design is commonly employed. In addition to disturbance estimation, to achieve smoother system behavior and better transient performance, the control strategy can also incorporate a designed reference trajectory that guides the evolution of the system states toward the desired setpoints.

## 4 Method

### 4.1 Closed-loop System Representation of Safe RL

Lagrangian-based Safe RL can be viewed as a feedback system: the policy affects the cumulative cost, which drives the Lagrange multiplier that in turn influences the policy update. We capture this interaction with a simple continuous-time closed-loop abstraction,

$$\begin{cases} x_1 = J_c, \\ \dot{x}_1 = x_2, \\ \dot{x}_2 = f(x_1, x_2, t) - \lambda_t, \end{cases} \tag{11}$$

where $x_1$ is the cumulative cost, $x_2$ its derivative, and $\lambda_t$ is the Lagrange multiplier. The total disturbance $f(x_1, x_2, t)$ aggregates all indirect effects of the multiplier on the cost, including optimizer dynamics, sampling

noise, and policy-induced nonstationarity; it is the quantity that existing dual updates fail to account for. In the discrete training loop, $x_1$ corresponds to the episodic cost estimate at each policy iteration and $x_2$ to its finite difference across iterations (cf. Algorithm 1). The negative sign reflects the fact that a larger multiplier penalizes cost more heavily in the primal objective, acting as negative feedback on the cost dynamics. This closed-loop view is consistent with the feedback perspective already implicit in prior work: the classical Lagrangian update (Sec. 3) is integral feedback on the violation signal, and PID Lagrangian methods (Stooke et al., 2020) augment it with proportional and derivative terms. Eqn. 11 makes this perspective explicit and, importantly, reveals the root cause of oscillations: the dynamics drift as the policy changes, while the multiplier behaves like an integral controller that lags behind disturbances. The formal regularity condition on $f$ is given by the disturbance class of Eqn. 19 in Sec. 4.4, which bounds how fast $f$ may vary. This is exactly the setting ADRC is designed for: any unmodeled effect on the cost dynamics is estimated online by the observer (Sec. 4.3) and compensated in real time.

Our goal is to replace the lagging integral mechanism with an observer-augmented update inspired by ADRC. By explicitly estimating the total disturbance from cost signals and compensating it in real time, while guiding constraint satisfaction through a smooth reference trajectory, our method achieves faster and more stable regulation than classical or PID-based approaches. The designs of the observer and reference signal are presented next.

### 4.2 Arranging a Transient Process

In Safe RL, the dual update implicitly drives the cumulative cost $x_1$ toward the safety threshold $d$. To formalize this objective, we introduce a reference signal $y^*(t)$, which represents the target trajectory that $x_1$ should ideally follow. Since the ultimate goal is constraint satisfaction, the natural choice of reference is a constant at the threshold:

$$y^*(t) = d. \tag{12}$$

However, tracking this signal directly can be problematic in practice. At the beginning of training, policies are usually far from safe, so the gap $x_1(0) - d$ is large. Forcing the multiplier to eliminate this gap immediately leads to abrupt updates, which amplify estimator noise and policy nonstationarity into overshoot and repeated violations. Empirically, this appears as sharp cost spikes in early training and oscillatory swings of $\lambda_t$, even when the constraint is ultimately feasible.

To prevent these instabilities, we need to arrange a transient process that gradually shrinks the effective budget from the current cost level toward $d$ with critically damped dynamics. In the Safe RL context, this corresponds to a smooth budget schedule: early training permits a controlled violation margin that decays over time, enabling exploration while guiding the system toward feasibility.

Concretely, we filter $y^*(t)$ through a second-order system:

$$\ddot{r} = -2c_r\dot{r} - c_r^2(r - d), \qquad r(0) = x_1(0), \ \dot{r}(0) = x_2(0), \tag{13}$$

where $c_r > 0$ controls the tightening speed. The resulting reference trajectory is

$$r(t) = d + \big(x_1(0) - d\big)e^{-c_r t} + \big(x_2(0) + c_r(x_1(0) - d)\big)te^{-c_r t}, \tag{14}$$

which starts from the current cost level and slope, then converges smoothly and non-oscillatorily to $d$. This shaped reference avoids abrupt enforcement in early training, stabilizes the multiplier dynamics, and reduces phase lag in constraint regulation.

A detailed derivation is provided in Appendix A.

### 4.3 Extended State Observation for Multiplier Updates

Training in Safe RL is inherently noisy and nonstationary: the measured cost fluctuates due to stochastic transitions, estimation error, and abrupt policy changes. If the multiplier reacts to these raw signals directly, it amplifies noise and tends to oscillate. What is missing is an online estimate of the *unmodeled dynamics*,

the effective disturbance $f(x_1, x_2, t)$ in Eqn. 11, so that the update can distinguish genuine constraint trends from transient fluctuations.

To this end, we borrow the idea of an *extended state observer* (ESO) from adaptive control, but use it in the simplest reduced-order form (Zhong et al., 2020a) suitable for RL. The ESO maintains an auxiliary state $\xi$ that is updated alongside observed costs:

$$\begin{cases} \dot{\xi} = -\omega_o \xi - \omega_o^2 x_2 + \omega_o \lambda, \\ \hat{f} = \xi + \omega_o x_2, \end{cases} \tag{15}$$

where $\hat{f}$ serves as a running estimate of the disturbance and $\omega_o > 0$ is the observer gain, controlling how aggressively the estimate adapts. The $+\omega_o \lambda$ term accounts for the negative-feedback channel in Eqn. 11. Intuitively, $\hat{f}$ behaves like a bias-correction term that smooths the effect of noise and policy shifts before they reach the multiplier.

With this estimate, the multiplier update is designed to track the transient reference $r(t)$ using proportional, derivative, and disturbance feedback:

$$\lambda_t = k_{ap}(x_1 - r) + k_{ad}(x_2 - \dot{r}) + \hat{f} - \ddot{r}. \tag{16}$$

Substituting (15) into (16), we obtain the ADRC-based update law:

$$\lambda_t = \Big( (k_{ap} + \omega_o k_{ad})(x_1 - r) + (k_{ad} + \omega_o)(x_2 - \dot{r}) \\ + \omega_o k_{ap} \int_0^t (x_1(\tau) - r(\tau))d\tau - \ddot{r} \Big)_+, \tag{17}$$

where $(\cdot)_+ = \max(0, \cdot)$ projects the multiplier onto the nonnegative orthant, exactly as in the PID rule Eqn. 10. Since projection is non-expansive (Appendix C.3.6), it does not amplify multiplier perturbations. For readability, the frequency-domain analysis below is carried out on the unprojected law.

**Proposition 4.1.** *If the observer gain $\omega_o$ is held constant and the transient reference is set to $r(t) = d$ (i.e., no transient shaping), then the ADRC update* (17) *reduces to the PID rule in Eqn. 10, up to the initialization of the observer state, under the mapping*

$$K_p = k_{ap} + \omega_o k_{ad}, \qquad K_i = \omega_o k_{ap}, \qquad K_d = k_{ad} + \omega_o. \tag{18}$$

*Setting $\omega_o = 0$ further recovers a pure PD controller.*

*Proof.* When $r(t) = d$, we have $\dot{r} = \ddot{r} = 0$ and $x_1 - r = J_c - d = g(\pi_\theta)$, $x_2 - \dot{r} = \dot{g}(\pi_\theta)$. Substituting into (17) gives $\lambda_t = \big( (k_{ap} + \omega_o k_{ad}) g + (k_{ad} + \omega_o) \dot{g} + \omega_o k_{ap} \int_0^t g \, d\tau \big)_+$, which matches Eqn. 10 under the identification (18). □

*Remark* 4.2. The mapping (18) characterizes which PID controllers arise from a frozen ADRC update. For any $k_{ap}, k_{ad}, \omega_o \geq 0$, the resulting gains satisfy $K_p K_d = (k_{ap} + \omega_o k_{ad})(k_{ad} + \omega_o) \geq \omega_o k_{ap} = K_i$. Conversely, every triple $K_p, K_i, K_d \geq 0$ with $K_p K_d \geq K_i$ is attained: setting $\omega_o = K_d - k_{ad}$ and $k_{ap} = K_p - \omega_o k_{ad}$, the product $\omega_o k_{ap}$ varies continuously from $K_p K_d$ at $k_{ad} = 0$ to 0 at $k_{ad} = K_d$, so some $k_{ad} \in [0, K_d]$ yields $\omega_o k_{ap} = K_i$. Thus the frozen-ADRC family covers a broad class of PID gains, and Theorem 4.3 leverages this correspondence to compare ADRC against the matched PID controller from Proposition 4.1 (see also Remark C.1).

### 4.4 The Lower Bound of Optimal Parameters

A central challenge in Safe RL is parameter sensitivity: the same multiplier update rule can behave well in one environment but oscillate or diverge in another. This is particularly acute for PID-based Lagrangian methods, whose stability depends heavily on hand-tuned gains. To make ADRC practical in Safe RL, we aim for a principled condition that guarantees stability and bounded estimation error across environments, thereby removing the need for brittle manual tuning.

We characterize the uncertainty in the closed-loop dynamics of Eqn. 11 by bounding how disturbances can depend on the current state and vary over time. Following Zhong et al. (2020a), we consider

$$f(x_1, x_2, t) = h(x_1, x_2) + w(t),$$

$$\left|\frac{\partial h}{\partial x_1}\right| \le L_1, \left|\frac{\partial h}{\partial x_2}\right| \le L_2, |w(t)|, |\dot{w}(t)| \le L_3, \tag{19}$$

where $L_1$ and $L_2$ bound the sensitivity of the state-dependent component $h$ to the cost $x_1$ and its rate $x_2$, while $L_3$ bounds the magnitude and variation of the purely time-dependent residual fluctuation $w(t)$. Eqn. 19 does not require stationarity or a known model: $f$ may be unknown, nonlinear, and time-varying, and only its sensitivity is constrained. The constants $L_1$ and $L_2$ are estimated online from observed costs via finite differences (Eqn. 22). As shown by Zhong et al. (2020a), the computable lower bound for the ESO bandwidth depends only on $(L_1, L_2, k_{ap}, k_{ad})$; $L_3$ affects the residual estimation-error constants (see Appendix C.1) but is not required for online gain selection.

Within this class, the admissible observer gains $\omega_o$ are constrained by a characteristic polynomial manifold

$$\Omega = \left\{\omega \in \mathbb{R} \,\big|\, n_0\omega^4 + n_1\omega^3 + n_2\omega^2 + n_3\omega + n_4 = 0\right\}, \tag{20}$$

where the coefficients $n_i$ depend on $(k_{ap}, k_{ad})$ and the state-sensitivity constants $(L_1, L_2)$ only (the full expressions are given in Appendix C.1). We define

$$\bar{\omega}_o = \begin{cases} \max\{\omega \mid \omega \in \Omega\}, & \text{if } \Omega \ne \emptyset, \\ 0, & \text{otherwise.} \end{cases}$$

The final lower bound ensuring stability and bounded estimation error is therefore

$$\omega_o^* = \max\left\{\bar{\omega}_o, \ 0, \ \frac{L_1 - k_{ap}}{k_{ad}}, \ L_2 - k_{ad}\right\}. \tag{21}$$

Intuitively, this bound ensures that the ESO adapts quickly enough to track disturbances while remaining stable. In practice, this means practitioners only need to set $\omega_o > \omega_o^*$, removing the trial-and-error search that plagues PID tuning.

Beyond stability, we also analyze how fast and accurately ADRC reacts compared to classical integral updates. Let $e(t) = x_1(t) - r(t)$ be the tracking error, $\hat{f}_I = k_i \int_0^t e(\tau)\, d\tau$ the disturbance estimate from integral control, and $\hat{f}$ the ADRC estimate. To make the comparison meaningful, the integral baseline is parameterized by the *equivalent gains* of Proposition 4.1. That is, ADRC is compared against the very PID/integral controller it reduces to when the observer is frozen. Their estimation errors are

$$e_{f_I}(t) = \hat{f}_I - f(x_1, x_2, t), \qquad e_f(t) = \hat{f} - f(x_1, x_2, t).$$

Denote the Laplace transforms of $e$, $e_f$, $e_{f_I}$, $\hat{f}_I$, and $\hat{f}$ by $E(s)$, $E_f(s)$, $E_{f_I}(s)$, $\hat{F}_I(s)$, and $\hat{F}(s)$ respectively, and let $F(s)$ be the transform of the disturbance $f$. Then $G_{e_f}(s)$ and $G_{e_{f_I}}(s)$ are the transfer functions from $F(s)$ to $E_f(s)$ and $E_{f_I}(s)$.

We establish the following result:

**Theorem 4.3.** *Suppose $k_{ap} > 0$, $k_{ad} > 0$, $\omega_o > \omega_o^*$, and let the integral baseline be parameterized by the equivalent gains of Proposition 4.1, $(k_p, k_i, k_d) = (k_{ap} + \omega_o k_{ad}, \ \omega_o k_{ap}, \ k_{ad} + \omega_o)$. Then, for every frequency $\omega > 0$, ADRC Lagrangian achieves uniformly lower disturbance estimation error than integral control:*

$$\frac{|G_{e_f}(i\omega)|}{|G_{e_{f_I}}(i\omega)|} < 1.$$

*Moreover, write $\hat{F}(s) = H(s)\, F(s)$ and $\hat{F}_I(s) = H_I(s)\, F(s)$ for the channels through which each estimate follows the disturbance, and let $\varphi(\omega)$ and $\varphi_I(\omega)$ denote their respective phase lags at frequency $\omega$. Then, for every $\omega > 0$,*

$$\varphi_I(\omega) - \varphi(\omega) \ = \ \arg\!\left(k_{ap} - \omega^2 + i\, k_{ad}\, \omega\right) \ \in \ (0, \pi),$$

*i.e., ADRC Lagrangian also exhibits strictly smaller phase lag, at every frequency and with no additional condition on $\omega_o$.*

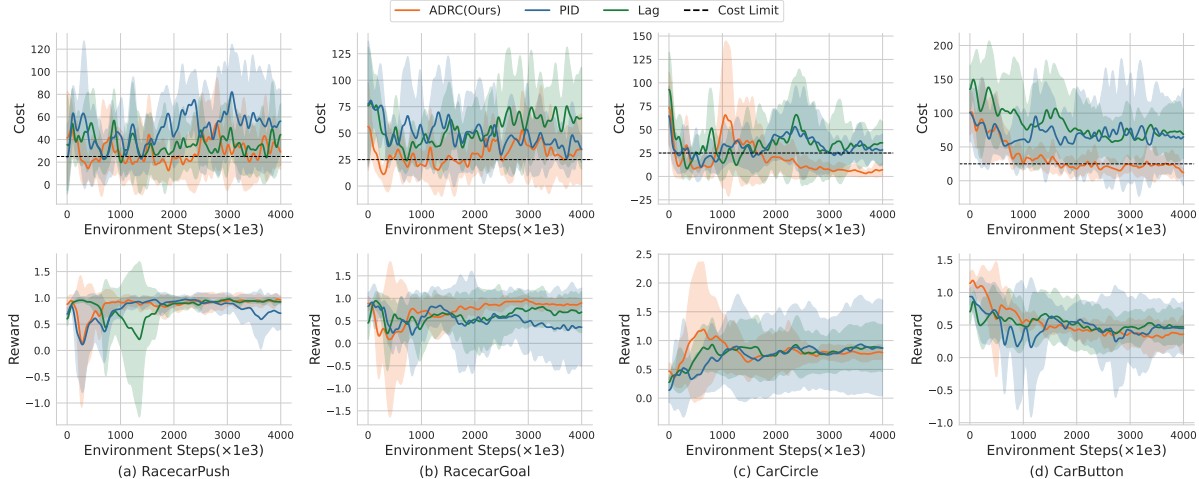

Figure 1: The training curves of PPO with various Lagrangian methods (denoted as CPPOLag, CPPOPID, CPPOADRC) across different tasks, showing episodic returns and costs over five random seeds. Solid lines represent mean values, while shaded areas denote one standard deviation across seeds. CPPOADRC demonstrates a shorter transient phase and lower costs compared to baselines, while achieving competitive rewards. Additional results are provided in Appendix F.

In words, the frequency-domain guarantees imply concrete benefits in Safe RL training. A smaller disturbance estimation error means the multiplier update is less sensitive to transient noise and policy-induced fluctuations, avoiding spurious reactions to single-batch variability. A smaller phase lag means constraint violations are corrected earlier, rather than several updates later, reducing the amplitude and duration of overshoot. Together, these properties yield training dynamics with fewer repeated safety violations, smoother multiplier trajectories, and faster convergence to the feasible region. The proof is provided in Appendix C.2.

**Stability Analysis under Surrogate Dynamics.** Appendix C.3 analyzes the ADRC multiplier update using a *surrogate* closed-loop model $x_1(t) = J_c(\pi(t))$ tracking $r(t) \to d$. By treating policy non-stationarity and noise as a lumped disturbance $f(t)$ within this simplified dynamic, we derive an ISS-type tracking bound $\mathcal{O}(L_f/\omega_o)$ and bounded time-average constraint violations. Crucially, we establish a scaling law linking the disturbance bound $L_f$ to Safe RL hyperparameters $(\delta, N, \Delta t)$, providing a principled basis for parameter tuning.

### 4.5 ADRC Lagrangian Methods in Safe RL

We now describe how ADRC Lagrangian methods are applied in Safe RL training. The central idea is to replace the hand-tuned, noise-sensitive dual update with an observer-augmented rule that adapts automatically to the evolving learning dynamics.

In practice, the Lagrangian multiplier $\lambda$ serves as a penalty knob: when the observed cumulative cost approaches the safety threshold, $\lambda$ should increase to discourage unsafe behavior; when costs fall well below the threshold, $\lambda$ can relax to allow more exploration. ADRC realizes this adaptivity by updating $\lambda$ according to Eqn. 17, where the observer $\hat{f}$ continuously estimates the effect of unmodeled disturbances and compensates for them in real time. This makes multiplier updates less myopic and more responsive than classical dual ascent or PID rules.

A key question is how to choose the observer gain $\omega_o$. As shown in Sec. 4.4, $\omega_o$ must exceed a lower bound $\omega_o^*$ that depends on the state-sensitivity constants $(L_1, L_2)$. Since $L_1$ and $L_2$ are unknown beforehand, we

Table 1: The proportion of constraint violations during training, the average violation magnitude, and the average cost for various algorithms. Our ADRC method consistently outperforms others; bold marks the best result in each column. Values are mean ± standard deviation over five random seeds.

| Algorithm | CarGoal | | | RacecarGoal | | |
|---|---|---|---|---|---|---|
| | Vio. Rate (%) | Magnitude | Avg. Cost | Vio. Rate (%) | Magnitude | Avg. Cost |
| CPPOLag | 65.20 ± 22.57 | 18.54 ± 11.41 | 40.05 ± 14.00 | 80.87 ± 19.17 | 31.18 ± 18.99 | 54.24 ± 21.01 |
| CPPOPID | 62.04 ± 18.18 | 12.95 ± 7.45 | 34.41 ± 9.04 | 72.30 ± 24.96 | 27.11 ± 15.58 | 49.02 ± 18.75 |
| CPPOADRC | **34.97 ± 13.88** | **4.72 ± 2.14** | **21.99 ± 5.32** | **47.08 ± 21.58** | **12.31 ± 9.34** | **30.12 ± 12.74** |
| DDPGLag | 52.44 ± 0.12 | 7.25 ± 0.93 | 28.35 ± 0.44 | 71.44 ± 9.47 | 18.47 ± 4.13 | 40.76 ± 5.00 |
| DDPGPID | 65.52 ± 4.60 | 12.53 ± 0.72 | 35.03 ± 0.53 | 72.05 ± 4.09 | 17.93 ± 2.81 | 40.29 ± 2.99 |
| DDPGADRC | **47.36 ± 1.90** | **2.88 ± 0.57** | **21.55 ± 0.29** | **68.41 ± 5.77** | **17.81 ± 1.34** | **39.41 ± 2.14** |
| TD3Lag | 70.47 ± 10.44 | 17.00 ± 1.29 | 38.90 ± 2.83 | 75.26 ± 2.13 | 19.93 ± 1.40 | 42.57 ± 1.59 |
| TD3PID | 80.94 ± 5.87 | 20.68 ± 3.94 | 43.43 ± 4.23 | 73.31 ± 6.06 | 19.19 ± 2.72 | 41.57 ± 3.22 |
| TD3ADRC | **40.62 ± 8.51** | **2.85 ± 0.31** | **20.65 ± 2.46** | **71.24 ± 3.00** | **17.55 ± 3.57** | **39.71 ± 3.94** |
| TRPOLag | 54.86 ± 6.74 | 10.46 ± 4.56 | 30.97 ± 4.69 | 64.31 ± 23.37 | 22.89 ± 18.69 | 43.67 ± 21.56 |
| TRPOPID | 44.79 ± 2.84 | 7.34 ± 1.31 | 25.84 ± 0.88 | 49.33 ± 15.00 | 11.70 ± 7.07 | 30.94 ± 8.81 |
| TRPOADRC | **29.11 ± 3.70** | **3.44 ± 1.21** | **20.48 ± 0.99** | **34.03 ± 8.06** | **6.16 ± 2.37** | **22.02 ± 3.61** |

estimate them online using finite differences of observed costs:

$$
\begin{aligned}
L_1 &\approx \max_t \left| \frac{\ddot{x}_1(t+1) - \ddot{x}_1(t)}{x_1(t+1) - x_1(t)} \right|, \\
L_2 &\approx \max_t \left| \frac{\ddot{x}_1(t+1) - \ddot{x}_1(t)}{x_2(t+1) - x_2(t)} \right|,
\end{aligned}
\tag{22}
$$

where $x_1$ is the cumulative cost and $x_2$ its derivative. These estimates allow us to adaptively compute $\omega_o$ via Eqns. 20 and 21, ensuring stability without manual tuning even as the training dynamics evolve, overcoming the parameter sensitivity and environment-specific retuning that plague PID-based and classical Lagrangian methods.

Finally, large values of $\lambda$ may destabilize policy learning by forcing aggressive gradient steps. Following the previous method (Stooke et al., 2020), we adopt a rescaled optimization objective:

$$
\theta^*(\lambda) = \arg\max_\theta \frac{1}{1+\lambda} \big( J(\pi_\theta) - \lambda J_c(\pi_\theta) \big),
$$

which tempers the effect of $\lambda$ while preserving constraint enforcement. This adjustment yields smoother policy updates and makes ADRC Lagrangian straightforward to integrate into standard Safe RL algorithms. The full training procedure is summarized in Algorithm 1.

Table 2: Performance comparison under RacecarGoal for TRPO and PPO with different $c_r$ values. Bold marks results that improve on the PID baseline.

| Method | TRPO (RacecarGoal) | | | PPO (RacecarGoal) | | |
|---|---|---|---|---|---|---|
| | Vio. Rate (%) | Magnitude | Avg. Cost | Vio. Rate (%) | Magnitude | Avg. Cost |
| Lag | 87.33 | 37.36 | 61.53 | 84.35 | 30.16 | 53.38 |
| PID | 44.60 | 7.04 | 26.15 | 79.25 | 23.88 | 46.44 |
| $c_r = 0.05$ | **33.98** | **5.25** | **20.83** | **34.38** | **3.90** | **22.45** |
| $c_r = 0.1$ | **29.05** | **3.44** | **18.95** | **33.08** | **5.78** | **21.22** |
| $c_r = 0.15$ | **31.25** | **5.34** | **21.16** | **52.88** | **10.69** | **31.37** |
| $c_r = 0.2$ | **40.65** | **6.10** | **23.67** | **48.95** | **8.77** | **26.91** |
| $c_r = 0.25$ | **38.50** | **6.71** | **23.40** | **62.83** | **13.37** | **33.99** |

# 5 Experiments

In this section, we conduct a series of experiments to evaluate the performance of our ADRC Lagrangian method. Specifically, we aim to answer the following questions: (1) Does it reduce training oscillations, having smaller phase-lag of the response, thereby minimizing constraint violations compared to baseline methods? (2) How robust is the method when facing different parameters that we set? (3) Does the ADRC Lagrangian update transfer across different Lagrangian-based safe RL algorithms, covering both on-policy and off-policy methods? (4) How does the ADRC-based Lagrangian method perform upon convergence? (5) How does our method compare against existing state-of-the-art Safe RL approaches?

We will address Questions 1 and 3 in Section 5.2, Question 2 in Section 5.3, Question 4 in Section 5.6 (with extended results in Appendix F.9), and Question 5 in Section 5.5 (with extended results in Appendix F.4).

## 5.1 Experimental Setups

**Environments**   We use OmniSafe (Ji et al., 2024) for our experiments, which provides a comprehensive and reliable benchmark for safe RL algorithms. We conduct our experiments using four safe RL algorithms with various combinations of agents and tasks. For more detail about the environment, please refer to Appendix E.2. Throughout the paper, wherever a tabulated result carries a $\pm$, it denotes one standard deviation over independent random seeds (five seeds per environment–algorithm pair for the main benchmark, three seeds for the velocity tasks in the appendix).

**Algorithms and Baseline**   We utilize two categories of algorithms that have been implemented in Omnisafe: on-policy methods (PPO, TRPO) and off-policy methods (DDPG, TD3). We use the classical Lagrangian method and the PID Lagrangian method as baselines.

**Hyperparameter Protocol**   To ensure a fair comparison, all methods share the same RL backbone hyperparameters taken directly from the official OmniSafe repository defaults without any method-specific tuning. For Lagrangian-specific parameters, our ADRC method directly inherits the OmniSafe default PID gains as $k_{ap}$ and $k_{ad}$; the additional parameters are the observer gain $\omega_o$, which is computed adaptively via Eqn. 21 rather than manually tuned, and the transient-process parameter $c_r$, which is fixed to 0.1 in all main experiments (Sec. 5.3.1 shows robustness over $c_r \in [0.05, 0.25]$). No additional hyperparameter search was performed for any method; the full parameter table is provided in Appendix D.1.

## 5.2 Performance Evaluation

Figure 1 illustrates the learning curves of PPO algorithms using various Lagrangian methods across different tasks. Our ADRC methods demonstrate superior constraint satisfaction and a shorter response lag while maintaining competitive reward compared to the baseline. To better compare with the baseline, Table 7 presents the violation rates, violation magnitude, and average cost during the training phase.

To illustrate the broad applicability of our ADRC Lagrangian update, which modifies only the multiplier update rule and is agnostic to the underlying policy optimizer, we conducted additional experiments across four backbone algorithms, covering both on-policy (TRPO, CPPO) and off-policy (DDPG, TD3) methods, in the RacecarGoal and CarGoal environments. The learning curves for the RacecarGoal tasks are presented in Figure 4, and the performance metrics are summarized in Table 1. The headline improvements quoted in the abstract correspond to the TD3 rows on CarGoal in this table: relative to TD3-Lag and TD3-PID, ADRC reduces the violation rate by 42–50% ($70.47\%/80.94\% \rightarrow 40.62\%$), the violation magnitude by 83–86% ($17.00/20.68 \rightarrow 2.85$), and the average cost by 47–52% ($38.90/43.43 \rightarrow 20.65$). Further experimental results across additional environments are provided in Appendix F.2.

## 5.3 Parameter Sensitivity Analysis

To demonstrate whether our ADRC Lagrangian methods are robust to different values of parameters, we test the parameter $c_r$ in Section 5.3.1 and the control gains $k_{ap}$ and $k_{ad}$ in Section 5.3.2. Sensitivity experiments use a fixed random seed for controlled comparison.

Table 3: Sensitivity analysis of tuning parameters $k_{ap}$ and $k_{ad}$ in the RacecarGoal environment.

| | Varying $k_{ap}$ | | | | Varying $k_{ad}$ | | |
|---|---|---|---|---|---|---|---|
| Setting | Vio. Rate (%) | Mag. | Avg. Cost | Setting | Vio. Rate (%) | Mag. | Avg. Cost |
| $k_{ap} = 1$ | **69.98** | **18.62** | **39.87** | $k_{ad} = 1$ | **28.55** | **6.17** | **20.23** |
| $k_{ap} = 0.1$ | **33.08** | **5.78** | **21.22** | $k_{ad} = 0.1$ | **38.25** | 9.12 | **23.92** |
| $k_{ap} = 0.01$ | **20.43** | **2.50** | **15.42** | $k_{ad} = 0.01$ | **39.08** | **5.75** | **23.33** |
| PID | 79.25 | 23.88 | 46.44 | PID | 44.60 | 7.04 | 26.15 |
| Lag | 84.35 | 30.16 | 53.38 | Lag | 87.33 | 37.36 | 61.53 |

### 5.3.1 Tuning Parameter $c_r$

We selected five different values for $c_r$ and conducted the experiments using TRPO and PPO under the RacecarGoal benchmark. As shown in Table 2, the results show that all selected values of the parameter $c_r$ outperform the baseline, demonstrating robustness to parameter variations. More experimental results can be found at Appendix F.5.3.

### 5.3.2 Tuning Parameters $k_{ap}$ and $k_{ad}$

We investigate the impact of the tuning parameters $k_{ap}$ and $k_{ad}$ under the RacecarGoal environment. For $k_{ap}$, experiments were conducted using PPO; for $k_{ad}$, we used TRPO. In both cases, we evaluated three orders of magnitude: 1, 0.1, and 0.01. As shown in Table 3, every tested value of $k_{ap}$ and $k_{ad}$ outperforms the PID and classical Lagrangian baselines on violation rate and average cost, with consistent magnitude improvements across the parameter range. For additional experimental details, please refer to Appendices F.5.1 and F.5.2.

## 5.4 Ablation Study

To evaluate the effectiveness of our proposed dynamic parameter adjustment and transient process, we conducted ablation studies, with results summarized in Table 4. In this table, "Delete $r(t)$" refers to the removal of the transient reference trajectory $r(t)$ (the multiplier then tracks the constant threshold $d$ directly), while "Delete $\omega_o$" refers to disabling the adaptive observer gain $\omega_o$ in the algorithm. The results show that removing either component results in a clear performance degradation in terms of violation rate, violation magnitude, and average cost. However, even with these removals, the performance of our approach remains superior to the baseline PID method, demonstrating the robustness of our framework. Additionally, the complete ADRC method achieves the best results across all metrics, further highlighting the significance of combining both $r(t)$ and $\omega_o$ in achieving optimal performance. For further details and results, please refer to Appendix F.7.

Table 4: Ablation study of CPPO algorithm under RacecarGoal. Bold marks the best result.

| Method | Vio. Rate(%) | Magnitude | Avg. Cost |
|---|---|---|---|
| Delete $r(t)$ | 54.08 | 15.23 | 34.66 |
| Delete $\omega_o$ | 65.40 | 13.99 | 36.38 |
| ADRC (Ours) | **33.08** | **5.78** | **21.22** |
| PID | 79.25 | 23.88 | 46.44 |
| Lag | 84.35 | 30.16 | 53.38 |

## 5.5 Comparison with State-of-the-Art Safe RL Algorithms

To test whether the ADRC multiplier update also improves safety when combined with other safe RL algorithms, we compare against both Lagrangian-based methods (RCPO, PDO) (Tessler et al., 2018; Chow et al., 2018a) and non-Lagrangian approaches (CUP, IPO) (Yang et al., 2022; Liu et al., 2019) on velocity-

control tasks. As detailed in Appendix F.4, ADRC variants reduce violation rates and average costs across these backbones while maintaining competitive reward.

### 5.6 Convergence Performance Analysis

To assess the final performance of the trained policies rather than intermediate training behavior, we conducted experiments on the Swimmer and Hopper environments from the Velocity tasks suite. The results compare ADRC-based and PID-based methods under CPPO and TRPO frameworks. Table 5 reports the Swimmer results; the corresponding Hopper results are provided in Appendix F.9.

Table 5: Final performance on the Swimmer environment. Bold marks the better value between PID and ADRC for each algorithm.

| Algorithm | Avg Reward | Avg Cost | Vio. Rate (%) |
|---|---|---|---|
| CPPOPID | **30.10** | 22.44 | 28.02 |
| CPPOADRC | 29.39 | **16.77** | **14.16** |
| TRPOPID | 28.72 | 21.34 | 37.85 |
| TRPOADRC | **36.32** | **19.03** | **12.16** |

As shown in Table 5, ADRC consistently improves constraint satisfaction over PID under both CPPO and TRPO, yielding lower average cost and violation rate; under TRPO, it also achieves a substantially higher average reward.

## 6 Conclusion

In this paper, we introduce an effective method to optimize the Lagrangian multiplier update process in safe RL, reducing oscillation during training. First, we model the safe RL learning process as a closed-loop system. Next, we introduce ADRC, a robust and innovative controller that estimates and compensates for overall disturbances. We consider the current cost as the control objective and design a second-order closed-loop system to regulate this cost, ensuring compliance with the safety constraint. Additionally, we employed a reduced-order ESO (Zhong et al., 2020a) to estimate the unknown nonlinear function affecting agent costs, and showed that frozen instances of our update law are exactly PID Lagrangian rules, with a precise characterization of the PID gains that arise as special cases of our approach. Theoretical analysis and extensive experiments across 15 environment–algorithm pairs support the effectiveness of our method.

**Limitations and future work.** Our analysis assumes the multiplier loop observes the episodic cost without dead time: each dual update reacts to the cost generated by the current policy. Two delay regimes should be distinguished. Delays *within* an episode (unsafe actions whose cost materializes only later in the same episode) are absorbed by the episodic aggregation $J_c$ and introduce no delay into the multiplier loop, although they do complicate credit assignment for the policy gradient, an orthogonal issue shared by all compared methods. In contrast, *cross-iteration* dead time, where the measured cost responds to a policy change only several iterations later (for instance under asynchronous evaluation or replay buffers mixing stale policies), introduces additional lag that classical ADRC handles via reduced observer bandwidth or predictive, delay-compensated variants (Zhao & Gao, 2014). Empirically, the off-policy setting (DDPG and TD3 in Table 1), where replay mixing acts as a mild effective delay, still shows consistent if smaller gains, suggesting that delay erodes rather than reverses the advantage; extending the ESO with explicit delay compensation is a natural next step. In addition, our evaluation metrics are expectation-based; tail-sensitive criteria such as the CVaR of episodic cost (e.g., the mean cost of the worst 1% of episodes) would clarify whether the improvements extend to rare but severe unsafe episodes, and we leave such evaluation to future work. Finally, while our method is validated extensively in simulated environments, applying it to real-world robotics or safety-critical systems remains an important open direction.

## Impact Statement

This paper presents work whose goal is to advance the field of safe reinforcement learning by improving the stability and robustness of Lagrangian-based constraint enforcement via an ADRC-inspired update rule. The primary positive impact of this work is to reduce safety violations during training and improve constraint satisfaction, which may benefit safety-critical applications such as robotics and autonomous systems by enabling more reliable policy learning.

Potential risks are mainly related to misuse and deployment overconfidence. While improved constraint regulation can support safer behavior, any method that accelerates or stabilizes reinforcement learning could also be applied to domains with harmful intent. Moreover, our approach is evaluated on simulated benchmarks, and real-world deployment may face distribution shift, sensor noise, and unmodeled dynamics beyond those captured in our experiments; careless application could lead to unsafe outcomes. We therefore emphasize that our results should not be interpreted as a guarantee of real-world safety without additional validation, monitoring, and fail-safe mechanisms.

We do not introduce new datasets containing personal data, and our method operates on standard RL training signals. The computational footprint is comparable to existing Lagrangian and PID-style methods, with the added observer computation being lightweight. Future work should investigate stronger robustness guarantees under real-world disturbances and incorporate evaluation protocols that better reflect safety-critical deployment conditions.

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

## Appendix Contents

## A    Derivation of the Transient Reference Trajectory $r(t)$

This appendix derives the closed-form reference trajectory Eqn. 14 used by the transient process in Sec. 4.2. We consider the ODE:

$$\ddot{r} = -2c_r\dot{r} - c_r^2(r - d), \quad r(0) = x_1(0), \quad \dot{r}(0) = x_2(0), \tag{23}$$

which can be rewritten in the standard form:

$$\ddot{r} + 2c_r\dot{r} + c_r^2 r = c_r^2 d. \tag{24}$$

First, we solve the associated homogeneous equation:

$$\ddot{r} + 2c_r\dot{r} + c_r^2 r = 0. \tag{25}$$

Assuming a solution of the form $r_h(t) = e^{\mu t}$ and substituting into Eqn. 25, we obtain the characteristic equation:

$$\mu^2 + 2c_r\mu + c_r^2 = 0. \tag{26}$$

Solving for $\mu$ yields:

$$\mu = -c_r. \tag{27}$$

Thus, the general solution for the homogeneous equation is:

$$r_h(t) = (A + Bt)e^{-c_r t}, \tag{28}$$

where $A$ and $B$ are constants determined by the initial conditions.

For the nonhomogeneous equation Eqn. 24, we assume a particular solution $r_p(t) = C$. Substituting into Eqn. 24 gives:

$$C = d. \tag{29}$$

The general solution to Eqn. 24 is:

$$r(t) = (A + Bt)e^{-c_r t} + d. \tag{30}$$

To determine $A$ and $B$, we use the initial conditions. From $r(0) = x_1(0)$:

$$A + d = x_1(0) \quad \Rightarrow \quad A = x_1(0) - d. \tag{31}$$

The derivative $\dot{r}(t)$ is:

$$\dot{r}(t) = \left(B - c_r(A + Bt)\right)e^{-c_r t}. \tag{32}$$

Substitute $t = 0$ into Eqn. 32 and use $\dot{r}(0) = x_2(0)$:

$$B - c_r A = x_2(0). \to B = x_2(0) + c_r(x_1(0) - d). \tag{33}$$

Substitute $A$ and $B$ into Eqn. 30 to obtain the final solution:

$$r(t) = d + (x_1(0) - d)e^{-c_r t} + (x_2(0) + c_r(x_1(0) - d))\,te^{-c_r t}. \tag{34}$$

## B    From the ESO to the ADRC Update Law

This appendix derives the ADRC update law Eqn. 17 in Sec. 4.3 by substituting the ESO Eqn. 15 into the control law. We consider the control law:

$$\lambda = k_{ap}(x_1 - r) + k_{ad}(x_2 - \dot{r}) + \hat{f} - \ddot{r}, \quad k_{ap} > 0, \quad k_{ad} > 0, \tag{35}$$

where $k_{ap}$ and $k_{ad}$ are tuning parameters, and the term $\hat{f}$ compensates for disturbances.

Substitute Eqn. 35 into the Eqn. 15:

$$\dot{\xi} = -\omega_o \xi - \omega_o^2 x_2 + \omega_o \left( k_{ap}(x_1 - r) + k_{ad}(x_2 - \dot{r}) + \hat{f} - \ddot{r} \right). \tag{36}$$

Simplify Eqn. 36:

$$\dot{\xi} = -\omega_o \xi - \omega_o^2 x_2 + \omega_o k_{ap}(x_1 - r) + \omega_o k_{ad}(x_2 - \dot{r}) + \omega_o \hat{f} - \omega_o \ddot{r}. \tag{37}$$

Given $\hat{f} = \xi + \omega_o x_2$, we have $\xi = \hat{f} - \omega_o x_2$ and $\dot{\xi} = \dot{\hat{f}} - \omega_o \dot{x}_2$. Substitute this into Eqn. 37:

$$\dot{\hat{f}} - \omega_o \dot{x}_2 = -\omega_o(\hat{f} - \omega_o x_2) - \omega_o^2 x_2 + \omega_o k_{ap}(x_1 - r) + \omega_o k_{ad}(x_2 - \dot{r}) + \omega_o \hat{f} - \omega_o \ddot{r}. \tag{38}$$

Simplify further:

$$\dot{\hat{f}} = \omega_o k_{ap}(x_1 - r) + \omega_o k_{ad}(x_2 - \dot{r}) - \omega_o \ddot{r} + \omega_o \dot{x}_2. \tag{39}$$

Integrating Eqn. 39 and absorbing the constant of integration into the observer initialization $\hat{f}(0)$, we have:

$$\hat{f} = \omega_o k_{ad}(x_1 - r) + \omega_o(x_2 - \dot{r}) + \omega_o k_{ap} \int_0^t (x_1(\tau) - r(\tau))d\tau. \tag{40}$$

Substitute Eqn. 40 back into Eqn. 35:

$$\lambda = (k_{ap} + \omega_o k_{ad})(x_1 - r) + (k_{ad} + \omega_o)(x_2 - \dot{r}) + \omega_o k_{ap} \int_0^t (x_1(\tau) - r(\tau))d\tau - \ddot{r}. \tag{41}$$

As in Eqn. 17, the multiplier actually applied is the projection of this expression onto $\lambda \geq 0$, preserving the nonnegativity required of a Lagrange multiplier.

## C  Proofs and Theoretical Analysis

This appendix contains the stability conditions and constants behind the observer-gain lower bound (Sec. C.1), the proof of Theorem 4.3 (Sec. C.2), and an extended surrogate analysis with robustness and safety guarantees (Sec. C.3). The proof of Proposition 4.1 appears inline in the main text, immediately after its statement.

### C.1  Convergence and Error Bounds

For completeness, we provide the detailed stability conditions and error analysis that were summarized in Sec. 4.4. We work with the disturbance class of Eqn. 19, augmented with one additional regularity condition, $\lim_{t \to \infty} w(t) = k$ for some constant $k$, required by the asymptotic convergence statements below:

$$\mathcal{F} = \Big\{ f \ \Big| \ f(x_1, x_2, t) = h(x_1, x_2) + w(t),$$
$$\left| \frac{\partial h}{\partial x_1} \right| \leq L_1, \quad \left| \frac{\partial h}{\partial x_2} \right| \leq L_2, \tag{42}$$
$$|w(t)| \leq L_3, \quad |\dot{w}(t)| \leq L_3, \quad \lim_{t \to \infty} w(t) = k \Big\},$$

where $L_1, L_2$ bound state-dependent sensitivity of $h$, and $L_3$ bounds the magnitude and rate of purely time-dependent residual fluctuations $w(t)$.

**Stability manifold and lower bound.** To guarantee convergence, the observer gain $\omega_o$ must lie in a feasible region determined by the characteristic polynomial

$$\Omega = \left\{ \omega \in \mathbb{R} \ \middle| \ n_0\omega^4 + n_1\omega^3 + n_2\omega^2 + n_3\omega + n_4 = 0 \right\}, \tag{43}$$

where, following Zhong et al. (2020a), the coefficients depend only on $(k_{ap}, k_{ad}, L_1, L_2)$:

$$
\begin{aligned}
n_0 &= k_{ad}^2, \\
n_1 &= 2k_{ad}\big[k_{ad}(k_{ad} - L_2) - L_1\big], \\
n_2 &= 2k_{ad}(k_{ap} - L_1)(k_{ad} - L_2) + \big[k_{ad}(k_{ad} - L_2) - L_1\big]^2 - L_2^2 k_{ap}, \\
n_3 &= 2\big[k_{ad}(k_{ad} - L_2) - L_1\big](k_{ap} - L_1)(k_{ad} - L_2) - L_2^2 k_{ap}(k_{ad} - L_2), \\
n_4 &= (k_{ap} - L_1)^2(k_{ad} - L_2)^2.
\end{aligned}
\tag{44}
$$

Let

$$\bar{\omega}_o = \begin{cases} \max\{\omega \mid \omega \in \Omega\}, & \text{if } \Omega \neq \emptyset, \\ 0, & \text{otherwise.} \end{cases}$$

The admissible observer gains are then those satisfying

$$\omega_o > \omega_o^* = \max\left\{ \bar{\omega}_o, \ 0, \ \tfrac{L_1 - k_{ap}}{k_{ad}}, \ L_2 - k_{ad} \right\}. \tag{45}$$

As noted in Remark 1 of Zhong et al. (2020a), $\omega_o^*$ depends only on $(L_1, L_2, k_{ap}, k_{ad})$ and is independent of the exogenous disturbance $w(t)$, the initial conditions, and the reference signal. Therefore $L_3$ is not required to compute $\omega_o^*$.

Suppose $f \in \mathcal{F}$ and $\omega_o > \omega_o^*$. Then:

- (*Convergence*) For any initial condition and any cost limit $d \in \mathbb{R}$, the system converges:

$$\lim_{t \to \infty} x_1(t) = d, \qquad \lim_{t \to \infty} x_2(t) = 0.$$

- (*Bounded estimation error*) Let $e(t) = x_1(t) - r(t)$ be the tracking error and $e_f(t) = \hat{f} - f(x_1, x_2, t)$ the disturbance estimation error. Then there exist constants $\eta_1, \eta_2$, depending on $e_f(0), k_{ap}, k_{ad}, L_1, L_2, L_3, \dot{r}, \ddot{r}$, such that

$$|\ddot{e}(t) + k_{ad}\dot{e}(t) + k_{ap}e(t)| = |e_f(t)| \ \leq \ \eta_1 e^{-\omega_o t} + \tfrac{\eta_2}{\omega_o}, \quad t \geq 0.$$

The first result shows that as long as the observer gain exceeds the lower bound $\omega_o^*$, the cumulative cost $x_1$ converges to the constraint threshold $d$ under the deterministic ADRC assumptions. The second result shows that the estimation error is always bounded, decays over time, and can be reduced by choosing larger $\omega_o$. Although $L_3$ does not enter $\omega_o^*$, it appears in the constants $\eta_1, \eta_2$ of the estimation-error bound, characterizing the residual error induced by the time-varying disturbance component $w(t)$. These results provide the stability and convergence foundation for our method; the ISS-type tracking-tube analysis in Appendix C.3 extends them to the Safe RL setting with bounded time-average constraint violations. The detailed proofs follow directly from Zhong et al. (2020a) and related ADRC analyses.

## C.2   Proof of Theorem 4.3

*Proof.* Throughout the proof we use the sign conventions of the main text: $e = x_1 - r$ is the tracking error, and $e_f = \hat{f} - f$, $e_{f_I} = \hat{f}_I - f$ are the estimation errors. From Theorems demonstrated by Zhong et al. (2020a), we know that both $f$ and $\dot{f}$ are bounded. Substituting the control law $\lambda = k_{ap}(x_1 - r) + k_{ad}(x_2 - \dot{r}) + \hat{f} - \ddot{r}$ (Eqn. 16) into $\dot{x}_2 = f - \lambda$ gives the error dynamics:

$$\begin{cases} \dot{e} = e_d, \\ \dot{e}_d = -k_{ap}e - k_{ad}e_d - e_f. \end{cases} \tag{46}$$

Taking the second derivative of $e$, we have:

$$\ddot{e} = -k_{ap}e - k_{ad}\dot{e} - e_f, \tag{47}$$

or equivalently:

$$e_f = -(\ddot{e} + k_{ad}\dot{e} + k_{ap}e), \tag{48}$$

consistent with Eqn. 69 in Appendix C.3.

Applying the Laplace transform to Eqn. 48 under zero initial conditions, we obtain:

$$E(s) = -\frac{1}{s^2 + k_{ad}s + k_{ap}}E_f(s). \tag{49}$$

For the observer error, substituting $\dot{x}_2 = f - \lambda$ into the ESO (Eqn. 15) yields $\dot{\hat{f}} = -\omega_o(\hat{f} - f) = -\omega_o e_f$, so the dynamics of $e_f$ are:

$$\dot{e}_f = -\omega_o e_f - \dot{f}. \tag{50}$$

Taking the Laplace transform of Eqn. 50, we have:

$$E_f(s) = G_{e_f}(s)F(s), \quad G_{e_f}(s) = -\frac{s}{s + \omega_o}. \tag{51}$$

Similarly, the integral estimate is $\hat{f}_I = k_i \int_0^t e(\tau)\,d\tau$, so that $\hat{F}_I(s) = k_i E(s)/s$ and $E_{f_I}(s) = \hat{F}_I(s) - F(s)$, where, as stated in Theorem 4.3, the baseline gains are the equivalent gains of Proposition 4.1,

$$k_p = k_{ap} + \omega_o k_{ad}, \qquad k_i = \omega_o k_{ap}, \qquad k_d = k_{ad} + \omega_o. \tag{52}$$

Substituting Eqns. 49 and 51 and using the factorization $s^3 + k_d s^2 + k_p s + k_i = (s + \omega_o)(s^2 + k_{ad}s + k_{ap})$, which holds precisely under the mapping (52), we obtain:

$$E_{f_I}(s) = \frac{s^2 + k_d s + k_p}{s^2 + k_{ad}s + k_{ap}}E_f(s), \tag{53}$$

and the transfer function for $E_{f_I}(s)$ can be expressed as:

$$E_{f_I}(s) = G_{e_{f_I}}(s)F(s), \quad G_{e_{f_I}}(s) = -\frac{s^3 + k_d s^2 + k_p s}{(s + \omega_o)(s^2 + k_{ad}s + k_{ap})}. \tag{54}$$

The overall negative signs in $G_{e_f}$ and $G_{e_{f_I}}$ are an artifact of the convention $e_f = \hat{f} - f$; they are immaterial for the magnitude comparison below and cancel in the ratio (55).

For every $\omega > 0$, both transfer functions are nonzero and the common factor $-i\omega/(i\omega + \omega_o)$ cancels, so the ratio of the squared magnitudes of $G_{e_f}(i\omega)$ and $G_{e_{f_I}}(i\omega)$ is given by:

$$\frac{|G_{e_f}(i\omega)|^2}{|G_{e_{f_I}}(i\omega)|^2} = \frac{(k_{ap} - \omega^2)^2 + k_{ad}^2\omega^2}{(k_p - \omega^2)^2 + k_d^2\omega^2}. \tag{55}$$

We now show that this ratio is strictly less than one at every such frequency. Substituting the matched gains (52) and expanding the two parts of the difference between the denominator and the numerator,

$$(k_p - \omega^2)^2 - (k_{ap} - \omega^2)^2 = \omega_o k_{ad}\left(2k_{ap} + \omega_o k_{ad} - 2\omega^2\right), \tag{56}$$

$$\left(k_d^2 - k_{ad}^2\right)\omega^2 = \left(2k_{ad}\,\omega_o + \omega_o^2\right)\omega^2. \tag{57}$$

Adding the two lines, the cross terms $\mp 2\omega_o k_{ad}\omega^2$ cancel, leaving

$$\left[(k_p - \omega^2)^2 + k_d^2\omega^2\right] - \left[(k_{ap} - \omega^2)^2 + k_{ad}^2\omega^2\right] = \omega_o^2\omega^2 + \omega_o k_{ad}\left(2k_{ap} + \omega_o k_{ad}\right) > 0 \tag{58}$$

for all $\omega \geq 0$, since $\omega_o, k_{ap}, k_{ad} > 0$. Hence

$$\frac{|G_{e_f}(i\omega)|}{|G_{e_{f_I}}(i\omega)|} < 1 \qquad \text{for every frequency } \omega > 0. \tag{59}$$

The gap (58) also reveals the structure of the advantage: the constant term $\omega_o k_{ad}(2k_{ap} + \omega_o k_{ad})$ provides a uniform margin at all frequencies, while the term $\omega_o^2 \omega^2$ grows with frequency, so the observer's benefit is largest precisely for fast-varying disturbances.

*Remark* C.1. The comparison uses the matched parameterization (52): ADRC is evaluated against the PID/integral controller that it reduces to under Proposition 4.1, providing a like-for-like baseline that isolates the effect of the adaptive observer.

In the low-frequency limit, the ratio remains bounded away from one:

$$\lim_{\omega \to 0} \frac{|G_{e_f}(i\omega)|}{|G_{e_{f_I}}(i\omega)|} = \frac{k_{ap}}{k_{ap} + \omega_o k_{ad}} < 1, \tag{60}$$

so the advantage does not vanish for slowly varying disturbances.

This completes the first part of this theorem.

We now turn to the phase comparison. The relevant quantity is the *phase lag* with which each estimator follows the disturbance, i.e., the phase of the estimation channels from $F(s)$ to $\hat{F}(s)$ and $\hat{F}_I(s)$. From $E_f(s) = \hat{F}(s) - F(s)$ and Eqn. 51,

$$\hat{F}(s) = \left(1 - \frac{s}{s + \omega_o}\right) F(s) = \frac{\omega_o}{s + \omega_o} F(s) =: H(s) F(s), \tag{61}$$

and from $\hat{F}_I(s) = k_i E(s)/s$ together with Eqns. 49 and 51,

$$\hat{F}_I(s) = \frac{k_i}{(s + \omega_o)(s^2 + k_{ad}s + k_{ap})} F(s) =: H_I(s) F(s). \tag{62}$$

Both channels have unit DC gain: $H(0) = 1$, and $H_I(0) = k_i/(\omega_o k_{ap}) = 1$ by the matched gains (52). Hence both estimators track constant disturbances exactly; they differ in how promptly they follow time-varying ones. Let $\varphi(\omega)$ and $\varphi_I(\omega)$ denote the phase lags of $H$ and $H_I$ at frequency $\omega > 0$, i.e., the negatives of their continuous (unwrapped) phases, as is standard in frequency-response analysis. Evaluating at $s = i\omega$ and summing the lags of the individual factors,

$$\varphi(\omega) = \tan^{-1}\left(\frac{\omega}{\omega_o}\right), \qquad \varphi_I(\omega) = \tan^{-1}\left(\frac{\omega}{\omega_o}\right) + \arg(k_{ap} - \omega^2 + i\,k_{ad}\,\omega), \tag{63}$$

where arg denotes the principal argument. For every $\omega > 0$, the number $k_{ap} - \omega^2 + i\,k_{ad}\,\omega$ has strictly positive imaginary part, hence lies in the open upper half-plane and its argument lies in $(0, \pi)$ regardless of the sign of $k_{ap} - \omega^2$. Therefore

$$\varphi_I(\omega) - \varphi(\omega) = \arg(k_{ap} - \omega^2 + i\,k_{ad}\,\omega) \in (0, \pi) \qquad \text{for all } \omega > 0, \tag{64}$$

with no additional condition on $\omega_o$: the ADRC estimate follows the disturbance with strictly smaller phase lag than the matched integral estimate at every frequency. For $\omega^2 < k_{ap}$ the gap equals $\tan^{-1}\left(\frac{k_{ad}\,\omega}{k_{ap} - \omega^2}\right)$; it passes through $\pi/2$ at $\omega^2 = k_{ap}$ and approaches $\pi$ at high frequencies, where the first-order ADRC channel lags by less than $\pi/2$ while the third-order integral channel accumulates up to an extra half-cycle of delay.

This completes the second part of this theorem.

$$\square$$

### C.3 Surrogate Analysis of ADRC Cost Regulation

**Scope**   This section provides a control-theoretic surrogate analysis for the cost-regulation channel induced by the ADRC multiplier update. We analyze the *population* discounted cost return

$$x_1(t) \; := \; J_c(\pi(t)),$$

where $\pi(t)$ is a smooth interpolation of the discrete policy iterates $\{\pi_{\theta_k}\}_{k \geq 0}$ produced by a trust-region backbone (TRPO/PPO). The guarantees below control the evolution of $x_1(t)$, the population discounted cost return. We derive robustness and bounded-violation guarantees under a standard ADRC disturbance-regularity envelope, and explicitly connect the disturbance envelope to Safe RL hyperparameters.

**Discrete training index and sampling interpretation.**   Let $k \in \{0, 1, 2, \dots\}$ index policy updates with update interval $\Delta t > 0$. We denote $x_{1,k} := x_1(k\Delta t) = J_c(\pi_{\theta_k})$ and the multiplier by $\lambda_k$. In implementation, $\lambda_k$ is held constant within iteration $k$ (zero-order hold), which corresponds to a piecewise-constant signal $\lambda(t) = \lambda_k$ for $t \in [k\Delta t, (k+1)\Delta t)$ in the surrogate analysis.

#### C.3.1   Surrogate cost channel and error dynamics

**Surrogate channel with correct feedback direction.**   In Lagrangian-based Safe RL, a larger multiplier $\lambda$ increases the penalty on cost in the primal objective, which acts as negative feedback on the cost return. Accordingly, we analyze the sign-normalized relative-degree-2 surrogate channel

$$\dot{x}_1(t) = x_2(t), \qquad \dot{x}_2(t) = f(t) \; - \; \lambda(t), \tag{65}$$

where $f(t)$ lumps all unmodeled effects (policy nonstationarity, approximation error of the abstraction, and sampling effects). Eqn. (65) captures the primal-dual interaction in a form amenable to control-theoretic analysis.

**Reference tracking and ADRC law.**   Let $r(t)$ be the critically damped reference trajectory from Sec. 4.2, converging to $d$ (or $d - \varepsilon$ for a margin variant). Define the tracking error and its derivative (consistent with the main text):

$$e(t) := x_1(t) - r(t), \qquad e_d(t) := \dot{e}(t) = x_2(t) - \dot{r}(t).$$

Under the ADRC design in Sec. 4.3, the multiplier is updated by

$$\lambda(t) = k_{ap}\big(x_1 - r\big) + k_{ad}\big(x_2 - \dot{r}\big) + \hat{f}(t) - \ddot{r}(t), \qquad k_{ap} > 0, \ k_{ad} > 0, \tag{66}$$

where $\hat{f}(t)$ is the ESO estimate of $f(t)$.

**Reduced-order ESO and estimation error dynamics.**   For the negative-feedback channel (65), the reduced-order ESO takes the sign-consistent form

$$\dot{\xi}(t) = -\omega_o \xi(t) - \omega_o^2 x_2(t) + \omega_o \lambda(t), \qquad \hat{f}(t) = \xi(t) + \omega_o x_2(t), \qquad \omega_o > 0. \tag{67}$$

Define the disturbance estimation error $e_f(t) := \hat{f}(t) - f(t)$. Combining (65) and (67) yields the standard ADRC error equation

$$\dot{e}_f(t) = -\omega_o e_f(t) - \dot{f}(t). \tag{68}$$

**Closed-loop tracking-error dynamics.**   Substituting (66) into (65) and using $e = x_1 - r$, $e_d = x_2 - \dot{r}$, we obtain

$$\ddot{e}(t) + k_{ad}\dot{e}(t) + k_{ap}e(t) = -e_f(t). \tag{69}$$

Thus the tracking channel is a stable second-order system driven by the estimation error $e_f(t)$.

### C.3.2 Disturbance regularity and an RL-grounded high-probability envelope

**Assumption C.2** (Disturbance regularity). The lumped disturbance $f(t)$ in (65) is absolutely continuous and satisfies

$$|\dot{f}(t)| \leq L_f$$

over the time interval of interest.

Assumption C.2 is standard in ADRC analyses: it formalizes that the total uncertainty seen by the observer cannot vary arbitrarily fast. We now justify that $L_f$ is *algorithmically controlled* in trust-region Safe RL and provide a finite-horizon high-probability envelope that links $L_f$ to $(\delta, N, \Delta t)$.

**Assumption C.3** (Bounded discounted cost return). There exists $B_c < \infty$ such that the discounted trajectory cost return $C(\tau) = \sum_{t=0}^{\infty} \gamma^t c_t$ satisfies $0 \leq C(\tau) \leq B_c$ almost surely.

**Assumption C.4** (Trust-region update). The backbone update satisfies a trust-region constraint $D_{\mathrm{KL}}(\pi_{\theta_k} \| \pi_{\theta_{k+1}}) \leq \delta$ (TRPO) or an implicit KL control induced by PPO clipping.

**Assumption C.5** (Population cost drift under trust regions). Under Assumptions C.3–C.4,

$$|J_c(\pi_{\theta_{k+1}}) - J_c(\pi_{\theta_k})| \leq \underbrace{\frac{2B_c}{1-\gamma}\sqrt{2\delta}}_{=:D_{\mathrm{TR}}}.$$

*Sketch.* The discounted performance-difference bound controls return differences by total-variation distance; Pinsker's inequality gives $\mathrm{TV} \leq \sqrt{D_{\mathrm{KL}}/2} \leq \sqrt{\delta/2}$, yielding the stated bound after collecting constants. □

**Lemma C.6** (Finite-horizon high-probability envelope for disturbance variation). *Assume (i) $\lambda_k \in [0, \lambda_{\max}]$ (projection and/or a max-penalty cap as in implementation), (ii) the cost estimate $\widehat{J}_c(\pi_{\theta_k})$ is computed from $N$ i.i.d. trajectories, and (iii) Assumptions C.3–C.4 hold. Fix a horizon $K$ and confidence $\eta \in (0,1)$, and define the (empirical) second-difference proxy*

$$\widehat{f}_k := \frac{\widehat{J}_c(\pi_{\theta_{k+1}}) - 2\widehat{J}_c(\pi_{\theta_k}) + \widehat{J}_c(\pi_{\theta_{k-1}})}{\Delta t^2} + \lambda_k.$$

*Then, with probability at least $1 - \eta$,*

$$\max_{1 \leq k \leq K-1} \frac{|\widehat{f}_{k+1} - \widehat{f}_k|}{\Delta t} \leq \frac{4}{\Delta t^3}\left(D_{\mathrm{TR}} + 2\varepsilon_N\right) + \frac{2\lambda_{\max}}{\Delta t}, \qquad \varepsilon_N := B_c\sqrt{\frac{\log(2K/\eta)}{2N}}.$$

*Consequently, on this event one may take $L_f = \mathcal{O}\left(\frac{\sqrt{\delta}}{\Delta t^3} + \frac{1}{\Delta t^3}\sqrt{\frac{\log(K/\eta)}{N}} + \frac{\lambda_{\max}}{\Delta t}\right)$ as a conservative envelope in Assumption C.2.*

*Sketch.* Hoeffding's inequality plus a union bound yields $|\widehat{J}_c(\pi_{\theta_k}) - J_c(\pi_{\theta_k})| \leq \varepsilon_N$ for all $k \leq K$ with prob. $\geq 1 - \eta$. Lemma C.5 bounds the population drift by $D_{\mathrm{TR}}$. Thus the per-iteration increment of the observed cost satisfies $|\widehat{J}_c(\pi_{\theta_{k+1}}) - \widehat{J}_c(\pi_{\theta_k})| \leq D_{\mathrm{TR}} + 2\varepsilon_N$, which bounds the third finite difference by $4(D_{\mathrm{TR}} + 2\varepsilon_N)$. The cap $\lambda_k \in [0, \lambda_{\max}]$ implies $|\lambda_{k+1} - \lambda_k| \leq 2\lambda_{\max}$. Combining these bounds yields the inequality. □

**Interpretation.** Lemma C.6 provides the missing bridge: the "disturbance speed" is controlled by the trust-region radius $\delta$ (or PPO clipping), batch size $N$, and update interval $\Delta t$, plus an explicit dependence on $\lambda_{\max}$ due to projection/capping.

### C.3.3 Robustness: ESO error bound and ISS-type tracking tube

**Lemma C.7** (ESO estimation-error bound). *Under Assumption C.2, the solution of (68) satisfies for all $t \geq 0$:*

$$|e_f(t)| \leq e^{-\omega_o t}|e_f(0)| + \frac{L_f}{\omega_o}. \tag{70}$$

**Theorem C.8** (ISS-type tracking bound). *Assume $k_{ap} > 0$, $k_{ad} > 0$ so that $s^2 + k_{ad}s + k_{ap}$ is Hurwitz. Let $h(t)$ be the impulse response of $H(s) = \frac{1}{s^2 + k_{ad}s + k_{ap}}$ and define $\|h\|_{L_1} := \int_0^\infty |h(\tau)|d\tau < \infty$. Then there exist constants $C_0 > 0$ and $\rho > 0$ (depending on $(k_{ap}, k_{ad})$ and initial conditions) such that*

$$|e(t)| \ \le \ C_0 e^{-\rho t} \ + \ \|h\|_{L_1} \left( e^{-\omega_o t}|e_f(0)| + \frac{L_f}{\omega_o} \right). \tag{71}$$

*In particular,*

$$\limsup_{t \to \infty} |e(t)| \ \le \ \|h\|_{L_1} \frac{L_f}{\omega_o} \ = \ \mathcal{O}\left( \frac{L_f}{\omega_o} \right). \tag{72}$$

**A computable constant.** If $k_{ad}^2 \ge 4k_{ap}$, then $h(t) \ge 0$ and $\|h\|_{L_1} = H(0) = 1/k_{ap}$, yielding the explicit tube radius $\limsup_{t \to \infty} |e(t)| \le \frac{1}{k_{ap}} \frac{L_f}{\omega_o}$.

### C.3.4   Safety: bounded time-average violation and eventual feasibility

**Theorem C.9** (Bounded time-average population violation). *Let $x_1(t) = J_c(\pi(t))$ and suppose $r(t) \to d$ as in Sec. 4.2. Under the conditions of Theorem C.8,*

$$\limsup_{T \to \infty} \frac{1}{T} \int_0^T \left( x_1(t) - d \right)_+ dt \ \le \ \|h\|_{L_1} \frac{L_f}{\omega_o}. \tag{73}$$

*Sketch.* Since $x_1 = r + e$,
$$(x_1 - d)_+ = (r + e - d)_+ \le (r - d)_+ + |e|.$$
Because $r(t) \to d$ exponentially, $\lim_{T \to \infty} \frac{1}{T} \int_0^T (r - d)_+ dt = 0$. The remaining term is controlled by the tracking tube in Theorem C.8. $\qquad\square$

**Corollary C.10** (Eventual feasibility with a fixed safety margin). *If the reference is chosen to converge to $d - \varepsilon$ with $\varepsilon > 0$ (replace $d$ by $d - \varepsilon$ in the transient process), and if $\varepsilon > \|h\|_{L_1} L_f/\omega_o$, then there exists $T_\varepsilon < \infty$ such that $x_1(t) \le d$ for all $t \ge T_\varepsilon$ in the surrogate channel.*

### C.3.5   Reward discussion: what ADRC preserves

ADRC modifies only the *dual* schedule $\{\lambda_k\}$ and does not change the primal trust-region mechanism. Therefore, for each fixed $\lambda_k$, standard TRPO/PPO analyses apply to the *penalized* objective $J(\pi) - \lambda_k J_c(\pi)$ (or the rescaled variant used in our implementation). In other words, ADRC does not invalidate the backbone optimizer's trust-region reasoning; it provides a smoother, less lagged multiplier signal, which empirically stabilizes the reward–cost tradeoff.

### C.3.6   Projection and saturation

In implementation, $\lambda_k$ is projected to $\lambda_k \ge 0$ (and optionally capped by $\lambda_{\max}$). Projection is non-expansive: for any $a, b \in \mathbb{R}$, $|\Pi_+(a) - \Pi_+(b)| \le |a - b|$ with $\Pi_+(x) = \max\{0, x\}$. Thus projection does not amplify multiplier perturbations; it only clamps the signal and ensures boundedness.

## D   Implementation Detail

This section outlines the details of the proposed method through the pseudo-code presented in Algorithm 1. The algorithm describes the procedure for adjusting the Lagrange multipliers using ADRC during training, ensuring robust performance and adaptability to varying conditions.

Two conventions in Algorithm 1 follow the standard PID Lagrangian implementation (Stooke et al., 2020): the derivative term $\partial$ and the integrator $I$ are each clipped at zero, in addition to the final projection of $\lambda$. These one-sided truncations, standard in PID Lagrangian implementations, preserve the non-expansive projection property established in Appendix C.3.6.

---

**Algorithm 1** ADRC-Controlled Lagrange Multiplier

---

**Require:** Chosen parameters $k_{ap}, k_{ad} > 0$
1: Integral: $I \leftarrow 0$
2: Previous Cost: $J_{C,\text{prev}} \leftarrow 0$
3: **repeat** at each iteration $t$
4:   Receive current cost $J_C$, reference cost $r$, its time derivative $\dot{r}$, $\ddot{r}$ and the optimal gain $\omega_o$.
5:   $\Delta \leftarrow J_C - r$
6:   $\partial \leftarrow (J_C - J_{C,\text{prev}} - \dot{r})_+$
7:   $I \leftarrow (I + \Delta)_+$
8:   $K_P \leftarrow k_{ap} + \omega_o k_{ad}$
9:   $K_I \leftarrow \omega_o k_{ap}$
10:   $K_D \leftarrow \omega_o + k_{ad}$
11:   $\lambda \leftarrow (K_P \Delta + K_I I + K_D \partial - \ddot{r})_+$
12:   $J_{C,\text{prev}} \leftarrow J_C$
13:   Apply $\lambda$ to the current policy update
14: **until** training terminates

---

### D.1   Hyper-parameters

For the on-policy algorithms TRPO and PPO, we adopt the default parameters provided by Omnisafe (Ji et al., 2024), as detailed in Table 6. These parameters are consistently applied across all tasks.

Table 6: Parameter Comparison: ADRC, PID, and Lag Methods

| Parameter | ADRC | PID | Lag |
|---|---|---|---|
| $k_p(k_{ap})$ | 0.1 | 0.1 | - |
| $k_i$ | - | 0.01 | 0.035 |
| $k_d(k_{ad})$ | 0.01 | 0.01 | - |
| $c_r$ | 0.1 | - | - |
| $\omega_o$ | adaptive (Eqn. 21) | - | - |
| Delay | 10 | 10 | - |
| EMA $\alpha$ (Proportional Term) | 0.95 | 0.95 | - |
| EMA $\alpha$ (Derivative Term) | 0.95 | 0.95 | - |
| Sum Normalization | True | True | - |
| Derivative Normalization | False | False | - |
| Cost Limit | 25.0 | 25.0 | 25.0 |
| Max Penalty Coefficient | 100.0 | 100.0 | - |
| Initial Lagrangian Multiplier | 0.001 | 0.001 | 0.001 |
| Hidden Layer Sizes (Actor) | [64, 64] | [64, 64] | [64, 64] |
| Activation Function (Actor) | tanh | tanh | tanh |
| Hidden Layer Sizes (Critic) | [64, 64] | [64, 64] | [64, 64] |
| Activation Function (Critic) | tanh | tanh | tanh |
| Critic Learning Rate | 0.0003 | 0.0003 | 0.0003 |
| Linear Learning Rate Decay | True | True | True |
| Clip Ratio | 0.2 | 0.2 | 0.2 |
| Target KL | 0.02 | 0.02 | 0.02 |
| Use Max Gradient Norm | True | True | True |
| Max Gradient Norm | 40.0 | 40.0 | 40.0 |

# E    Experimental Details

## E.1    Baseline and Hyperparameter Protocol

To comprehensively evaluate the effectiveness of our proposed ADRC method, we compare it against four well-established reinforcement learning algorithms. These include two off-policy algorithms, TD3 and DDPG, as well as two on-policy algorithms, PPO and TRPO. These algorithms were chosen due to their widespread adoption and proven performance across various RL tasks, providing a robust foundation for benchmarking.

**Fair comparison protocol.**    All methods (Classical Lagrangian, PID Lagrangian, and our ADRC Lagrangian) are evaluated under the same RL backbone hyperparameters taken directly from the official OmniSafe repository[1] without any method-specific tuning. For Lagrangian-specific parameters, the PID and Classical Lagrangian baselines use OmniSafe's official default gains. Our ADRC method directly inherits the OmniSafe PID defaults by setting $k_{ap} = k_p$ and $k_{ad} = k_d$; the additional parameters are the observer gain $\omega_o$, computed adaptively at each iteration via Eqn. 21 rather than manually tuned, and the transient-process parameter $c_r$, fixed to 0.1. Table 6 lists all shared and method-specific parameters.

*No additional hyperparameter search was performed for any method.* This protocol ensures that performance differences stem from the algorithmic design rather than from unequal tuning effort.

## E.2    Tasks Specification

To demonstrate the effectiveness and generalizability of our proposed methods, we conduct comprehensive experiments across diverse environments. We select three distinct agents, namely Car, Racecar, and Ant, each governed by different physical dynamics.

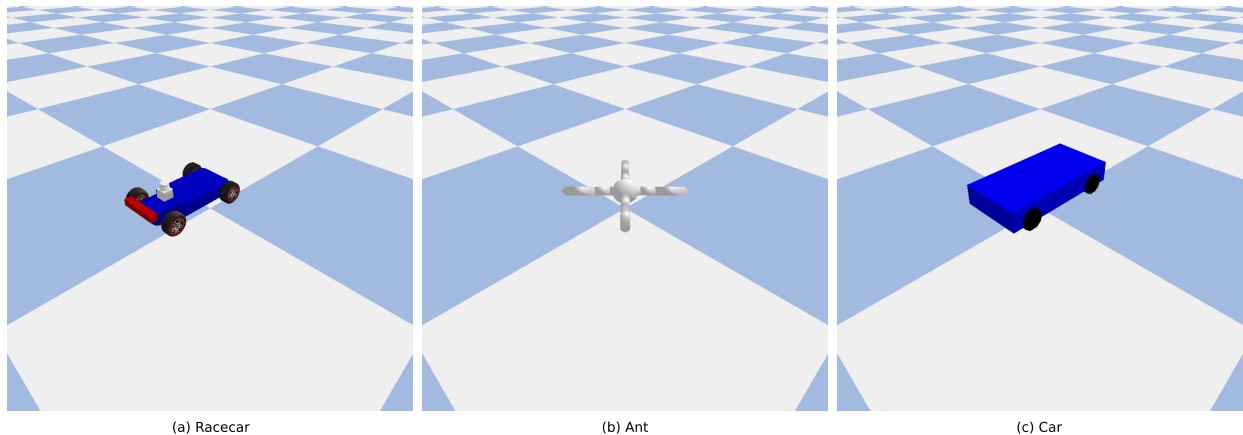

(a) Racecar                    (b) Ant                    (c) Car

Figure 2: Illustration of the three distinct agents used in our experiments. Car: A simple wheeled agent with low degrees of freedom. Racecar: A dynamic and agile wheeled agent with higher motion complexity. Ant: A multi-legged bionic agent with high degrees of freedom and non-linear dynamics. These agents represent diverse physical characteristics, allowing us to comprehensively evaluate the performance of our method under various physical dynamics.

As illustrated in Figure 2, the three agents represent diverse physical characteristics, enabling us to evaluate the performance of our method comprehensively across varying physical dynamics.

We consider four tasks in our experiments, as shown in Figure 3:

- **Goal Task** The robot must navigate to a specified goal region while avoiding hazards.

---

[1]https://github.com/PKU-Alignment/omnisafe

- **Button Task** The robot must press the correct button while avoiding hazards and gremlins, and must not press any wrong buttons.

- **Push Task** The robot must push a box to the goal region while avoiding hazards. A pillar is present but does not penalize collisions.

- **Circle Task** The robot moves around a circular track, without additional objects or hazards. This is mainly for testing circular navigation behavior.

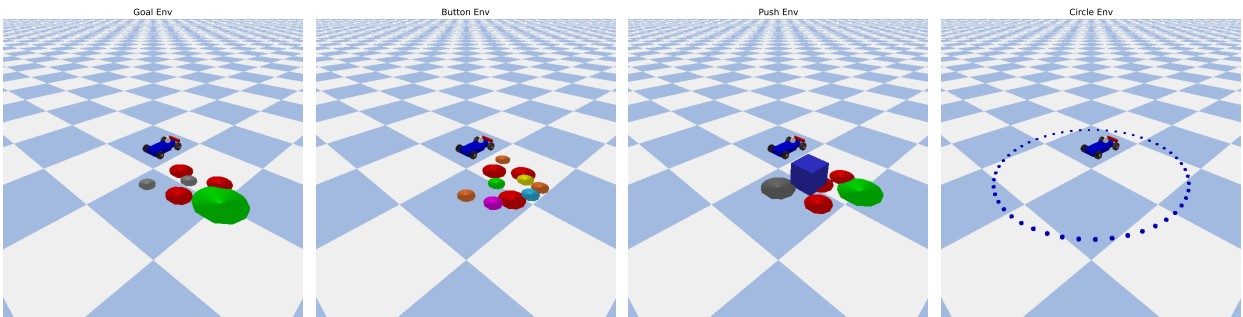

Figure 3: Four different tasks used in our experiments. (a) *Goal Task*: The agent must reach the goal area (blue sphere) without entering dangerous zones (red circles). (b) *Button Task*: The agent must press the correct button (green) and avoid pressing wrong ones (yellow, purple, etc.) or colliding with gremlins. (c) *Push Task*: The agent must push the box to the goal location (green circle) while avoiding hazards (red). (d) *Circle Env*: The agent moves around a simple circular track.

### E.3 Evaluation Metrics

To comprehensively assess the performance of the proposed reinforcement learning algorithms, we employ several evaluation metrics. These metrics evaluate both the agent's ability to minimize costs and its adherence to safety constraints.

**Reward and Cost** The primary performance metrics are the **reward** and **cost**, which respectively measure the benefits and penalties accumulated by the agent over the course of an episode. For an episode consisting of $T$ time steps:

- The **return reward**, $R$, is defined as:

$$R = \sum_{t=1}^{T} r_t,$$

where $r_t$ is the reward received at time step $t$. This metric reflects the agent's ability to achieve its objective efficiently.

- The **return cost**, $C$, is calculated as:

$$C = \sum_{t=1}^{T} c_t,$$

where $c_t$ is the cost incurred at time step $t$. This metric assesses the penalties associated with the agent's actions, capturing its safety and resource efficiency.

**Violation Rate (Vio. Rate)** The Violation Rate quantifies the proportion of episodes during training in which the agent breaches predefined safety constraints. It is expressed as:

$$\text{Vio Rate} = \frac{N_{\text{violations}}}{N_{\text{total\_episodes}}},$$

where $N_{\text{violations}}$ is the number of episodes in which the agent's cumulative cost $C$ exceeds the allowable threshold, and $N_{\text{total\_episodes}}$ is the total number of training episodes. A lower violation rate indicates better safety performance.

**Constraint Violation Magnitude (Magnitude)**   The Violation Magnitude measures the severity of constraint violations in episodes where breaches occur. It is calculated as the average amount by which the return cost exceeds the allowable threshold across all violating episodes:

$$\text{Violation Magnitude} = \frac{1}{N_{\text{violations}}} \sum_{i=1}^{N_{\text{violations}}} \max(0, C_i - d),$$

where $C_i$ is the return cost of the $i$-th violating episode and $d$ is the cost threshold that we set. Smaller magnitudes indicate less severe constraint violations.

**Average Cost (Avg. Cost)**   To evaluate the overall performance during training, we calculate the Average Cost across all episodes:

$$\text{Average Cost} = \frac{1}{N_{\text{total\_episodes}}} \sum_{i=1}^{N_{\text{total\_episodes}}} C_i,$$

where $C_i$ is the return cost of the $i$-th episode.

By analyzing these metrics, we can comprehensively assess the effectiveness of each algorithm in achieving a balance between reward maximization and safety constraint adherence.

## F   More Experimental Results

### F.1   Tables and Figures Referenced in the Main Text

Table 7: Constraint violation rate (Vio.), violation magnitude (Mag.), and average cost (Cost) during PPO training with various Lagrangian methods.

| Task | Method | Vio. (%) | Mag. | Cost |
|------|--------|----------|------|------|
| CarButton | Lag | 89.77 $\pm$ 19.38 | 59.77 $\pm$ 39.05 | 84.05 $\pm$ 40.18 |
| | PID | 85.09 $\pm$ 16.67 | 45.27 $\pm$ 46.80 | 68.95 $\pm$ 47.94 |
| | ADRC | **50.16 $\pm$ 17.08** | **14.80 $\pm$ 3.96** | **34.20 $\pm$ 5.75** |
| CarCircle | Lag | 46.74 $\pm$ 20.16 | 15.85 $\pm$ 17.40 | 33.54 $\pm$ 20.09 |
| | PID | 52.78 $\pm$ 17.62 | 12.29 $\pm$ 8.14 | 31.24 $\pm$ 10.31 |
| | ADRC | **21.35 $\pm$ 13.09** | **7.74 $\pm$ 6.46** | **18.85 $\pm$ 9.30** |
| RacecarGoal | Lag | 80.87 $\pm$ 19.17 | 31.18 $\pm$ 18.99 | 54.24 $\pm$ 21.01 |
| | PID | 72.30 $\pm$ 24.96 | 27.11 $\pm$ 15.58 | 49.02 $\pm$ 18.75 |
| | ADRC | **47.08 $\pm$ 21.58** | **12.31 $\pm$ 9.34** | **30.12 $\pm$ 12.74** |
| RacecarPush | Lag | 57.91 $\pm$ 22.12 | 15.45 $\pm$ 12.92 | 35.54 $\pm$ 15.56 |
| | PID | 70.84 $\pm$ 23.97 | 28.16 $\pm$ 16.84 | 49.67 $\pm$ 19.96 |
| | ADRC | **47.28 $\pm$ 17.05** | **12.50 $\pm$ 7.05** | **29.35 $\pm$ 11.03** |

As shown in Table 7, the ADRC methods significantly reduce violation rate and violation magnitude, indicating reduced oscillations and a shorter phase lag in response. The calculation of metrics is detailed in Appendix E.3.

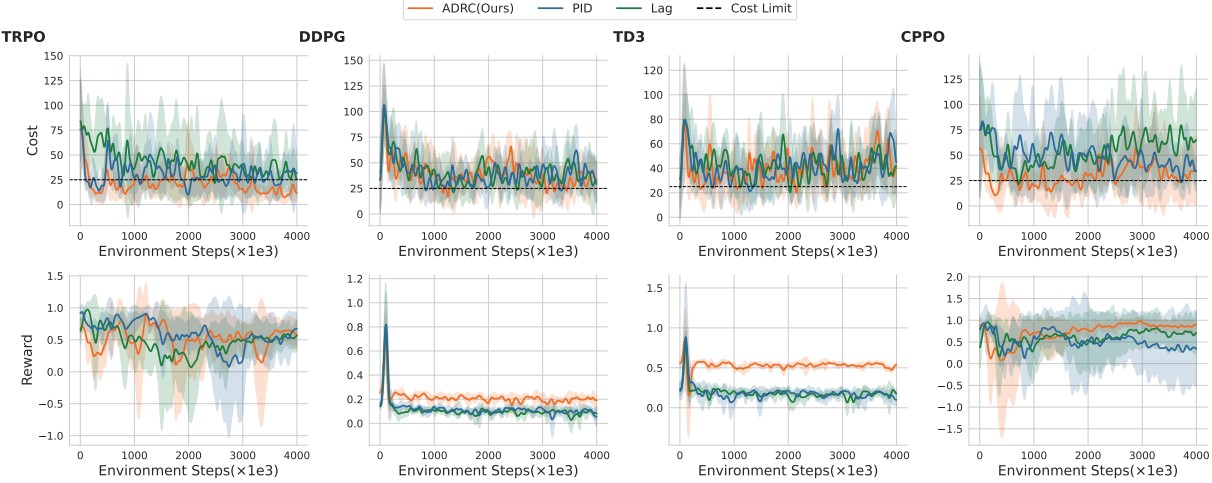

Figure 4: Training curves of RacecarGoal task.

Figure 4 shows training curves for different constraint-handling methods (ADRC, PID, and Classical La-grangian) across four RL algorithms (TRPO, DDPG, TD3, and PPO) in the RacecarGoal environment. Results show that ADRC consistently maintains cost below the limit while achieving competitive or superior rewards compared to other methods across different RL backbones.

## F.2 Main Results

To ensure clarity and readability, we present the training curves for each environment separately, avoiding the complexity of overlaying multiple curves on a single plot. This approach allows for a more intuitive comparison of performance across different settings. For a comprehensive evaluation of our method's effectiveness, we conducted experiments across three agents (Ant, Racecar, and Car) and four reinforcement learning tasks (Goal, Circle, Button, and Push). This setup resulted in a total of 12 experimental groups. For each group, we ran experiments with 5 different random seeds to account for variability and ensure statistical robustness. Furthermore, we benchmarked our method against four widely used reinforcement learning algorithms: TRPO, PPO, DDPG, and TD3, covering both on-policy and off-policy approaches. This rigorous experimental design provides a thorough validation of our method's adaptability and performance across diverse scenarios.

### F.2.1 Ant Environments

Figures 5 to Figure 8 present the training curves for the Ant environment across four tasks: Button, Circle, Goal, and Push. Each plot illustrates the episodic returns and costs averaged over five random seeds, with solid lines representing the mean and shaded areas denoting one standard deviation across seeds.

To provide a more thorough and quantitative evaluation of our method, we report the results of experiments conducted on four challenging environments, AntButton, AntCircle, AntPush and AntGoal in Table 8 and Table 9. The metrics compared include violation rate (%), magnitude of violations, and average cost. Across these experiments, our ADRC method outperforms or matches the baseline approaches (PID and Lagrange) in 83 of 96 pairwise comparisons (four environments, four RL algorithms, three metrics, two baselines). The exceptions concentrate on AntCircle and AntPush, where all methods already attain violation rates below 1% and the remaining differences amount to fractions of a cost unit. These results, validated across four RL algorithms (TRPO, PPO, DDPG, TD3), demonstrate the effectiveness and robustness of ADRC in handling constraint-aware reinforcement learning tasks.

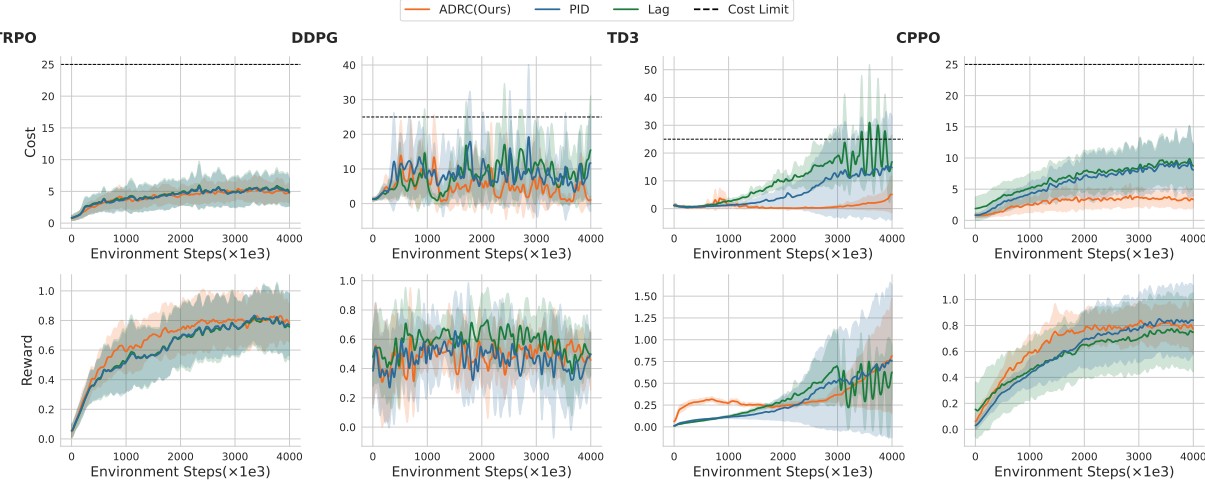

Figure 5: The training curves of AntButton with various Lagrangian methods across different algorithms.

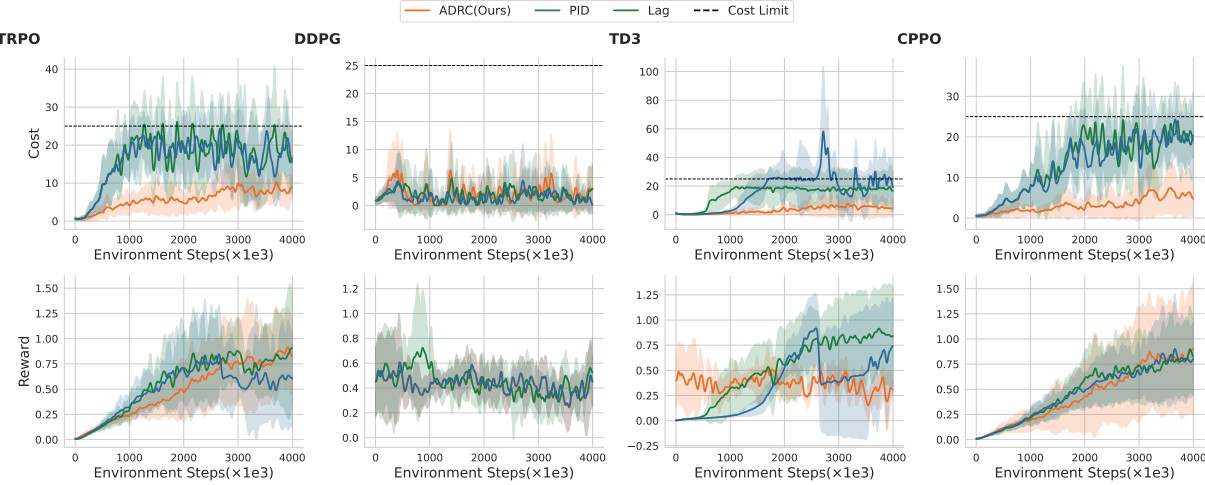

Figure 6: The training curves of AntCircle with various Lagrangian methods across different algorithms.

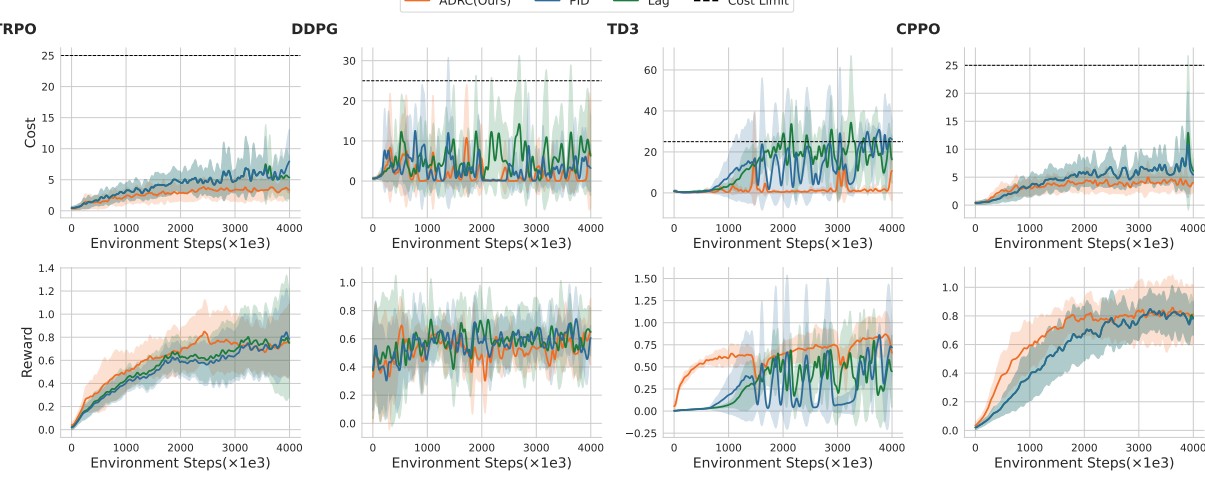

Figure 7: The training curves of AntGoal with various Lagrangian methods across different algorithms.

Table 8: Comparison of violation rate, magnitude, and average cost on AntButton and AntCircle. Bold marks ADRC results that improve on the PID Lagrangian baseline.

| Algorithm | AntButton | | | AntCircle | | |
|---|---|---|---|---|---|---|
| | Vio. Rate (%) | Magnitude | Avg. Cost | Vio. Rate (%) | Magnitude | Avg. Cost |
| CPPOLag | $0.22 \pm 0.03$ | $0.01 \pm 0.01$ | $6.09 \pm 3.94$ | $13.98 \pm 3.11$ | $6.59 \pm 1.89$ | $17.30 \pm 3.00$ |
| CPPOPID | $0.30 \pm 0.28$ | $0.01 \pm 0.01$ | $6.95 \pm 2.11$ | $10.82 \pm 3.77$ | $0.62 \pm 0.36$ | $12.95 \pm 0.92$ |
| CPPOADRC | $\mathbf{0.01 \pm 0.01}$ | $\mathbf{0.00 \pm 0.00}$ | $\mathbf{2.69 \pm 0.72}$ | $\mathbf{0.00 \pm 0.00}$ | $\mathbf{0.00 \pm 0.00}$ | $\mathbf{3.23 \pm 2.24}$ |
| DDPGLag | $5.72 \pm 4.67$ | $0.30 \pm 0.24$ | $7.35 \pm 2.97$ | $0.07 \pm 0.15$ | $0.00 \pm 0.00$ | $1.93 \pm 0.68$ |
| DDPGPID | $6.22 \pm 7.10$ | $0.47 \pm 0.65$ | $7.92 \pm 3.86$ | $0.04 \pm 0.07$ | $0.00 \pm 0.00$ | $1.59 \pm 0.50$ |
| DDPGADRC | $\mathbf{1.93 \pm 1.15}$ | $\mathbf{0.12 \pm 0.10}$ | $\mathbf{4.87 \pm 0.64}$ | $0.15 \pm 0.27$ | $0.00 \pm 0.00$ | $2.15 \pm 0.41$ |
| TD3Lag | $3.31 \pm 5.67$ | $0.32 \pm 0.64$ | $6.22 \pm 4.96$ | $31.26 \pm 2.86$ | $1.76 \pm 0.22$ | $15.02 \pm 0.99$ |
| TD3PID | $2.24 \pm 5.01$ | $0.11 \pm 0.25$ | $3.14 \pm 4.22$ | $21.32 \pm 11.16$ | $3.61 \pm 2.06$ | $13.74 \pm 4.75$ |
| TD3ADRC | $\mathbf{0.00 \pm 0.00}$ | $\mathbf{0.00 \pm 0.00}$ | $\mathbf{0.86 \pm 0.41}$ | $\mathbf{0.02 \pm 0.02}$ | $\mathbf{0.00 \pm 0.00}$ | $\mathbf{2.49 \pm 2.61}$ |
| TRPOLag | $0.01 \pm 0.02$ | $0.00 \pm 0.00$ | $4.40 \pm 1.48$ | $20.80 \pm 3.52$ | $1.64 \pm 0.37$ | $16.58 \pm 2.01$ |
| TRPOPID | $0.01 \pm 0.02$ | $0.00 \pm 0.00$ | $4.29 \pm 1.35$ | $17.73 \pm 2.21$ | $1.06 \pm 0.20$ | $16.14 \pm 1.31$ |
| TRPOADRC | $\mathbf{0.00 \pm 0.00}$ | $\mathbf{0.00 \pm 0.00}$ | $\mathbf{4.15 \pm 0.58}$ | $\mathbf{0.14 \pm 0.28}$ | $\mathbf{0.00 \pm 0.01}$ | $\mathbf{5.74 \pm 1.93}$ |

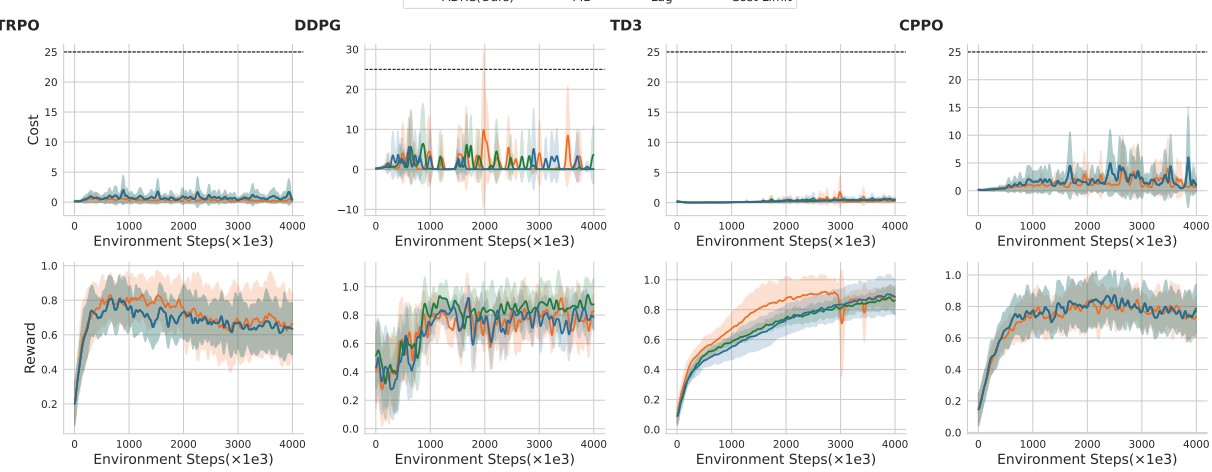

Figure 8: The training curves of AntPush with various Lagrangian methods across different algorithms.

Table 9: Comparison of violation rate, magnitude, and average cost on AntGoal and AntPush. Bold marks ADRC results that improve on the PID Lagrangian baseline.

| Algorithm | AntGoal | | | AntPush | | |
|---|---|---|---|---|---|---|
| | Vio. Rate (%) | Magnitude | Avg. Cost | Vio. Rate (%) | Magnitude | Avg. Cost |
| CPPOLag | $0.41 \pm 0.66$ | $0.05 \pm 0.01$ | $4.68 \pm 1.05$ | $0.48 \pm 0.48$ | $3.65 \pm 3.93$ | $0.61 \pm 0.55$ |
| CPPOPID | $0.31 \pm 0.46$ | $0.03 \pm 0.06$ | $4.63 \pm 0.96$ | $0.36 \pm 0.54$ | $0.02 \pm 0.03$ | $1.70 \pm 0.96$ |
| CPPOADRC | $\mathbf{0.00 \pm 0.00}$ | $\mathbf{0.00 \pm 0.00}$ | $\mathbf{3.43 \pm 0.47}$ | $\mathbf{0.09 \pm 0.13}$ | $\mathbf{0.00 \pm 0.01}$ | $\mathbf{1.25 \pm 0.37}$ |
| DDPGLag | $3.09 \pm 1.37$ | $0.32 \pm 0.20$ | $5.34 \pm 1.18$ | $0.03 \pm 0.04$ | $0.00 \pm 0.00$ | $1.05 \pm 0.47$ |
| DDPGPID | $1.42 \pm 0.58$ | $0.19 \pm 0.07$ | $3.03 \pm 1.01$ | $0.17 \pm 0.24$ | $0.02 \pm 0.02$ | $\mathbf{0.82 \pm 0.51}$ |
| DDPGADRC | $\mathbf{1.19 \pm 1.27}$ | $\mathbf{0.15 \pm 0.15}$ | $\mathbf{1.88 \pm 1.19}$ | $0.67 \pm 1.19$ | $0.11 \pm 0.20$ | $1.50 \pm 0.80$ |
| TD3Lag | $20.21 \pm 4.27$ | $1.94 \pm 0.64$ | $13.46 \pm 1.39$ | $0.00 \pm 0.00$ | $0.00 \pm 0.00$ | $\mathbf{0.12 \pm 0.07}$ |
| TD3PID | $18.74 \pm 12.17$ | $2.00 \pm 1.75$ | $11.81 \pm 5.51$ | $0.00 \pm 0.00$ | $0.00 \pm 0.00$ | $0.16 \pm 0.14$ |
| TD3ADRC | $\mathbf{1.54 \pm 1.50}$ | $\mathbf{0.20 \pm 0.22}$ | $\mathbf{2.04 \pm 1.34}$ | $0.00 \pm 0.00$ | $0.00 \pm 0.00$ | $0.18 \pm 0.16$ |
| TRPOLag | $0.25 \pm 0.50$ | $0.02 \pm 0.04$ | $4.10 \pm 0.89$ | $0.00 \pm 0.00$ | $0.00 \pm 0.00$ | $0.73 \pm 0.26$ |
| TRPOPID | $0.15 \pm 0.30$ | $0.01 \pm 0.02$ | $4.13 \pm 0.94$ | $0.00 \pm 0.00$ | $0.00 \pm 0.00$ | $0.73 \pm 0.26$ |
| TRPOADRC | $\mathbf{0.00 \pm 0.00}$ | $\mathbf{0.00 \pm 0.00}$ | $\mathbf{2.70 \pm 0.69}$ | $0.00 \pm 0.00$ | $0.00 \pm 0.00$ | $\mathbf{0.34 \pm 0.15}$ |

### F.2.2 Car Environments

Figures 9 to Figure 12 present the training curves for the Car environment across four tasks: Button, Circle, Goal, and Push. Each plot illustrates the episodic returns and costs averaged over five random seeds, with solid lines representing the mean and shaded areas denoting one standard deviation across seeds.

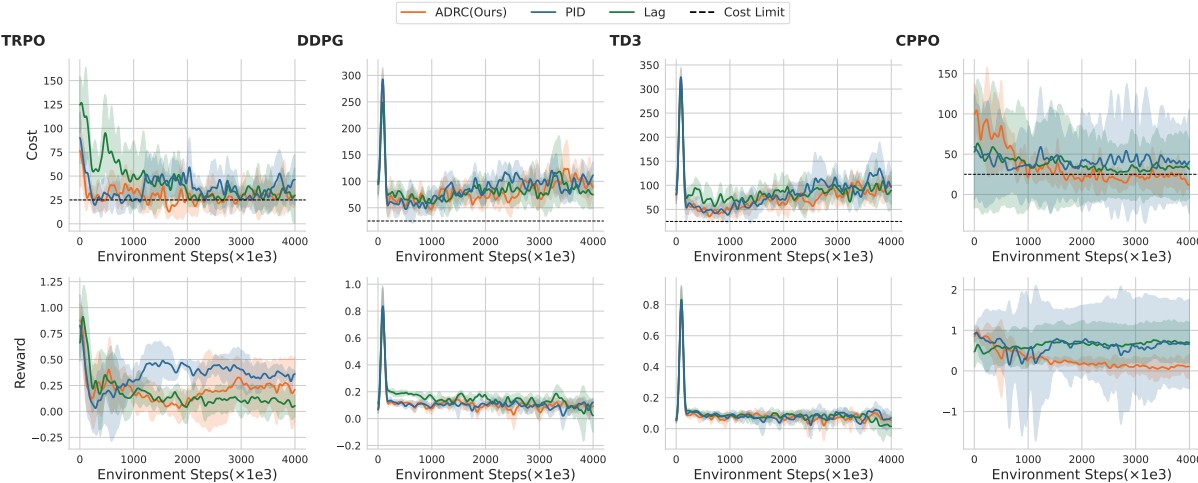

Figure 9: The training curves of CarButton with various Lagrangian methods across different algorithms.

To provide a more thorough and quantitative evaluation of our method, we report the results of experiments conducted on four challenging environments, CarButton, CarCircle, CarGoal, and CarPush in Table 10 and Table 11. The metrics compared include violation rate (%), magnitude of violations, and average cost. Across these experiments, our ADRC method outperforms or matches the baseline approaches (PID and Lagrange) in 90 of 96 pairwise comparisons. The exceptions are the violation rate under the off-policy algorithms on CarButton (where all methods saturate near 100%) and on CarCircle, together with two CarButton DDPG cells in which ADRC trails the classical Lagrangian by less than one unit.

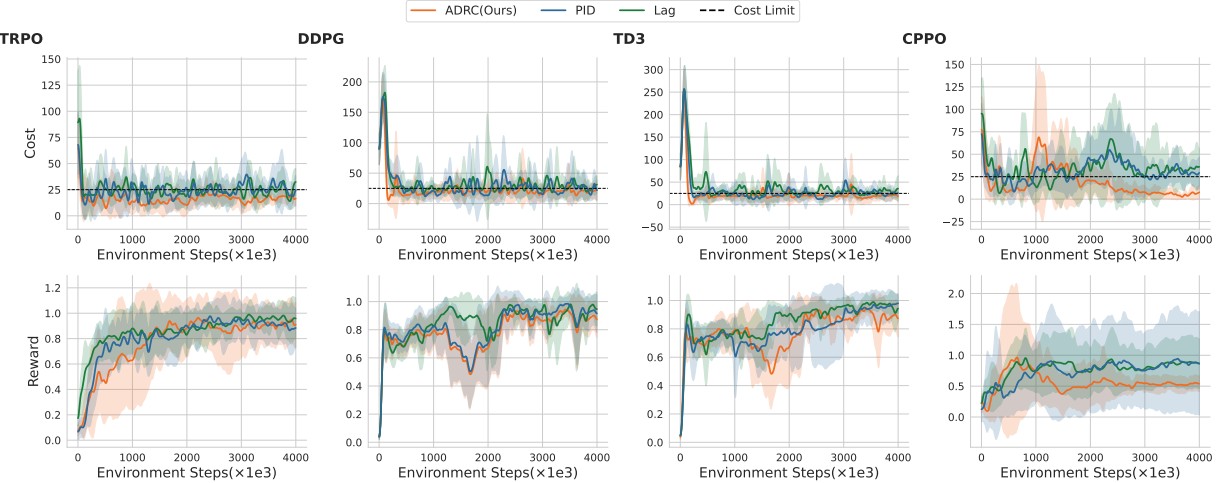

Figure 10: The training curves of CarCircle with various Lagrangian methods across different algorithms.

Table 10: Comparison of violation rate, magnitude, and average cost on CarButton and CarCircle. Bold marks ADRC results that improve on the PID Lagrangian baseline.

| Algorithm | CarButton | | | CarCircle | | |
|---|---|---|---|---|---|---|
| | Vio. Rate (%) | Magnitude | Avg. Cost | Vio. Rate (%) | Magnitude | Avg. Cost |
| CPPOLag | $89.77 \pm 19.38$ | $59.77 \pm 39.05$ | $84.05 \pm 40.18$ | $46.73 \pm 20.16$ | $15.85 \pm 17.40$ | $33.54 \pm 20.09$ |
| CPPOPID | $85.09 \pm 16.67$ | $45.27 \pm 46.80$ | $68.95 \pm 47.94$ | $52.77 \pm 17.62$ | $12.29 \pm 8.14$ | $31.24 \pm 10.31$ |
| CPPOADRC | $\mathbf{50.16 \pm 17.08}$ | $\mathbf{14.80 \pm 3.96}$ | $\mathbf{34.20 \pm 5.75}$ | $\mathbf{21.35 \pm 13.09}$ | $\mathbf{7.74 \pm 6.46}$ | $\mathbf{18.85 \pm 9.30}$ |
| DDPGLag | $99.93 \pm 0.11$ | $58.18 \pm 12.27$ | $83.17 \pm 12.27$ | $51.85 \pm 0.74$ | $13.55 \pm 2.86$ | $33.49 \pm 2.45$ |
| DDPGPID | $98.87 \pm 2.05$ | $64.10 \pm 6.43$ | $89.05 \pm 6.51$ | $39.77 \pm 4.31$ | $13.56 \pm 1.49$ | $30.89 \pm 1.44$ |
| DDPGADRC | $99.49 \pm 0.08$ | $\mathbf{58.62 \pm 7.25}$ | $\mathbf{83.60 \pm 7.25}$ | $51.50 \pm 5.10$ | $\mathbf{7.73 \pm 1.79}$ | $\mathbf{23.82 \pm 1.33}$ |
| TD3Lag | $99.97 \pm 0.04$ | $62.20 \pm 10.10$ | $87.20 \pm 10.11$ | $53.00 \pm 1.06$ | $17.18 \pm 1.25$ | $38.06 \pm 1.57$ |
| TD3PID | $99.03 \pm 1.00$ | $59.44 \pm 11.92$ | $84.40 \pm 11.95$ | $39.20 \pm 1.56$ | $12.52 \pm 1.64$ | $30.87 \pm 1.18$ |
| TD3ADRC | $99.04 \pm 0.73$ | $\mathbf{50.84 \pm 9.18}$ | $\mathbf{75.80 \pm 9.21}$ | $48.41 \pm 6.50$ | $\mathbf{7.49 \pm 1.54}$ | $\mathbf{24.26 \pm 1.28}$ |
| TRPOLag | $74.24 \pm 11.05$ | $21.90 \pm 7.38$ | $44.84 \pm 8.09$ | $40.04 \pm 1.30$ | $6.45 \pm 1.28$ | $25.52 \pm 0.51$ |
| TRPOPID | $69.04 \pm 17.53$ | $14.60 \pm 4.66$ | $37.11 \pm 6.10$ | $40.67 \pm 3.51$ | $6.58 \pm 1.15$ | $24.24 \pm 1.44$ |
| TRPOADRC | $\mathbf{53.28 \pm 15.44}$ | $\mathbf{8.53 \pm 4.33}$ | $\mathbf{29.57 \pm 6.18}$ | $\mathbf{17.71 \pm 2.92}$ | $\mathbf{1.92 \pm 0.64}$ | $\mathbf{16.22 \pm 1.75}$ |

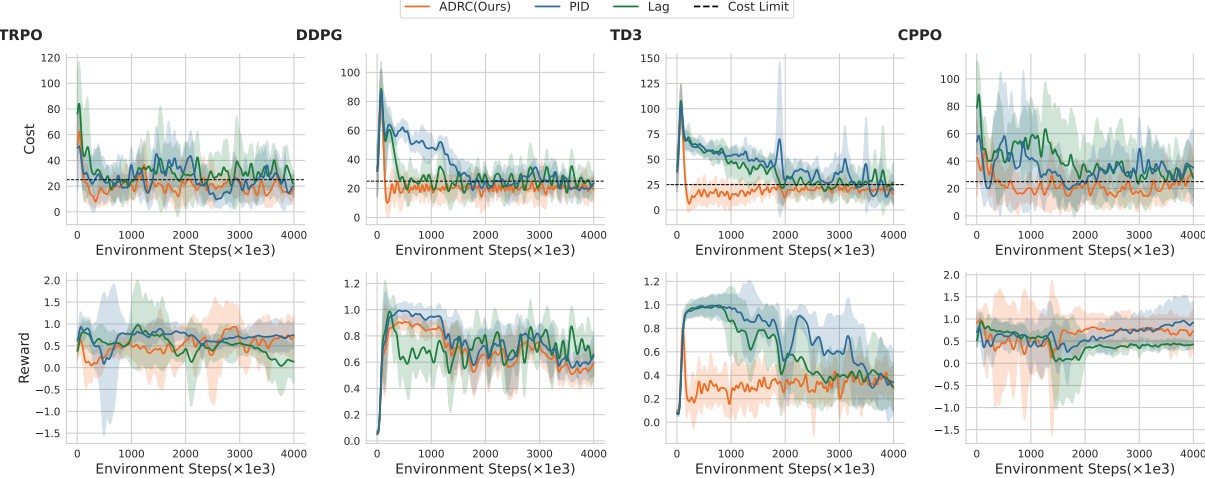

Figure 11: The training curves of CarGoal with various Lagrangian methods across different algorithms.

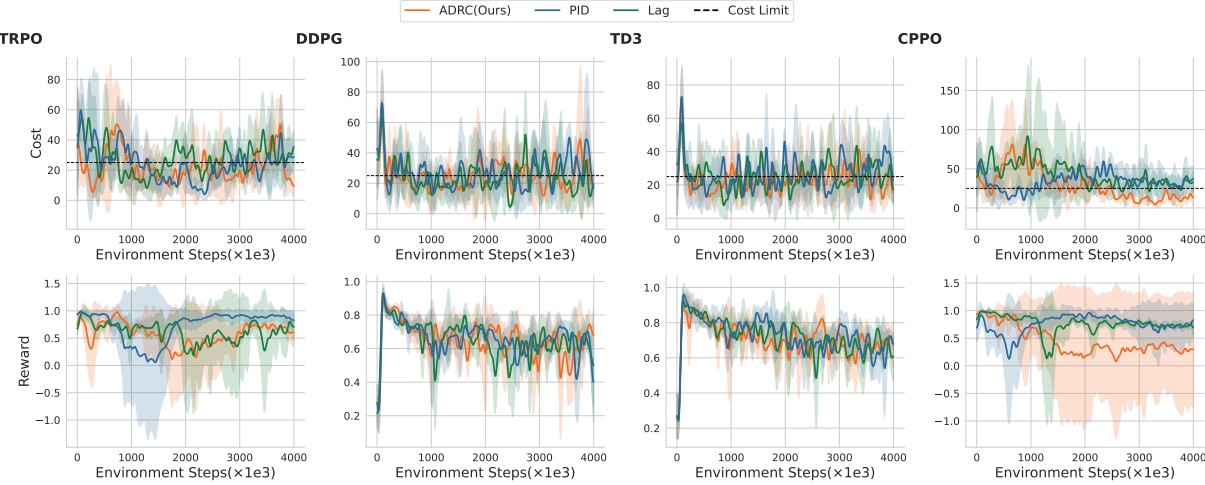

Figure 12: The training curves of CarPush with various Lagrangian methods across different algorithms

Table 11: Comparison of violation rate, magnitude, and average cost on CarGoal and CarPush. Bold marks ADRC results that improve on the PID Lagrangian baseline.

| Algorithm | CarGoal | | | CarPush | | |
|---|---|---|---|---|---|---|
| | Vio. Rate (%) | Magnitude | Avg. Cost | Vio. Rate (%) | Magnitude | Avg. Cost |
| CPPOLag | 65.20 ± 22.57 | 18.54 ± 11.41 | 40.05 ± 14.00 | 68.45 ± 18.98 | 21.48 ± 16.81 | 43.38 ± 18.71 |
| CPPOPID | 62.04 ± 18.18 | 12.95 ± 7.45 | 34.41 ± 9.04 | 62.36 ± 10.66 | 12.40 ± 3.08 | 33.74 ± 3.88 |
| CPPOADRC | **34.97 ± 13.88** | **4.72 ± 2.14** | **21.99 ± 5.32** | **42.23 ± 15.34** | **11.68 ± 7.99** | **29.16 ± 10.19** |
| DDPGLag | 52.44 ± 0.12 | 7.25 ± 0.93 | 28.35 ± 0.44 | 41.46 ± 1.76 | 6.95 ± 0.88 | 24.39 ± 0.85 |
| DDPGPID | 65.52 ± 4.60 | 12.53 ± 0.72 | 35.03 ± 0.53 | 48.81 ± 1.62 | 8.20 ± 1.38 | 27.37 ± 1.19 |
| DDPGADRC | **47.36 ± 1.90** | **2.88 ± 0.57** | **21.55 ± 0.29** | **42.43 ± 9.36** | **7.07 ± 1.69** | **25.70 ± 2.85** |
| TD3Lag | 70.47 ± 10.44 | 17.00 ± 1.29 | 38.90 ± 2.83 | 46.97 ± 2.20 | 6.27 ± 1.53 | 25.51 ± 1.36 |
| TD3PID | 80.94 ± 5.87 | 20.68 ± 3.94 | 43.43 ± 4.23 | 49.82 ± 0.77 | 7.65 ± 1.15 | 26.66 ± 1.14 |
| TD3ADRC | **40.62 ± 8.51** | **2.85 ± 0.31** | **20.65 ± 2.46** | **39.92 ± 1.66** | **5.15 ± 0.82** | **23.64 ± 0.93** |
| TRPOLag | 54.86 ± 6.74 | 10.46 ± 4.56 | 30.97 ± 4.69 | 48.24 ± 6.24 | 8.51 ± 2.35 | 27.58 ± 3.16 |
| TRPOPID | 44.79 ± 2.84 | 7.34 ± 1.31 | 25.84 ± 0.88 | 40.05 ± 1.69 | 6.66 ± 1.82 | 23.92 ± 1.74 |
| TRPOADRC | **29.12 ± 3.70** | **3.44 ± 1.21** | **20.48 ± 0.99** | **34.75 ± 8.43** | **6.32 ± 2.13** | **22.40 ± 2.34** |

### F.2.3  Racecar Environments

Figures 13 to Figure 16 present the training curves for the Racecar environment across four tasks: Button, Circle, Goal, and Push. Each plot illustrates the episodic returns and costs averaged over five random seeds, with solid lines representing the mean and shaded areas denoting one standard deviation across seeds.

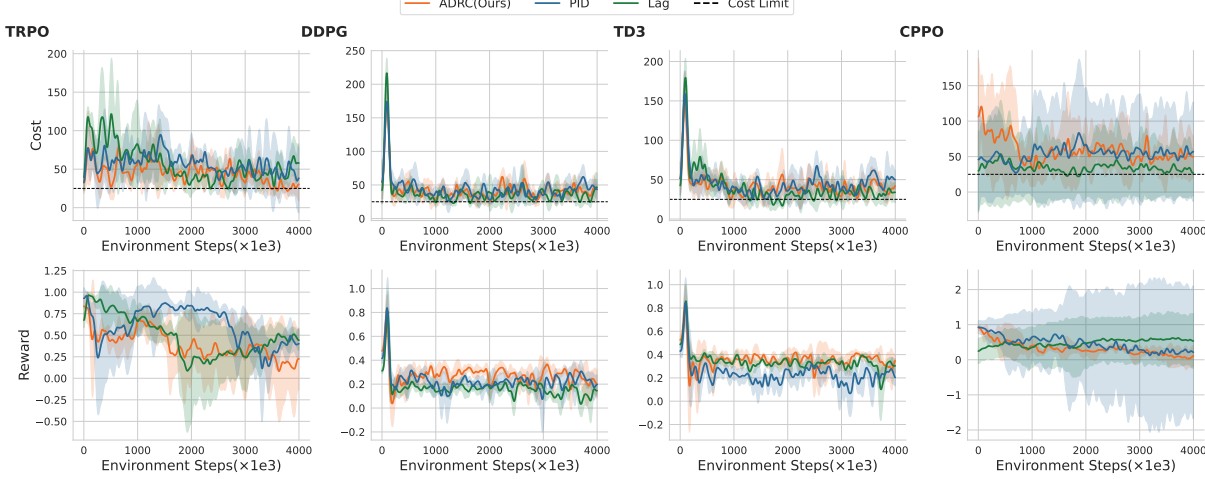

Figure 13: The training curves of RacecarButton with various Lagrangian methods across different algorithms.

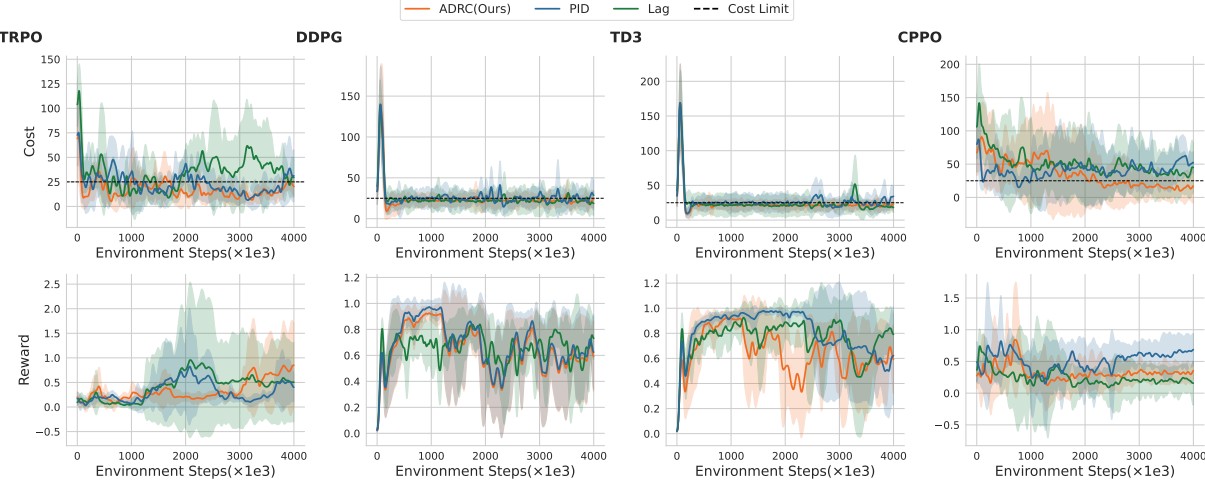

Figure 14: The training curves of RacecarCircle with various Lagrangian methods across different algorithms.

To provide a more thorough and quantitative evaluation of our method, we report the results of experiments conducted on four challenging environments, RacecarButton, RacecarCircle, RacecarGoal, and RacecarPush in Table 12 and Table 13. The metrics compared include violation rate (%), magnitude of violations, and average cost. Across these experiments, our ADRC method outperforms or matches the baseline approaches (PID and Lagrange) in 86 of 96 pairwise comparisons. The main exception is RacecarButton under the off-policy algorithms, where ADRC improves on PID across all metrics but the classical Lagrangian attains the lowest violation rates and magnitudes; the remaining exceptions are the violation rate of RacecarCircle under DDPG and two cells within 0.1 of the corresponding baseline.

Table 12: Comparison of violation rate, magnitude, and average cost on RacecarButton and RacecarCircle. Bold marks ADRC results that improve on the PID Lagrangian baseline.

| Algorithm | RacecarButton | | | RacecarCircle | | |
|---|---|---|---|---|---|---|
| | Vio. Rate (%) | Magnitude | Avg. Cost | Vio. Rate (%) | Magnitude | Avg. Cost |
| CPPOLag | $97.38 \pm 2.85$ | $65.75 \pm 16.98$ | $90.53 \pm 17.10$ | $58.32 \pm 31.47$ | $30.72 \pm 37.77$ | $50.41 \pm 41.68$ |
| CPPOPID | $97.37 \pm 4.24$ | $78.44 \pm 43.36$ | $103.21 \pm 43.64$ | $56.82 \pm 31.13$ | $21.77 \pm 25.28$ | $40.95 \pm 29.63$ |
| CPPOADRC | $\mathbf{81.18 \pm 20.97}$ | $\mathbf{35.76 \pm 29.48}$ | $\mathbf{59.11 \pm 31.27}$ | $\mathbf{39.94 \pm 38.28}$ | $21.78 \pm 31.79$ | $\mathbf{35.72 \pm 39.00}$ |
| DDPGLag | $75.88 \pm 4.41$ | $15.77 \pm 3.10$ | $39.47 \pm 3.34$ | $50.52 \pm 0.25$ | $7.87 \pm 0.74$ | $25.03 \pm 0.67$ |
| DDPGPID | $88.26 \pm 2.48$ | $20.12 \pm 2.35$ | $44.62 \pm 2.47$ | $46.17 \pm 1.40$ | $7.94 \pm 1.09$ | $27.09 \pm 0.56$ |
| DDPGADRC | $\mathbf{85.18 \pm 6.77}$ | $\mathbf{18.49 \pm 2.43}$ | $\mathbf{42.87 \pm 2.74}$ | $50.92 \pm 2.27$ | $\mathbf{5.00 \pm 0.79}$ | $\mathbf{24.51 \pm 0.33}$ |
| TD3Lag | $72.49 \pm 2.60$ | $16.31 \pm 2.92$ | $39.65 \pm 3.12$ | $49.42 \pm 0.19$ | $8.22 \pm 0.52$ | $24.95 \pm 0.52$ |
| TD3PID | $85.99 \pm 7.60$ | $21.46 \pm 4.96$ | $45.75 \pm 5.43$ | $48.72 \pm 1.24$ | $8.57 \pm 1.07$ | $27.76 \pm 1.30$ |
| TD3ADRC | $\mathbf{84.48 \pm 4.33}$ | $\mathbf{18.37 \pm 2.74}$ | $\mathbf{42.65 \pm 2.93}$ | $\mathbf{42.08 \pm 1.39}$ | $\mathbf{4.56 \pm 0.79}$ | $\mathbf{24.21 \pm 0.82}$ |
| TRPOLag | $87.31 \pm 4.65$ | $33.07 \pm 7.84$ | $57.34 \pm 7.59$ | $51.39 \pm 10.47$ | $16.10 \pm 10.65$ | $34.48 \pm 11.14$ |
| TRPOPID | $87.04 \pm 3.69$ | $32.94 \pm 3.94$ | $56.86 \pm 3.65$ | $37.99 \pm 3.47$ | $7.92 \pm 2.78$ | $23.45 \pm 1.84$ |
| TRPOADRC | $\mathbf{80.06 \pm 4.19}$ | $\mathbf{21.38 \pm 5.00}$ | $\mathbf{45.00 \pm 5.21}$ | $\mathbf{23.64 \pm 3.83}$ | $\mathbf{3.54 \pm 0.43}$ | $\mathbf{17.30 \pm 1.32}$ |

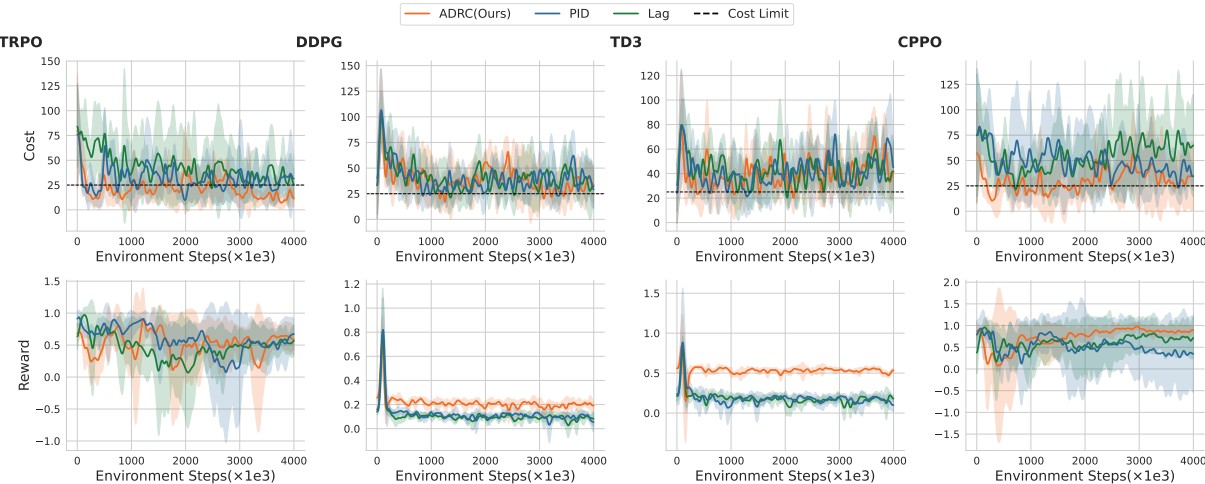

Figure 15: The training curves of RacecarGoal with various Lagrangian methods across different algorithms.

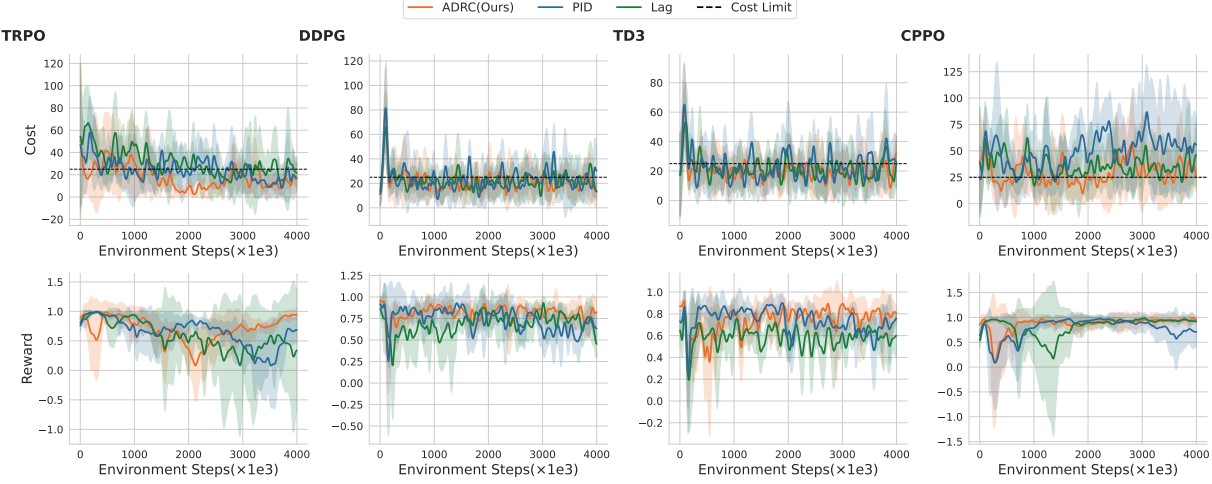

Figure 16: The training curves of RacecarPush with various Lagrangian methods across different algorithms.

Table 13: Comparison of violation rate, magnitude, and average cost on RacecarGoal and RacecarPush. Bold marks ADRC results that improve on the PID Lagrangian baseline.

| Algorithm | RacecarGoal | | | RacecarPush | | |
|---|---|---|---|---|---|---|
| | Vio. Rate (%) | Magnitude | Avg. Cost | Vio. Rate (%) | Magnitude | Avg. Cost |
| CPPOLag | 80.87 ± 19.17 | 31.18 ± 18.99 | 54.24 ± 21.01 | 57.91 ± 22.12 | 15.45 ± 12.92 | 35.54 ± 15.56 |
| CPPOPID | 72.30 ± 24.96 | 27.11 ± 15.58 | 49.02 ± 18.75 | 70.84 ± 23.97 | 28.16 ± 16.84 | 49.67 ± 19.96 |
| CPPOADRC | **47.08 ± 21.59** | **12.31 ± 9.34** | **30.12 ± 12.74** | **47.28 ± 17.05** | **12.50 ± 7.05** | **29.35 ± 11.03** |
| DDPGLag | 71.44 ± 9.47 | 18.47 ± 4.13 | 40.76 ± 5.00 | 39.58 ± 2.98 | 5.00 ± 0.49 | 22.52 ± 0.58 |
| DDPGPID | 72.05 ± 4.09 | 17.93 ± 2.81 | 40.29 ± 2.99 | 39.31 ± 1.84 | 6.89 ± 0.81 | 23.46 ± 0.98 |
| DDPGADRC | **68.41 ± 5.77** | **17.81 ± 1.34** | **39.41 ± 2.14** | **38.08 ± 3.96** | **5.10 ± 0.98** | **21.99 ± 1.13** |
| TD3Lag | 75.26 ± 2.13 | 19.93 ± 1.40 | 42.57 ± 1.59 | 36.76 ± 1.53 | 5.25 ± 0.92 | 21.64 ± 0.97 |
| TD3PID | 73.31 ± 6.06 | 19.19 ± 2.72 | 41.57 ± 3.22 | 38.87 ± 1.99 | 6.76 ± 1.65 | 23.32 ± 2.45 |
| TD3ADRC | **71.24 ± 3.00** | **17.55 ± 3.57** | **39.71 ± 3.94** | **36.21 ± 3.11** | **4.25 ± 1.50** | **20.70 ± 1.49** |
| TRPOLag | 64.31 ± 23.37 | 22.89 ± 18.69 | 43.67 ± 21.56 | 51.18 ± 11.61 | 12.82 ± 3.76 | 31.19 ± 5.63 |
| TRPOPID | 49.33 ± 15.00 | 11.70 ± 7.07 | 30.94 ± 8.81 | 40.06 ± 1.74 | 7.98 ± 1.92 | 24.46 ± 1.55 |
| TRPOADRC | **34.03 ± 8.06** | **6.16 ± 2.37** | **22.02 ± 3.61** | **26.06 ± 11.09** | **5.41 ± 2.66** | **18.28 ± 5.41** |

## F.3 Velocity Control Results

To further evaluate our method's performance in dynamic and velocity-sensitive environments, we conducted experiments on the Safety Velocity Control tasks, including SafetySwimmer and SafetyHopper. These tasks pose additional challenges by requiring agents to manage both positional constraints and velocity profiles.

The following tables present the violation rates, violation magnitudes, average costs, and average rewards achieved by different methods. Our ADRC Lagrangian method demonstrates superior safety performance in 11 of 12 safety comparisons (two environments, two algorithms, three safety metrics), the exception being the violation magnitude of TRPO-ADRC on Swimmer. Rewards are higher in one setting and lower in the others, reflecting a safety-reward trade-off that is most visible on Hopper.

Table 14: Performance comparison on SafetySwimmer environment. Bold marks the better value between PID and ADRC for each algorithm.

| Algorithm | Vio. Rate (%) | Magnitude | Avg Cost | Avg Reward |
|---|---|---|---|---|
| CPPOPID | 28.33 | 1.84 | 23.64 | **32.54** |
| CPPOADRC | **6.95** | **1.56** | **18.34** | 29.07 |
| TRPOPID | 35.43 | 1.78 | 22.48 | 27.73 |
| TRPOADRC | **11.30** | 2.44 | 20.82 | **35.66** |

Table 15: Performance comparison on SafetyHopper environment. Bold marks the better value between PID and ADRC for each algorithm.

| Algorithm | Vio. Rate (%) | Magnitude | Avg Cost | Avg Reward |
|---|---|---|---|---|
| CPPOPID | 40.33 | 5.32 | 23.92 | **1365.60** |
| CPPOADRC | **17.40** | **1.84** | **17.35** | 1155.59 |
| TRPOPID | 39.33 | 7.52 | 24.41 | **1448.46** |
| TRPOADRC | **0.93** | **0.06** | **12.02** | 1080.80 |

These results validate that the ADRC-based methods significantly improve safety metrics (lower violation rate and cost) in dynamic velocity control tasks, at the price of some reward on Hopper. This further demonstrates the effectiveness and robustness of ADRC Lagrangian formulations under more complex and realistic settings.

### F.4 Comparison with State-of-the-Art Safe RL Algorithms

To test whether the ADRC multiplier update also improves safety when combined with other safe RL algorithms, we compare against both Lagrangian-based methods (RCPO, PDO) (Tessler et al., 2018; Chow et al., 2018a) and non-Lagrangian approaches (CUP, IPO) (Yang et al., 2022; Liu et al., 2019) on velocity-control tasks.

We evaluate all methods on two challenging continuous control environments: HalfCheetah-Velocity and Hopper-Velocity from the Safety-Gymnasium benchmark. Each algorithm is trained with identical hyperparameters and evaluated using three random seeds. We report both training metrics (averaged over the entire training process) and final policy evaluation results to provide comprehensive performance assessment.

Table 16: Training performance on HalfCheetah-Velocity. Best results in **bold**, runner-up in underline.

| Algorithm | Vio. Rate (%) | Magnitude | Avg. Cost | Avg. Reward |
|---|---|---|---|---|
| CUP | 22.63±7.21 | 4.48±4.58 | 16.25±5.08 | 1532.23±255.05 |
| IPO | 29.17±1.63 | 0.81±0.02 | 19.60±0.08 | 1460.56±210.91 |
| PDO | 31.95±5.09 | 9.68±2.84 | 22.16±1.47 | 1690.62±421.72 |
| RCPO | 18.26±9.69 | 4.75±3.71 | 15.77±7.03 | 1497.99±410.94 |
| RCPO-ADRC | **0.00±0.00** | **0.00±0.00** | **8.40±2.40** | 1329.62±293.58 |
| TRPO-ADRC | 1.19±0.25 | 0.09±0.10 | 10.89±0.34 | **1743.09±295.33** |
| CPPO-ADRC | 8.53±12.06 | 0.55±0.78 | 15.36±2.05 | 1504.17±198.53 |

Table 17: Training performance on Hopper-Velocity. Best results in **bold**, runner-up in underline.

| Algorithm | Vio. Rate (%) | Magnitude | Avg. Cost | Avg. Reward |
|---|---|---|---|---|
| CUP | 37.11±3.59 | 5.47±0.59 | 21.73±1.03 | 1085.14±204.86 |
| IPO | 51.99±8.89 | 1.68±0.18 | 24.70±0.84 | 1082.95±103.43 |
| PDO | 27.56±9.25 | 8.42±3.48 | 20.00±7.01 | 1098.22±178.78 |
| RCPO | 37.57±7.00 | 5.45±0.93 | 23.24±2.72 | **1247.72±292.68** |
| RCPO-ADRC | **2.57±2.19** | **0.06±0.05** | **14.23±2.74** | 1186.87±70.94 |
| TRPO-ADRC | 7.76±9.59 | 0.33±0.40 | 15.13±3.04 | 1167.62±90.74 |
| CPPO-ADRC | 11.01±5.65 | 0.92±0.67 | 15.43±1.40 | 1083.94±68.49 |

Table 18: Evaluation performance on HalfCheetah-Velocity. Best results in **bold**, runner-up in underline.

| Algorithm | Reward | Cost | Length |
|---|---|---|---|
| CUP | 2175.61±491.09 | 28.57±21.15 | 1000.00±0.00 |
| IPO | 1819.75±292.45 | 16.10±12.99 | 1000.00±0.00 |
| PDO | **2468.78±581.21** | **5.77±7.32** | 1000.00±0.00 |
| RCPO | 2296.82±665.64 | 15.00±8.81 | 1000.00±0.00 |
| RCPO-ADRC | 1642.22±211.39 | 10.23±3.51 | 1000.00±0.00 |
| TRPO-ADRC | 2394.09±419.84 | 14.63±15.65 | 1000.00±0.00 |
| CPPO-ADRC | 2098.71±464.92 | 13.87±11.43 | 1000.00±0.00 |

Tables 16 and 17 show that our ADRC variants markedly enhance *training-time stability*: compared with existing safe RL methods, ADRC achieves consistently lower violation rates, smaller violation magnitudes, and reduced average costs. For example, on HalfCheetah, RCPO-ADRC eliminates violations entirely (0.00±0.00% vs. 18.26±9.69% for RCPO) and attains the lowest training cost (8.40±2.40); on Hopper, RCPO-ADRC sharply suppresses violations (2.57±2.19%) with the smallest magnitudes (0.06±0.05) and cost (14.23±2.74). Crucially, this improved safety does *not* come at the expense of learning quality: ADRC maintains competitive training rewards and can even be better; e.g., TRPO-ADRC attains the highest

Table 19: Evaluation performance on Hopper-Velocity. Best results in **bold**, runner-up in underline.

| Algorithm | Reward | Cost | Length |
|---|---|---|---|
| CUP | 1326.84±386.22 | 26.90±10.59 | 854.40±165.37 |
| IPO | 1216.01±129.12 | 27.57±4.95 | 797.33±118.41 |
| PDO | 1177.19±135.02 | 18.03±25.50 | 797.57±167.85 |
| RCPO | **1554.56±223.49** | 37.53±24.35 | 979.30±29.27 |
| RCPO-ADRC | 1248.94±220.47 | **9.27±5.18** | 818.10±128.67 |
| TRPO-ADRC | 1470.61±152.19 | 10.67±3.76 | **1000.00±0.00** |
| CPPO-ADRC | 1322.83±157.06 | 10.43±7.12 | 910.20±127.00 |

training reward on HalfCheetah (1743.09±295.33) with only 1.19±0.25% violations, indicating stable and efficient optimization.

Tables 18 and 19 further examine *convergence-time* performance (evaluation). Even without explicitly measuring constraints at evaluation, ADRC remains competitive or even superior on task metrics: on HalfCheetah, TRPO-ADRC reaches runner-up reward (2394.09±419.84), close to the best; on Hopper, RCPO-ADRC achieves the lowest evaluation cost (9.27±5.18) and TRPO-ADRC sustains the maximum horizon (1000.00±0.00) with strong reward (1470.61±152.19). Together, these results confirm that ADRC improves training stability and safety while preserving (and in cases improving) final task performance and convergence behavior, offering a plug-and-play safety enhancement over existing safe RL baselines.

### F.5 Parameter Sensitivity Analysis

All runs in this section use a single random seed.

#### F.5.1 Tuning parameter $k_{ap}$

To assess the effect of the control gain $k_{ap}$ on the performance of ADRC-based Lagrangian methods, we conducted a series of ablation experiments. Specifically, we evaluated three distinct values of $k_{ap}$ $(0.01, 0.1, 1)$ and compared them with existing approaches, including PID-based and classical Lagrangian methods. These experiments were carried out in two challenging environments, *CarPush* and *RacecarGoal*, using two reinforcement learning algorithms, *CPPO* and *TRPO*. The results highlight ADRC's ability to dynamically adjust the control gain, demonstrating superior adaptability and improved performance with carefully selected parameter settings.

Table 20: The proportion of constraint violations during training (Vio. Rate), the average magnitude of violations (Magnitude), and the average cost (Avg. Cost) for TRPO and CPPO algorithms across CarPush and RacecarGoal environments with various $k_{ap}$ values, PID, and Lag methods. Bold values indicate better performance compared to PID.

| Algorithm | Method | CarPush | | | RacecarGoal | | |
|---|---|---|---|---|---|---|---|
| | | Vio. Rate (%) | Magnitude | Avg. Cost | Vio. Rate (%) | Magnitude | Avg. Cost |
| TRPO | $k_{ap} = 1$ | **36.38** | **4.51** | 21.99 | **32.73** | **5.78** | **22.27** |
| | $k_{ap} = 0.1$ | **30.20** | **3.86** | **20.36** | **29.05** | **3.44** | **18.95** |
| | $k_{ap} = 0.01$ | **34.98** | 7.68 | 24.27 | **23.83** | **4.29** | **19.36** |
| | PID | 38.40 | 4.84 | 21.96 | 44.60 | 7.04 | 26.15 |
| | Lag | 39.88 | 5.38 | 23.31 | 87.33 | 37.36 | 61.53 |
| CPPO | $k_{ap} = 1$ | 86.08 | 24.38 | 48.31 | **69.98** | **18.62** | **39.87** |
| | $k_{ap} = 0.1$ | **16.25** | **4.05** | **13.46** | 33.08 | **5.78** | 21.22 |
| | $k_{ap} = 0.01$ | **42.83** | **5.75** | **23.56** | 20.43 | **2.50** | **15.42** |
| | PID | 67.28 | 12.80 | 34.67 | 79.25 | 23.88 | 46.44 |
| | Lag | 46.43 | 6.67 | 25.90 | 84.35 | 30.16 | 53.38 |

As shown in Table 20, we report the **Violation Rate (Vio. Rate)**, the **Magnitude** of constraint violations, and the **Average Cost (Avg. Cost)** for both the *CarPush* and *RacecarGoal* environments using the *TRPO*

and *CPPO* algorithms. The results demonstrate the superior performance of our ADRC approach with varying $k_{ap}$ values compared to baseline methods (PID and Lag). Specifically:

- For the **CarPush** environment:
    - Under **TRPO**, the configuration $k_{ap} = 0.1$ achieves the lowest violation rate (30.20%) and magnitude (3.86), alongside a reduced average cost (20.36), outperforming both PID and Lag.
    - For **CPPO**, $k_{ap} = 0.1$ shows remarkable results, with a violation rate of 16.25%, the smallest magnitude (4.05), and the lowest average cost (13.46). This highlights the adaptability of ADRC at this gain level.

- For the **RacecarGoal** environment:
    - With **TRPO**, $k_{ap} = 0.01$ demonstrates the best performance, achieving a violation rate of 23.83%, a moderate magnitude (4.29), and a reduced average cost (19.36). This represents a clear improvement over both PID and Lag methods.
    - Similarly, under **CPPO**, $k_{ap} = 0.01$ achieves the best performance with a violation rate of 20.43%, the smallest magnitude (2.50), and the lowest average cost (15.42). These results further emphasize ADRC's effectiveness.

- For both environments, the baseline PID and Lag methods generally exhibit higher violation rates, magnitudes, and costs. Lag in particular performs poorly, especially in the *RacecarGoal* environment, where it yields the highest violation rates and costs.

These results show that our ADRC method outperforms traditional methods for moderate gains, particularly $k_{ap} = 0.1$, while extreme settings can underperform PID on individual metrics (for instance $k_{ap} = 1$ under CPPO on CarPush), underlining the value of the adaptive gain selection in Sec. 4.4.

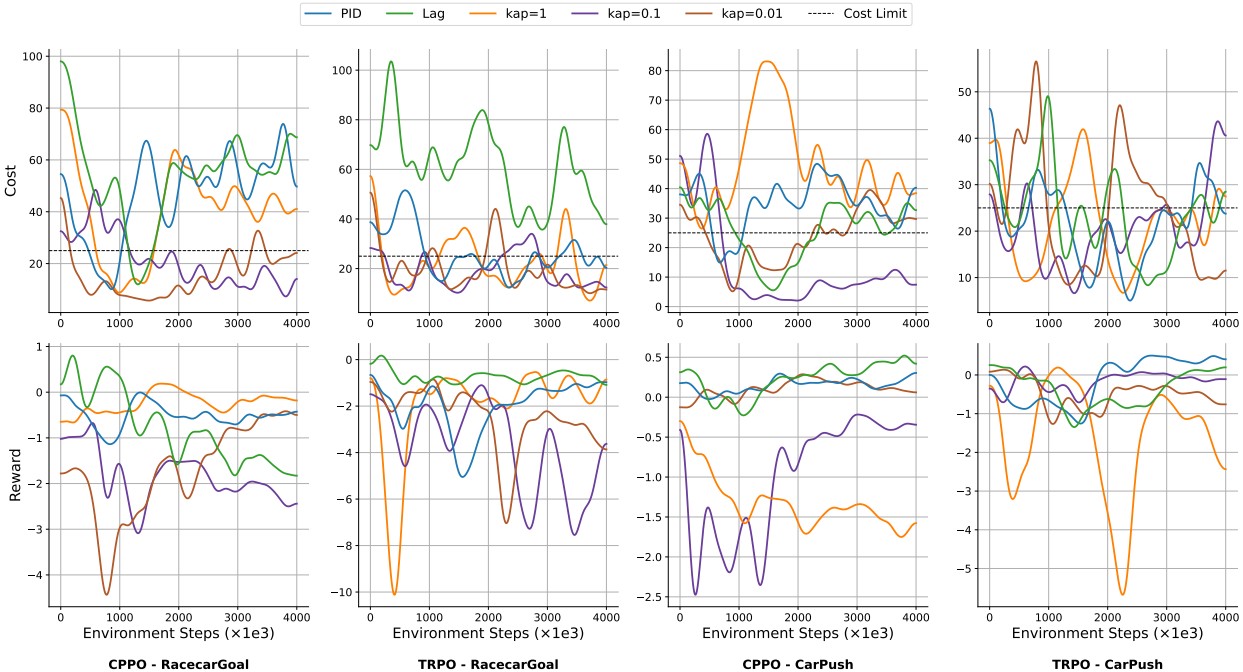

Figure 17: The training curve for TRPO and CPPO algorithms across CarPush and RacecarGoal environments with various $k_{ap}$ values, PID, and Lag methods.

Figure 17 provides the training curves for reward and cost across the evaluated $k_{ap}$ values, PID, and Lag methods. These curves illustrate the consistent performance improvements of our method throughout the

training process. Our approach not only converges more effectively but also demonstrates a more favorable trade-off between reward maximization and cost minimization.

### F.5.2 Tuning parameter $k_{ad}$

To assess the effect of the control gain $k_{ad}$ on the performance of ADRC-based Lagrangian methods, we conducted a series of ablation experiments. Specifically, we evaluated three distinct values of $k_{ad}$ $(0.01, 0.1, 1)$ and compared them with existing approaches, including PID-based and classical Lagrangian methods. These experiments were carried out in two challenging environments, *CarPush* and *RacecarGoal*, using two reinforcement learning algorithms, *CPPO* and *TRPO*. The results highlight ADRC's ability to dynamically adjust the control gain, demonstrating superior adaptability and improved performance with carefully selected parameter settings. To evaluate the impact of the tuning parameter $k_{ad}$ on ADRC Lagrangian methods' performance, we conducted ablation experiments by selecting three different values of $k_{ad} = 0.01, 0.1, 1$ and comparing them against existing methods, including PID Lagrangian methods and classical Lagrangian methods. The experiments were performed across two environments which are CarPush and RacecarGoal and adopt two algorithms which are CPPO and TRPO.

Table 21: The proportion of constraint violations during training (Vio. Rate), the average magnitude of violations (Magnitude), and the average cost (Avg. Cost) for TRPO and CPPO algorithms across CarPush and RacecarGoal environments with various $k_{ad}$ values, PID, and Lag methods. Bold values indicate better performance compared to PID.

| Algorithm | Method | CarPush | | | RacecarGoal | | |
|---|---|---|---|---|---|---|---|
| | | Vio. Rate (%) | Magnitude | Avg. Cost | Vio. Rate (%) | Magnitude | Avg. Cost |
| TRPO | $k_{ad} = 1$ | **38.23** | 6.16 | 23.11 | **28.55** | **6.17** | **20.23** |
| | $k_{ad} = 0.1$ | **36.68** | 6.29 | 21.99 | **38.25** | 9.12 | **23.92** |
| | $k_{ad} = 0.01$ | **30.20** | **3.86** | **20.36** | **39.08** | **5.75** | **23.33** |
| | PID | 38.40 | 4.84 | 21.96 | 44.60 | 7.04 | 26.15 |
| | Lag | 39.88 | 5.38 | 23.31 | 87.33 | 37.36 | 61.53 |
| CPPO | $k_{ad} = 1$ | **16.20** | **1.78** | **16.60** | **48.68** | **9.12** | **27.80** |
| | $k_{ad} = 0.1$ | **15.08** | **5.06** | **15.45** | **48.55** | **8.59** | **27.80** |
| | $k_{ad} = 0.01$ | **16.25** | **4.05** | **13.46** | **33.08** | **5.78** | **21.22** |
| | PID | 67.28 | 12.80 | 34.67 | 79.25 | 23.88 | 46.44 |
| | Lag | 46.43 | 6.67 | 25.90 | 84.35 | 30.16 | 53.38 |

As shown in Table 21, we report the **Violation Rate (Vio. Rate)**, the **Magnitude** of constraint violations, and the **Average Cost (Avg. Cost)** for both the *CarPush* and *RacecarGoal* environments using the *TRPO* and *CPPO* algorithms. The results highlight the superior performance of our ADRC approach with varying $k_{ad}$ values compared to the baseline methods (PID and Lag). Specifically:

- For the **CarPush** environment:

  - Under **TRPO**, $k_{ad} = 0.01$ achieves the lowest violation rate (30.20%) and the smallest magnitude (3.86), alongside a reduced average cost (20.36). This indicates better constraint satisfaction and efficiency compared to PID and Lag.
  - For **CPPO**, $k_{ad} = 0.1$ yields the best performance with the lowest violation rate (15.08%), a moderate magnitude (5.06), and the smallest average cost (15.45). These results highlight ADRC's adaptability at this parameter setting.

- For the **RacecarGoal** environment:

  - With **TRPO**, $k_{ad} = 1$ shows excellent performance, achieving the lowest violation rate (28.55%) and average cost (20.23), alongside a relatively small magnitude (6.17).
  - Under **CPPO**, $k_{ad} = 0.01$ demonstrates the best results, with a low violation rate (33.08%), a reduced magnitude (5.78), and the smallest average cost (21.22). This showcases ADRC's ability to manage constraints effectively in this challenging environment.

- Across both environments, the baseline methods (PID and Lag) exhibit higher violation rates throughout, and larger magnitudes and costs in most cases; PID retains an edge on individual magnitude and cost cells (for instance $k_{ad} = 0.1$ on RacecarGoal). Lag performs particularly poorly, with significantly higher metrics, especially in the *RacecarGoal* environment, where it records the highest violation rate (87.33%) and average cost (61.53).

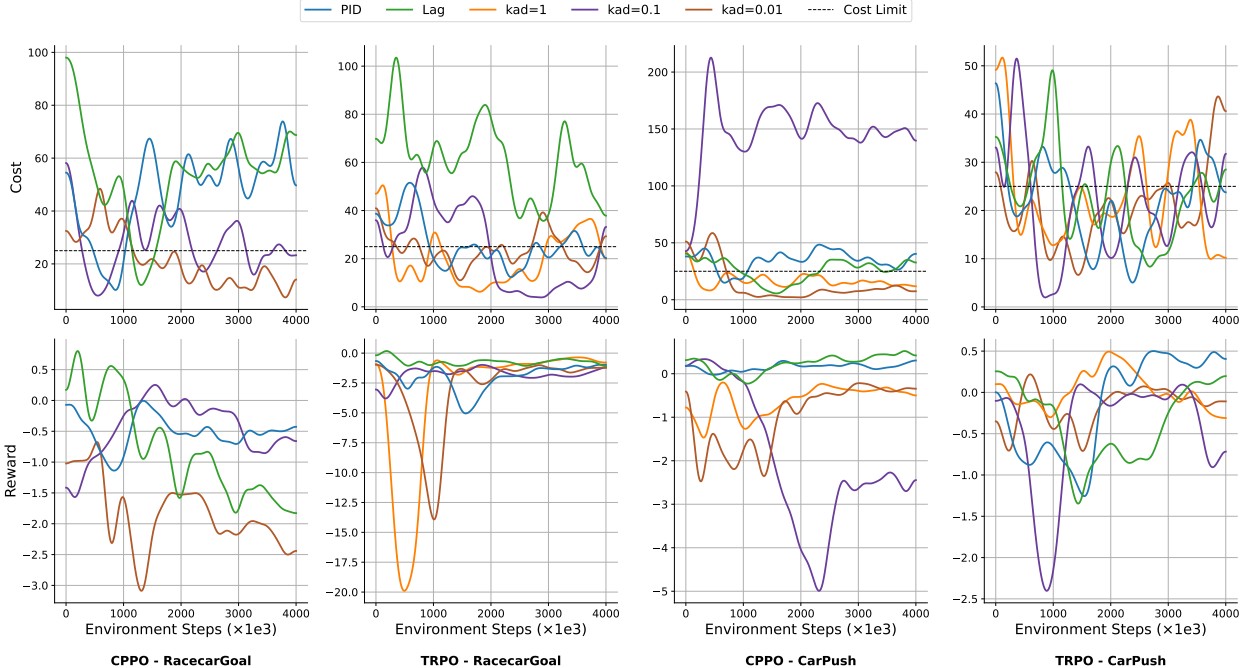

Figure 18: The training curve for TRPO and CPPO algorithms across CarPush and RacecarGoal environments with various $k_{ad}$ values, PID, and Lag methods.

### F.5.3 Tuning parameter $c_r$

To evaluate the impact of the tuning parameter $c_r$ on ADRC Lagrangian methods' performance, we conducted ablation experiments by selecting five different values of $c_r = 0.05, 0.1, 0.15, 0.2, 0.25$ and comparing them against existing methods, including PID Lagrangian methods and classical Lagrangian methods. The experiments were performed across two environments which are CarPush and RacecarGoal and adopt two algorithms which are CPPO and TRPO.

As shown in Table 22, we report the **Violation Rate (Vio. Rate)**, the **Magnitude** of constraint violations, and the **Average Cost (Avg. Cost)**. The results demonstrate that our method consistently outperforms the baseline methods (PID Lagrangian methods and classical Lagrangian methods) across a majority of the $c_r$ values. Specifically:

- For the CarPush environment, our method achieves lower violation rates and magnitudes in most cases, while maintaining competitive average costs.

- Similarly, in the RacecarGoal environment, our approach demonstrates significant improvements, particularly with $c_r = 0.1$ and $c_r = 0.2$, where it achieves the lowest violation rates and magnitudes.

Figure 19 provides the training curves for reward and cost across the evaluated $c_r$ values, PID, and Lag methods. These curves illustrate the consistent performance improvements of our method throughout the training process. Our approach not only converges more effectively but also demonstrates a more favorable trade-off between reward maximization and cost minimization.

Table 22: The proportion of constraint violations during training (Vio. Rate), the average magnitude of violations (Magnitude), and the average cost (Avg. Cost) for TRPO and CPPO algorithms across CarPush and RacecarGoal environments with various $c_r$ values, PID, and Lag methods. Bold values indicate the better performance compared to PID.

| Algorithm | Method | CarPush | | | RacecarGoal | | |
|---|---|---|---|---|---|---|---|
| | | Vio. Rate (%) | Magnitude | Avg. Cost | Vio. Rate (%) | Magnitude | Avg. Cost |
| | Lag | 43.83 | 7.99 | 26.19 | 87.33 | 37.36 | 61.53 |
| | PID | 38.40 | 4.84 | 21.96 | 44.60 | 7.04 | 26.15 |
| | $c_r = 0.05$ | 46.25 | 6.35 | 24.79 | **33.98** | **5.25** | **20.83** |
| TRPO | $c_r = 0.1$ | **30.20** | **3.86** | **20.36** | **29.05** | **3.44** | **18.95** |
| | $c_r = 0.15$ | **34.83** | 10.92 | 27.00 | **31.25** | **5.34** | **21.16** |
| | $c_r = 0.2$ | **28.60** | 7.32 | **20.72** | **40.65** | 6.10 | 23.67 |
| | $c_r = 0.25$ | **34.13** | 6.26 | 22.46 | **38.50** | **6.71** | 23.40 |
| | Lag | 46.43 | 6.67 | 25.90 | 84.35 | 30.16 | 53.38 |
| | PID | 67.28 | 12.80 | 34.67 | 79.25 | 23.88 | 46.44 |
| | $c_r = 0.05$ | **44.73** | **5.95** | **24.80** | **34.38** | **3.90** | **22.45** |
| CPPO | $c_r = 0.1$ | **16.25** | **4.05** | **13.46** | **33.08** | **5.78** | **21.22** |
| | $c_r = 0.15$ | **56.70** | **10.22** | **30.40** | **52.88** | **10.69** | **31.37** |
| | $c_r = 0.2$ | **12.30** | **2.31** | **13.45** | **48.95** | **8.77** | **26.91** |
| | $c_r = 0.25$ | **34.00** | **6.12** | **22.09** | **62.83** | **13.37** | **33.99** |

Overall, the results highlight the robustness of our method, as it achieves superior performance in the majority of scenarios. This indicates that the choice of $c_r$ significantly influences the balance between reward and cost, and our approach consistently outperforms existing methods under comparable conditions.

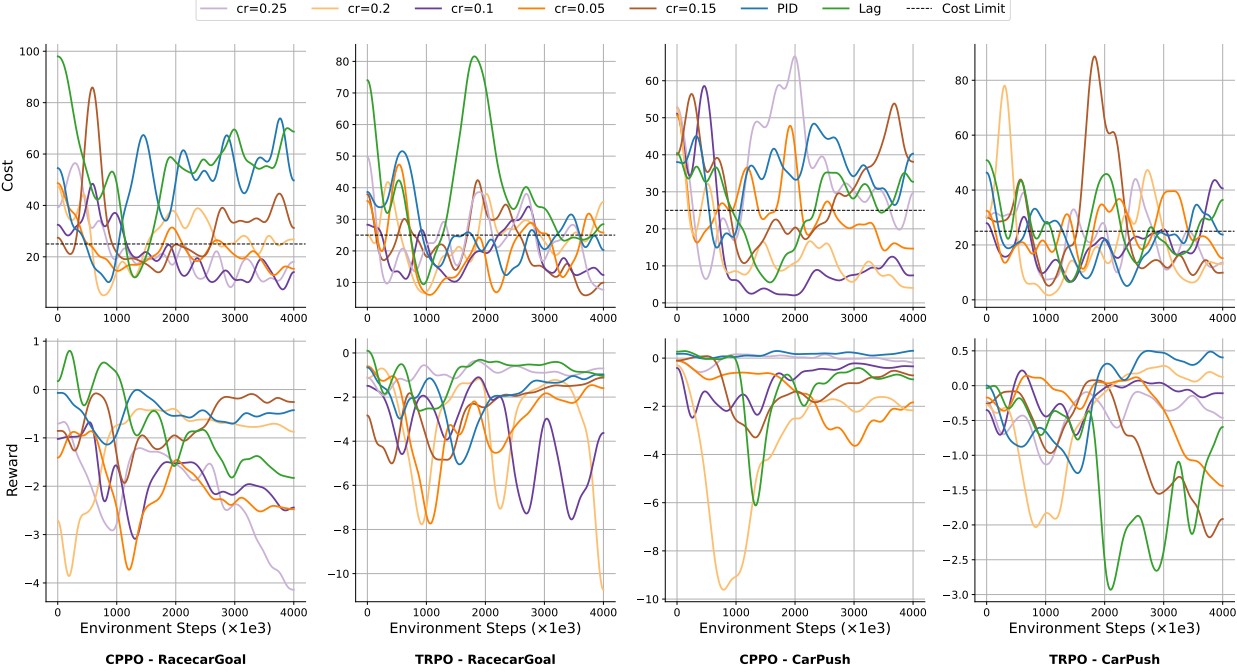

Figure 19: The training curve for TRPO and CPPO algorithms across CarPush and RacecarGoal environments with various $c_r$ values, PID, and Lag methods.

## F.6 Noise Sensitivity Analysis

Tables 23–24 report the sensitivity of our TRPO-based ADRC approach to injected noise in the SWIMMER-Velocity environment. The training row at $\sigma=0$ coincides with the run reported in Table 14, whereas the evaluation study uses an independent rerun of the noise-free configuration, so its $\sigma=0$ row differs slightly

Table 23: Training noise sensitivity on Swimmer-Velocity.

| Method / $\sigma$ | Vio. Rate (%) | Magnitude | Avg. Cost | Avg. Reward |
|---|---|---|---|---|
| PID | 35.43 | 1.78 | 22.48 | 27.73 |
| $\sigma = 0$ | 11.30 | 2.44 | 20.82 | 35.66 |
| $\sigma = 2$ | 10.93 | 2.55 | 21.93 | 22.41 |
| $\sigma = 5$ | **5.93** | 1.62 | **8.04** | 16.18 |
| $\sigma = 10$ | 23.95 | 3.55 | 19.93 | 12.41 |

Table 24: Evaluation noise sensitivity on Swimmer-Velocity.

| Method / $\sigma$ | Avg. Cost | Vio. Rate (%) | Magnitude | Avg. Reward |
|---|---|---|---|---|
| PID | 23.05 | 39.79 | 6.23 | 32.19 |
| $\sigma = 0$ | 18.40 | 12.07 | 11.35 | 39.55 |
| $\sigma = 2$ | 18.00 | 27.60 | 14.52 | 19.72 |
| $\sigma = 5$ | **4.16** | 4.57 | 9.18 | 17.23 |
| $\sigma = 10$ | 18.20 | 28.72 | 15.06 | 20.51 |

from the final-performance numbers in Table 5. The results demonstrate that TRPO-ADRC is robust to disturbances: compared with PID-type controllers, TRPO-ADRC achieves consistently lower violation rates and reduced average costs while maintaining stable reward learning, although its violation magnitudes exceed PID's at most noise levels. For example, during training, TRPO-ADRC reduces the violation rate from 35.43% (PID) to 11.30% under $\sigma=0$, and further to only 5.93% under $\sigma=5$, with the average cost dropping from 22.48 to 8.04.

Importantly, these safety and stability benefits do not compromise convergence performance. At evaluation, TRPO-ADRC achieves competitive or even higher rewards while retaining robustness. Under $\sigma=0$, TRPO-ADRC attains higher reward than TRPO-PID (39.55 vs. 32.19) while simultaneously lowering both cost and violation rate. Even when the disturbance level increases ($\sigma=5, 10$), TRPO-ADRC sustains reasonable rewards with substantially reduced safety violations compared to PID. These findings confirm that TRPO-ADRC improves training stability and safety without hindering convergence, demonstrating strong robustness to noise in the SWIMMER-Velocity environment.

### F.7 Ablation Study

To evaluate the contribution of key components in the ADRC-Lagrangian framework, we conducted an ablation study by systematically modifying specific features of the proposed method. Specifically, we examined the impact of replacing the transient process $r(t)$ with a static reference signal and fixing the dynamically adjusted compensation gain $\omega_o$ to a constant value. These modifications simplify the framework to a configuration resembling a PID-based Lagrangian method, enabling a fair comparison of their relative contributions.

In the first ablation, the transient process $r(t)$ is replaced with a fixed reference signal $r(t) = d$, as used in traditional PID Lagrangian methods. While $r(t)$ is designed in the full ADRC framework to provide a smooth transition toward the cost threshold denoted as $d$, setting $r(t) = d$ eliminates this smoothing effect. This simplification forces the system to directly track the constant reference signal, potentially causing abrupt updates to the policy parameters and destabilizing training. The updating law is transformed into:

$$\lambda_t = (k_{ap} + \omega_o k_{ad})(x_1 - d) + (k_{ad} + \omega_o) x_2 + \omega_o k_{ap} \int_0^t (x_1(\tau) - d) \, d\tau. \tag{74}$$

For the second ablation, we disabled the adaptive computation of $\omega_o$ by fixing it to a constant value throughout training. The ADRC gains $k_{ap}$ and $k_{ad}$ are kept at their defaults, and the update law retains the same form as Eqn. 17 but with a frozen observer bandwidth. This isolates the contribution of the adaptive gain mechanism: any performance gap relative to full ADRC is attributable to the loss of online bandwidth

adjustment. With constant $\omega_o$, the update law reduces to a static PID-like controller whose equivalent gains follow from Proposition 4.1:

$$\lambda_t = k_p(x_1 - r) + k_d(x_2 - \dot{r}) + k_i \int_0^t (x_1(\tau) - r(\tau))d\tau - \ddot{r}. \tag{75}$$

The ablation experiments evaluate the importance of the dynamic transient process $r(t)$ and the adaptive compensation gain $\omega_o$ in the ADRC-Lagrangian framework. Specifically, we tested two simplified configurations: "delete_r(t)," where the transient process $r(t)$ is replaced with a static reference signal $r(t) = d$, and "delete_$\omega_o$," where the adaptive computation of $\omega_o$ is disabled by fixing it to a constant value. These modifications isolate the contributions of each component. The experiments were conducted in the CarButton and RacecarGoal environments using TRPO and CPPO as base algorithms, with the results summarized in Table 25.

Table 25: The proportion of constraint violations during training (Vio. Rate), the average magnitude of violations (Magnitude), and the average cost (Avg. Cost) for TRPO and CPPO algorithms across CarButton and RacecarGoal environments with various methods. Bold values indicate better performance compared to PID.

| Algorithm | Method | CarButton | | | RacecarGoal | | |
|---|---|---|---|---|---|---|---|
| | | Vio. Rate (%) | Magnitude | Avg. Cost | Vio. Rate (%) | Magnitude | Avg. Cost |
| TRPO | Lag | 55.90 | 17.65 | 39.28 | 87.33 | 37.36 | 61.53 |
| | PID | 85.15 | 18.12 | 42.13 | 44.60 | 7.04 | 26.15 |
| | delete_$\omega_o$ | **71.85** | **9.42** | **32.56** | **40.53** | **7.29** | **26.14** |
| | delete_r(t) | **74.95** | **16.36** | **40.23** | **31.65** | **6.15** | **22.26** |
| | ADRC | **26.35** | **6.14** | **23.42** | **29.05** | **3.44** | **18.95** |
| CPPO | Lag | 99.88 | 98.34 | 123.31 | 84.35 | 30.16 | 53.38 |
| | PID | 99.88 | 57.95 | 82.92 | 79.25 | 23.88 | 46.44 |
| | delete_$\omega_o$ | 99.90 | **51.61** | **76.58** | **65.40** | **13.99** | **36.38** |
| | delete_r(t) | **92.53** | **33.78** | **58.20** | **54.08** | **15.23** | **34.66** |
| | ADRC | **93.68** | **33.97** | **58.74** | **33.08** | **5.78** | **21.22** |

Replacing $r(t)$ with a static reference signal ("delete_r(t)") resulted in higher violation rates and magnitudes compared to the full ADRC framework but still outperformed the PID baseline. For instance, in the Racecar-Goal environment with CPPO, the violation rate decreased from 79.25% (PID) to 54.08% ("delete_r(t)"), but ADRC achieved a further reduction to 33.08%. Similarly, fixing $\omega_o$ ("delete_$\omega_o$") impaired the system's adaptability to environmental changes, leading to increased average costs. In the same environment, the average cost dropped from 46.44 (PID) to 36.38 ("delete_$\omega_o$") but was substantially lower with ADRC at 21.22. These results demonstrate that while both components are essential for optimal performance, even the simplified versions outperform PID, highlighting the robustness of the ADRC-Lagrangian framework.

Figure 20 provides the training curves for reward and cost across the ablation study. These curves illustrate the consistent performance improvements of our method throughout the training process. Our approach not only converges more effectively but also demonstrates a more favorable trade-off between reward maximization and cost minimization.

## F.8 Case Study

To gain a deeper understanding of how our ADRC Lagrangian methods outperform the baseline, we conduct a case study adopting the TRPO algorithm in the CarCircle-1 environment. In this environment, agents are tasked with navigating around a fixed-radius circle. The agents' goal is to maintain a smooth circular trajectory while staying within the designated circular boundary and avoiding collisions with obstacles.

The reward structure in the CarCircle-1 environment is designed to encourage agents to follow the circle boundary as closely as possible while maintaining a smooth motion. High rewards are achieved when the agent's trajectory aligns with the circle's radius, and its velocity vector aligns tangentially to the circular

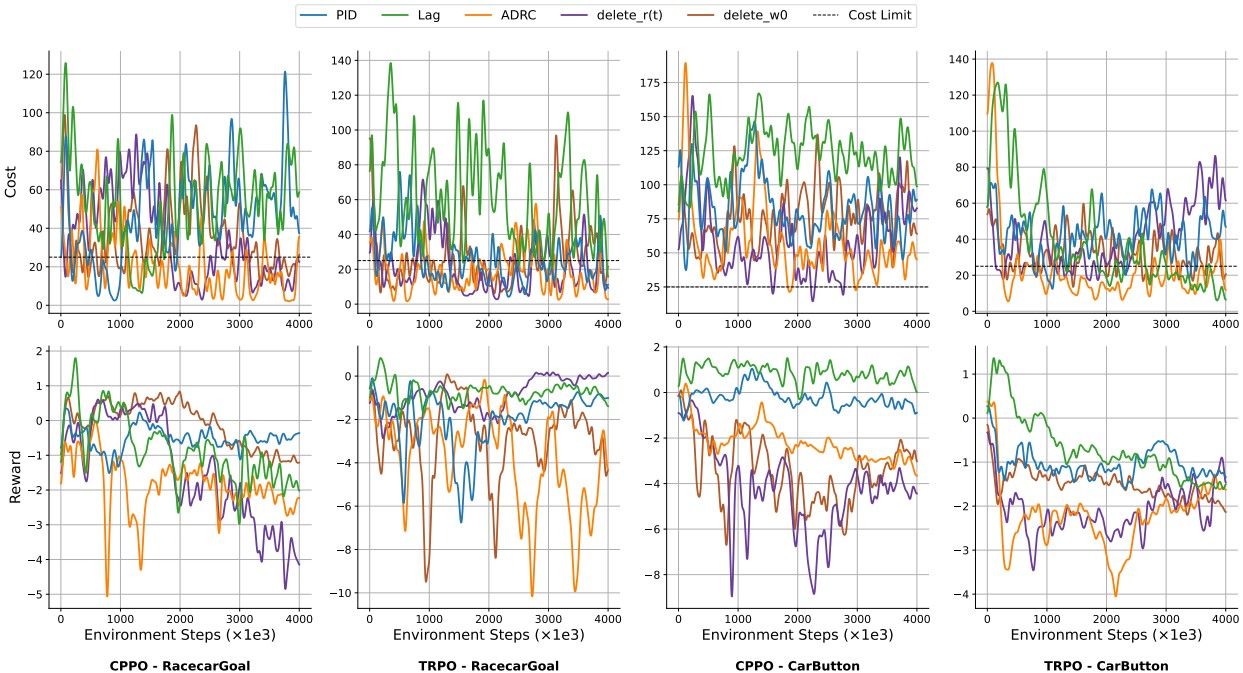

Figure 20: The training curve for TRPO and CPPO algorithms across CarButton and RacecarGoal environments with ablation study.

path. The reward for the agent is calculated using the following formula:

$$\text{Reward} = \frac{\frac{-u \cdot y + v \cdot x}{r}}{1 + |r - R|} \cdot \text{reward\_factor}, \tag{76}$$

where $x, y$ represent the agent's position, $u, v$ represent the agent's velocity, $r$ is the distance from the agent to the circle center ($r = \sqrt{x^2 + y^2}$), $R$ is the fixed radius of the circle, and reward\_factor is a scaling constant. This formula incentivizes agents to maintain a smooth and stable trajectory around the circle.

To increase the complexity of the environment, CarCircle-1 introduces two vertical walls positioned symmetrically near the circle's boundary. These walls present an additional challenge, as agents must avoid crossing into the wall regions while navigating the circle. Costs are incurred when the agent violates safety constraints, such as exceeding the circle's radius or crossing the boundaries defined by the walls. The cost is computed using the following conditions:

$$\text{Cost} = \begin{cases} 1 & \text{if } |x| > \text{wall\_threshold or } \sqrt{x^2 + y^2} > R, \\ 0 & \text{otherwise}, \end{cases} \tag{77}$$

where wall\_threshold is the horizontal boundary defined by the walls, and $R$ is the circle's radius.

In this study, we adopt the default settings in OmniSafe (Ji et al., 2024), with $R = 1.5$, wall\_threshold $= 1.125$, and reward\_factor $= 0.1$. The study is conducted using the TRPO algorithm, with hyperparameters detailed in Appendix D.1. The models are trained over 4,000 episodes, with each episode consisting of 1,000 steps. After training, the final checkpoint is used for evaluation. During evaluation, we simulate a single episode of 5,000 steps, where rewards are calculated using Eqn. 76, and costs are determined by Eqn. 77. We collected the position and the corresponding reward and cost of the agents and the results of this case study are presented in Figure 21.

In Figure 21, the color of the points represents the reward, with deeper colors indicating higher rewards. The first row visualizes the reward density at each position during the episode, while the second row illustrates the

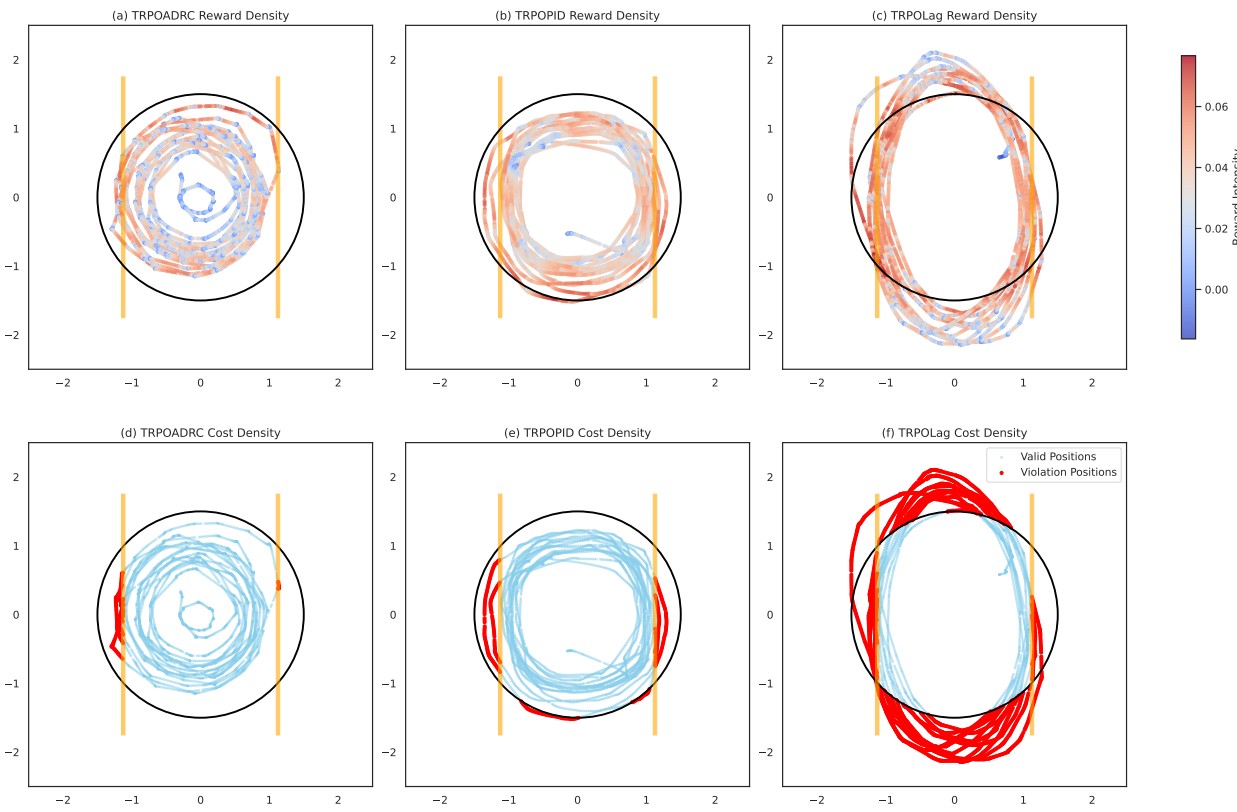

Figure 21: Reward density and cost analysis for TRPO algorithm adapting three Lagrangian methods under CarCircle environment.

cost associated with each position. Blue points indicate safe positions, whereas red points represent unsafe ones. The results show that the classical Lagrangian method achieves the highest reward, but the trajectory deviates from a perfect circle, forming an ellipse instead. Agents trained with this method learn to avoid the walls but fail to recognize the importance of staying within the circle. To maximize rewards and bypass the walls, these agents move outside the circle, resulting in a total of 496 safety violations. In contrast, agents trained with the PID Lagrangian method demonstrate better safety awareness, recognizing that moving outside the circle is unsafe. However, to achieve higher rewards, they still cross the walls frequently, leading to 320 safety violations. Finally, agents trained with our ADRC method maintain a strict adherence to staying within the circle and exhibit only 132 safety violations, establishing a superior safety performance.

The improved results achieved by the ADRC method can be attributed to its ability to reduce phase lag and minimize oscillations, thereby enhancing the stability of training. These properties enable ADRC to maintain better control over the agent's behavior, ensuring stricter adherence to safety constraints while still optimizing for rewards. The stability and precision offered by ADRC during training allow the agent to effectively balance the trade-off between maximizing rewards and minimizing safety violations, demonstrating the advantages of our proposed method in challenging environments.

### F.9 Final Policy Performance

To assess the final performance of the trained policies rather than intermediate training behavior, we conducted experiments on the Swimmer and Hopper environments from the Velocity tasks suite. The results compare ADRC-based and PID-based methods under CPPO and TRPO frameworks.

Table 26: Performance on Swimmer environment.

| Algorithm | Avg Reward | Avg Cost | Vio. Rate (%) |
|---|---|---|---|
| CPPOPID | 30.10 | 22.44 | 28.02 |
| CPPOADRC | 29.39 | 16.77 | 14.16 |
| TRPOPID | 28.72 | 21.34 | 37.85 |
| TRPOADRC | 36.32 | 19.03 | 12.16 |

Table 27: Performance on Hopper environment.

| Algorithm | Avg Reward | Avg Cost | Vio. Rate (%) |
|---|---|---|---|
| CPPOPID | 1466.47 | 48.20 | 30.00 |
| CPPOADRC | 1520.18 | 8.20 | 24.63 |
| TRPOPID | 1038.47 | 18.70 | 29.76 |
| TRPOADRC | 1384.11 | 12.90 | 10.98 |

These results demonstrate that the ADRC-based method achieves lower constraint violation rates and costs, while maintaining or improving the overall reward compared to PID-based baselines.

### F.10 Sensitivity Analysis of PID Lagrangian Methods

We also empirically validated the sensitivity of PID Lagrangian methods to the control gain tuning, particularly for the derivative term $k_d$. Experiments were conducted in the CarPush and CarButton environments using CPPO algorithms with varying $k_d$ values.

These results clearly illustrate that PID Lagrangian methods are highly sensitive to the choice of the $k_d$ value. Suboptimal tuning can lead to substantial degradation in both safety and overall performance.

Table 28: Sensitivity analysis of PID Lagrangian methods by varying the derivative gain $k_d$ on CarPush and CarButton environments (using CPPO). Bold highlights the worst-performing configuration in each environment, illustrating the sensitivity of PID to $k_d$.

| Environment | $k_d$ **Value** | **Vio. Rate (%)** | **Magnitude** | **Avg Cost** | **Avg Reward** |
|---|---|---|---|---|---|
| | 1 | 50.55 | 19.17 | 35.28 | -1.37 |
| | 0.1 | 66.00 | 24.87 | 46.81 | -2.26 |
| CarPush (CPPO) | 0.01 | 67.28 | 12.80 | 34.67 | 0.15 |
| | 0.001 | **99.80** | **96.50** | **121.45** | -0.02 |
| | 1 | **99.88** | **89.36** | **114.33** | 0.21 |
| CarButton (CPPO) | 0.1 | 99.80 | 43.32 | 68.29 | -1.89 |
| | 0.01 | 99.88 | 64.74 | 89.72 | -0.69 |

## G   Large Language Models Usage

We used GPT-5 (OpenAI) for grammar and style editing of the paper and for debugging auxiliary code (e.g., resolving error messages and minor refactoring). All technical ideas, method designs, experiments, and conclusions were created and verified by the authors. No confidential or reviewer-only information was shared with the model.

## H   Computational Cost Analysis

In this section, we evaluate the computational cost of our ADRC Lagrangian method compared to the PID and classical Lagrangian methods (Lag). The evaluation includes normalized computation time during the rollout phase (interaction with the environment) and the update phase (policy updates). The analysis is conducted using the DDPG and CPPO algorithms on the RacecarButton task, which features a multi-dimensional action space.

Table 29: Normalized Computation Time for DDPG and CPPO under RacecarButton task.

| **Metric** | **DDPG** | | | **CPPO** | | |
|---|---|---|---|---|---|---|
| | **PID** | **Lag** | **ADRC** | **PID** | **Lag** | **ADRC** |
| Rollout Time | 0.95 | 1.00 | 0.95 | 0.95 | 1.00 | 0.95 |
| Update Time | 0.94 | 1.00 | 1.00 | 0.94 | 1.00 | 1.00 |
| Total Time | 0.95 | 1.00 | 0.97 | 0.95 | 1.00 | 0.97 |

Compared with PID, at each episode, our ADRC method introduces minimal additional computation. Specifically, ADRC calculates the reference signal $r(t)$, solves the equation defined by Eqn. 20, and determines the optimal parameters $\omega_o$ based on Eqn. 21. These operations involve fixed and lightweight calculations that do not scale with the problem size, ensuring no additional time complexity is introduced.

The results in Table 29 confirm that ADRC achieves comparable computation times to the baselines. The rollout time of ADRC matches that of PID and slightly outperforms Lag, demonstrating efficiency in policy adjustments. During the update phase, ADRC incurs no additional cost compared to PID and Lag, aligning with the theoretical analysis that these computations are efficiently integrated into the training process.

All experiments were conducted on a machine equipped with an NVIDIA RTX 3090 GPU with 24GB of memory. Each task was trained for 4 million steps. For TRPO and CPPO, each training run takes approximately 10 GPU-hours, while DDPG and TD3 require about 18 GPU-hours per run.

