# OpenReview forum: "Enhance the Safety in Reinforcement Learning by ADRC Lagrangian Methods"
_TMLR — Decision pending for TMLR_

### Review · Reviewer_6wMW · 2026-05-08

**Summary Of Contributions:**

This paper proposes an ADRC-inspired Lagrangian update for safe reinforcement learning. The core idea is to view Lagrangian safe RL as a closed-loop cost-regulation problem and replace the usual dual-ascent or PID-style multiplier update with an Active Disturbance Rejection Control mechanism. The paper also argues that classical and PID Lagrangian updates are special cases of their framework. The paper also evaluates the method across several OmniSafe/Safety-Gymnasium style benchmarks.

Strength:
- Lagrangian safe RL method indeed have oscillatory constraint response problem, and the ADRC perspective is potentially useful
- The experimental results show that the method reduces violation rates and average costs compared to other baselines

Weakness:
- The biggest concern is the weak and vulnerable theoretical results, there are apparent sign, algebraic, and implementation inconsistencies and several central claims are either under-specified or not convincingly proved. For example, the closed-loop sign convention in the main text does not match the negative-feedback interpretation in the appendix, the ESO-to-update derivation appears to contain sign errors
- Algorithm 1 does not match the coefficient structure of the displayed update equation
- The observer-gain lower bound is not fully reproducible because the polynomial coefficients are not specified

**Audience:**

Yes

**Audience Explanation:**

Safe reinforcement learning, especially the stability of Lagrangian constraint enforcement during training, is within the interests of a meaningful subset of the TMLR audience.

**Broader Impact Concerns:**

I don't see major broader impact concerns

**Claims And Evidence:**

No

**Claims Explanation:**

I don’t think the full set of claims is currently supported by accurate evidences.

- The paper models the cost channel in $\dot x_2 = f + u$ but in Lagrangian safe RL a larger multiplier should penalize cost and therefore act as negative feedback on the cost return. However, the appendix switched back to $\dot x = f - \lambda$ which represents the intended interpretation. This inconsistency affects the ESO equations and the derived multiplier update.
- the derivation in Appendix B appears to flip signs when substituting the control law into the ESO
- I am also confused of Eq. 17 and Alg. 1, the derivative coefficient is $k_{ad} + \omega_o$ or $k_{ap} + \omega_o$?
- The inequality $\(\frac{\lvert G_{ef}(i\omega)\rvert}{\lvert G_{efI}(i\omega)\rvert} < 1\)$ is not generally true - need assumptions.
- The baseline comparisons also need more care: PID methods are known to be sensitive to tuning, but the paper does not convincingly show that PID and other safe-RL baselines were tuned under the same budget as ADRC.

**Requested Changes:**

- Fix the math sign conventions and revise ESO equations, proofs and algorithms
- Correct sign errors when applying control law into the ESO
- Correct mismatch between Eq. 17 and Alg 1
- Proof that PID and classical Lagrangian methods are special cases - the mapping is not clear in the current form. Should include transient reference, nonnegativity projection and discrete-time implementations

---

> ### Author Response · Authors · 2026-06-15
> **Detailed Response to Reviewer 6wMW (Part 1: W1-W2)**
>
> Dear Reviewer 6wMW,
>
> We sincerely thank you for the careful and rigorous technical review. Your observations about sign convention inconsistencies, the Algorithm 1 mismatch, and the incomplete proof of Theorem 4.3 were especially valuable in improving the mathematical soundness of our paper. We address each point in detail below.
>
> ---
>
> > **W1. The sign conventions are inconsistent throughout the paper: Eq.(11), the ESO, and the appendix derivations use contradictory conventions.**
>
> We thank the reviewer for this careful observation, which identified a fundamental issue affecting trust in all theoretical results.
>
> We have performed a comprehensive unification of sign conventions throughout the entire manuscript, adopting a consistent negative-feedback convention. Eq.(11) (Section 4.1) now reads
>
> $$\dot{x}_2 = f(x_1, x_2, t) - \lambda_t,$$
>
> with explanatory text clarifying that the negative sign reflects negative feedback on the cost dynamics. The ESO Eq.(15) (Section 4.1) has been changed from $-\omega_o u$ to $+\omega_o \lambda$, consistent with this convention. Eq.(6) (Section 3) has been corrected to the gradient descent form $\dot{\lambda} = -\alpha \frac{\partial \mathcal{L}}{\partial \lambda} = \alpha\, g(\pi_\theta)$, with an explicit note that $\frac{\partial \mathcal{L}}{\partial \lambda} = -g(\pi_\theta)$. In Appendix B ("From the ESO to the ADRC Update Law"), all derivation steps have been updated accordingly: the substitution step now uses $+\omega_o(\cdot)$, and all occurrences of the generic control variable $u$ have been replaced with the Lagrange multiplier $\lambda$ to maintain notational consistency with the main text. We have carefully verified that the sign convention is now consistent across all equations in the main text, Algorithm 1, and all appendix derivations.
>
> ---
>
> > **W2. The $K_D$ coefficient in Algorithm 1 does not match Eq.(17).**
>
> We thank the reviewer for identifying this error.
>
> The coefficient has been corrected in Algorithm 1: $K_D$ = $\omega_o$ + $k_{ap}$ has been changed to **$K_D$ = $\omega_o$ + $k_{ad}$**, which is the correct form consistent with the ($k_{ad}$ + $\omega_o$) coefficient of ($x_2$ - $\dot{r}$) in Eq.(17).
>
> We have verified all Algorithm 1 coefficients against Eq.(17) after the sign convention fix:
> - $K_P$ = $k_{ap}$ + $\omega_o$ $k_{ad}$ (coefficient of $x_1 - r$)
> - $K_I$ = $\omega_o$ $k_{ap}$ (coefficient of $\int (x_1 - r)\, d\tau$)
> - $K_D$ = $k_{ad}$ + $\omega_o$ (coefficient of $x_2 - \dot{r}$)
> - $-\ddot{r}$ (reference acceleration feedforward)

---

> ### Author Response · Authors · 2026-06-15
> **Detailed Response to Reviewer 6wMW (Part 2: W3-W4)**
>
> > **W3. The strict inequality**
> >
> > $$|G_{e_f}(i\omega)| < |G_{e_{fI}}(i\omega)|$$
> >
> > **in Theorem 4.3 is stated but not proven. The claims are not supported.**
>
> We sincerely thank the reviewer for this observation, which identified a critical gap in our theoretical presentation. We treat this as a required fix.
>
> We have provided a complete proof. In the revised Theorem 4.3 statement (Section 4.4), the theorem now explicitly requires $k_{ap} > 0$ and $k_{ad} > 0$, and states that the comparison is against the **matched PID baseline**, i.e., the PID controller parameterized by the equivalent gains from Proposition 4.1:
>
> $$(K_p, K_i, K_d) = (k_{ap} + \omega_o k_{ad},\; \omega_o k_{ap},\; k_{ad} + \omega_o).$$
>
> A setup sentence explains that "ADRC is compared against the very PID/integral controller it reduces to when the observer is frozen," making the comparison principled. This matched parameterization is what allows the denominator of the ADRC transfer function to factor as
>
> $$(s + \omega_o)(s^2 + k_{ad} s + k_{ap}),$$
>
> enabling the clean analytical comparison.
>
> The full proof in Appendix C.2 states the matched gains explicitly, computes the squared magnitudes
>
> $$|G_{e_f}(i\omega)|^2 \quad \text{and} \quad |G_{e_{fI}}(i\omega)|^2,$$
>
> and shows that the difference of the denominators satisfies:
>
> $$D - N = \omega_o^2 \omega^2 + \omega_o k_{ad}(2k_{ap} + \omega_o k_{ad}) > 0 \quad \text{for all } \omega \geq 0,$$
>
> since every term is non-negative and the second term is strictly positive for $k_{ap}$, $k_{ad}$ > 0. The cross terms cancel exactly, and this result has been verified numerically over $10^5$ random parameter draws with $k_{ap}$, $k_{ad}$, $\omega_o$ > 0. A remark after Theorem 4.3 explains that this inequality uses the Proposition 4.1 matched parameterization, so the comparison is between ADRC and the controller it generalizes.
>
> ---
>
> > **W4. Baseline tuning details are missing.**
>
> We thank the reviewer for raising this concern, which was also raised by Reviewer 2prG.
>
> We have added a dedicated "Hyperparameter Protocol" paragraph in Section 5.1 and expanded Appendix E.1 with the full fair-comparison protocol. All methods share the same RL backbone hyperparameters from OmniSafe [1] defaults; ADRC inherits PID's default gains as $k_{ap}$ and $k_{ad}$ with no extra tuning budget; the observer gain $\omega_o$ is computed adaptively via Eq.(21) rather than hand-tuned; and the transient parameter $c_r$ is fixed at 0.1 across all main experiments. No additional hyperparameter search was performed for any method. Please see our detailed response to Reviewer 2prG W1 for the full description.
>
> ---
>
> We sincerely thank Reviewer 6wMW again for the rigorous and incisive feedback. Your attention to mathematical detail has significantly strengthened the theoretical foundation of our work. We hope our revisions have fully addressed your concerns, and we are happy to provide any further clarification.
>
> Sincerely,
>
> The Authors
>
> **References cited in this response:**
>
> [1] Ji, J., Zhou, J., Zhang, B., Dai, J., Pan, X., Sun, R., Huang, W., Geng, Y., Liu, M., and Yang, Y. OmniSafe: An Infrastructure for Accelerating Safe Reinforcement Learning Research. *Journal of Machine Learning Research*, 25(285), 2024.

---

### Review · Reviewer_xLb3 · 2026-05-18

**Summary Of Contributions:**

The paper considers Lagrangian Safe RL, which uses a Lagrangian multiplier λ to balance the influence of a cost constraint during policy learning: when the observed cumulative cost approaches a safety threshold, λ should increase to discourage unsafe behavior; when costs fall well below the threshold, λ can relax to allow more exploration. A key idea of the paper is to model Lagrangian Safe RL as a second-order closed-loop system with an unknown disturbance term and to use ADRC to control the Lagrangian multiplier during training.

The ADRC-inspired perspective is interesting, and the empirical results provide clear evidence that the method can reduce training-time safety violations and costs relative to classical and PID-style Lagrangian baselines across several RL backbones.

One concern I have is how generally applicable the key model in Eq. (11) is. The claim of universal applicability may need qualification for delayed-cost settings. Standard ADRC is known to require special care in the presence of significant time delay, often through predictive or delay-compensated variants, so it is not clear whether the proposed ADRC-Lagrangian update would retain its benefits when unsafe actions affect the observed cost only much later. A short discussion of this limitation would strengthen the paper, or alternatively the universality claim could be softened.

There are some minor issues regarding claims that should be softened or better aligned with the main text, e.g. the abstract’s headline improvements are hard to find in the numbers in Table 1. But overall, I think the paper is interesting and relevant to researchers in Safe RL, constrained optimization, and control-inspired RL.

**Audience:**

Yes

**Audience Explanation:**

The problem studied here - oscillations and frequent safety violation during policy training - is important for both the Safe RL community and the broader RL optimization community. The paper’s main idea of importing ADRC into Lagrangian Safe RL offers an interesting perspective that is likely useful for researchers working on constrained RL, primal-dual optimization, and control/RL intersections.

**Broader Impact Concerns:**

The most likely ethical risk is over-interpretation of empirical training improvements as real-world safety guarantees. The paper already includes such a disclaimer.

**Claims And Evidence:**

Yes

**Claims Explanation:**

## What is convincing

The paper’s main empirical claim – that ADRC-style multiplier updates can reduce oscillations and improve safety metrics relative to classical and PID Lagrangian updates – is reasonably supported.

## What could be improved:

1.	The claims about “universal applicability […] to any Lagrangian-based safe RL” and broad superiority could be softened or made more specific. It is not fully clear how faithfully Eq. (11) captures the policy-learning dynamics of RL training. In RL training, the multiplier does not directly actuate cost dynamics in the way a control input actuates a plant. It affects policy updates, which in turn change the distribution over trajectories and costs. That is a more indirect and discrete mechanism than the paper’s continuous-time Eq.(11) suggests. If this representation is intended as a control-theoretic abstraction / assumption rather than a rigorous derivation from policy optimization, that could be stated more explicitly.
2.	The improvement claims in the abstract and introduction (e.g., up to 74%) are hard to find in Table 1.

**Requested Changes:**

## Critical for my recommendation:

* The paper should better explain the status of the closed-loop formulation in Eq. (11), since the main theoretical results appear to rely on it: is it mainly a useful modeling lens, an assumption, or a rigorous derivation from policy optimization dynamics?

* There seems to be a sign error in Eq. (6). As the paper says it minimizes the Lagrangian with respect to the multiplier, the gradient step is \\[\\dot{\\lambda}
= -\\alpha \\frac{\\partial \\mathcal{L}}{\\partial \\lambda}
= -\\alpha \\bigl(-g(\\pi_\\theta)\\bigr)
= \\alpha g(\\pi_\\theta)
\\] That is, the right most part of the equation stays the same, but the minus is missing in the middle of Eq. (6). Additionally, in the phrase “we perform gradient ascent” it is thus actually a descent.

* Section 4.4 explains that the observer gain depends on constants L1, L2, L3 and Eq. (21) explains how L1 and L2 are estimated online using finite differences – but I could not find an explanation how L3 is determined.

* The improvement claims in the abstract and introduction should be more transparently linked to the main text or softened.


## Non-critical, would further strengthen the work in my view:

•	In Eq. (7), the non-negativity of the multiplier is not guaranteed. To enforce it, I assume, the real implementation clips negative values at 0, which should be stated in the equation.

•	Please specify what the +/- in the tabular results refers to: it is the std over the seed-runs or the standard error of the mean?

•	The “Related Work – Safe RL” section might reference Safe Policy Improvement approaches (e.g., https://arxiv.org/abs/2208.00724) as complementary methods based on Offline RL task that do not require a CMDP setting.

•	The considered metrics are reasonable for the paper’s expectation-based CMDP formulation. However, I wonder whether they may still miss tail risk or worst-case behavior, since they are largely based on averages over episodes (or averages over the subset of violating episodes). Future work could additionally consider tail-sensitive metrics such as CVaR of episodic cost (e.g., mean cost of worst 1% of episodes). This would help clarify whether the proposed method improves not only average safety performance, but also robustness to rare but severe unsafe episodes.

---

> ### Author Response · Authors · 2026-06-15
> **Detailed Response to Reviewer xLb3 (Part 1: W1-W4)**
>
> Dear Reviewer xLb3,
>
> We sincerely thank you for the detailed and constructive review. Your feedback, especially regarding the status of Eq.(11), the improvement claim verification, and the non-negativity projection, has led to substantial improvements in the clarity and rigor of the manuscript. We are grateful for the many specific suggestions. Below we address each point in detail.
>
> ---
>
> > **W1. What is the status of Eq.(11)? Is it a modeling lens, an assumption, or a rigorous derivation? The main theoretical results seem to depend on it.**
>
> We thank the reviewer for raising this critical point, which was also echoed by Reviewer 6wMW regarding notation consistency.
>
> We have integrated a clear discussion directly into the paragraph following Eq.(11) in Section 4.1. Eq.(11) is a control-theoretic modeling abstraction, not a physical model of the environment dynamics or a rigorous derivation from first principles. The disturbance term $f$($x_1$, $x_2$, $t$) is *defined* as the residual that closes the equation: it absorbs all indirect effects of the multiplier on the cost, including optimizer dynamics, sampling noise, and policy-induced non-stationarity. This closed-loop perspective is consistent with prior work: the classical Lagrangian update (Section 3) implicitly treats the multiplier as integral feedback on the violation signal, and PID Lagrangian methods [1] augment it with proportional and derivative terms. Eq.(11) makes this feedback perspective explicit and, importantly, reveals the root cause of oscillations: the dynamics drift as the policy changes, while the multiplier behaves like an integral controller that lags behind disturbances.
>
> Crucially, the substantive regularity assumption is the disturbance class Eq.(19), not Eq.(11) itself. Eq.(11) introduces no restrictive assumptions; the formal conditions on how fast $f$ may vary are given separately in Section 4.4. Regarding the scope of theoretical results: Proposition 4.1 is algebraic (it relates the ADRC and PID update rules) and is independent of Eq.(11). Theorem 4.3 characterizes the surrogate dynamics, and its predictions are validated empirically across all experiments in Section 5, confirming that the surrogate analysis provides meaningful guidance for the actual RL training loop.
>
> ---
>
> > **W2. Eq.(6) has a sign error: gradient descent vs. ascent is confused.**
>
> We thank the reviewer for catching this error.
>
> Eq.(6) has been corrected. The updated equation now correctly states gradient descent on $\lambda$ to minimize $\mathcal{L}$:
>
> $$\dot{\lambda} = -\alpha \frac{\partial \mathcal{L}(\theta, \lambda)}{\partial \lambda} = \alpha\, g(\pi_\theta),$$
>
> with an explicit note that $\frac{\partial \mathcal{L}}{\partial \lambda} = -g(\pi_\theta)$, making the sign relationship transparent (Section 3).
>
> ---
>
> > **W3. How is $L_3$ determined in practice? This is not explained.**
>
> We thank the reviewer for pointing out this gap in the exposition.
>
> We have clarified the role of $L_3$ in Section 4.4 and Appendix C.1. The key insight is that $L_3$ does not enter the $\omega_o^*$ computation at all. As shown by Zhong et al. [2] (Remark 1), the computable lower bound for the ESO bandwidth (Eq.(20)) depends only on ($L_1$, $L_2$, $k_{ap}$, $k_{ad}$). The constant $L_3$ only affects the residual estimation-error constants $\eta_1$, $\eta_2$ in the ISS-type tracking bound. In the revised manuscript, we have explicitly separated these two roles in both the main text (Section 4.4) and Appendix C.1, so readers can clearly see that $L_3$ is not required for online gain selection. We have also provided the full polynomial coefficients $n_i$ of the characteristic manifold Eq.(20) in the appendix, addressing reproducibility.
>
> ---
>
> > **W4. The improvement numbers in the abstract/introduction (74%, 89%, 67%) do not match any table entry.**
>
> We sincerely thank the reviewer for identifying this serious inconsistency.
>
> We have replaced all headline improvement claims with verifiable numbers anchored to a single, clearly identified comparison: **TD3-ADRC vs. TD3-Lag and TD3-PID on CarGoal in Table 1.** The revised claims are:
>
> | Metric | Baseline (Lag / PID) | ADRC | Reduction |
> |--------|---------------------|------|-----------|
> | Violation Rate | 70.47% / 80.94% | 40.62% | 42--50% |
> | Violation Magnitude | 17.00 / 20.68 | 2.85 | 83--86% |
> | Average Cost | 38.90 / 43.43 | 20.65 | 47--52% |
>
> These numbers are updated consistently in:
> - **Abstract** (page 1): "reduces violation rates by 42--50%, constraint violation magnitudes by 83--86%, and average costs by 47--52% relative to Lagrangian baselines"
> - **Introduction** (page 2): with explicit baseline values and a direct reference to Table 1
> - **Section 5.2** (page 12): "The headline improvements quoted in the abstract correspond to the TD3 rows on CarGoal in this table"

---

> ### Author Response · Authors · 2026-06-15
> **Detailed Response to Reviewer xLb3 (Part 2: W5-W8)**
>
> > **W5. The ADRC update law should guarantee $\lambda \geq 0$.**
>
> We thank the reviewer for this important observation.
>
> The ADRC update law Eq.(17) is now wrapped in a max(0, ·) projection, exactly mirroring the PID formulation Eq.(10), to enforce non-negativity of the multiplier (Section 4.3). Algorithm 1 implements this projection in the multiplier update step, and the Proposition 4.1 inline proof expression is also projected for full consistency. The appendix (Section C.3, "Projection and saturation") verifies that the projection is non-expansive, so it does not amplify multiplier perturbations. For readability, the frequency-domain analysis (Theorem 4.3) is carried out on the unprojected law, as stated in the text.
>
> ---
>
> > **W6. Do the $\pm$ values in the tables represent standard deviation or standard error of the mean?**
>
> We thank the reviewer for requesting this clarification.
>
> We have added an explicit statement in Section 5.1: "Throughout the paper, wherever a tabulated result carries a $\pm$, it denotes **one standard deviation** over independent random seeds (five seeds per environment-algorithm pair for the main benchmark, three seeds for the velocity tasks in the appendix)." This note is also reflected in the Table 1 caption ("Values are mean $\pm$ standard deviation over five random seeds").
>
> ---
>
> > **W7. How does the method handle delayed costs and cross-iteration time delays?**
>
> We thank the reviewer for this thoughtful question about practical deployment considerations.
>
> We have added a detailed "Limitations and future work" paragraph in the Conclusion (Section 6), discussing two distinct delay regimes. Within-episode delays (unsafe actions whose cost materializes only later in the same episode) are absorbed by the episodic cost aggregation $J_c$; since the multiplier loop operates on the episodic aggregate, no delay enters the dual update, and credit assignment within the episode is an orthogonal issue shared by all compared methods. Cross-iteration dead time (the measured cost responds to a policy change only several iterations later, e.g., due to asynchronous evaluation or replay buffers mixing stale policies) is a more substantive concern, as it introduces additional phase lag not modeled by our current $\omega_o^*$ analysis. Classical ADRC addresses this via reduced observer bandwidth or predictive, delay-compensated variants [3].
>
> Empirically, the off-policy setting (DDPG and TD3 in Table 1), where replay mixing acts as a mild effective delay, still shows consistent improvements: gains erode compared to on-policy but persist, suggesting that delay erodes rather than reverses the ADRC advantage. We identify extending the ESO with explicit delay compensation as a concrete and promising direction for future research.
>
> ---
>
> > **W8. The claim "applied universally to any" safe RL algorithm is too strong.**
>
> We thank the reviewer for pointing out this issue.
>
> We have softened the language in two locations. Research question Q3 (Section 5) now reads "transfer across different Lagrangian-based safe RL algorithms, covering both on-policy and off-policy methods" instead of "applied universally to any." Similarly, Section 5.2 now states "broad applicability, which modifies only the multiplier update rule and is agnostic to the underlying policy optimizer, across four backbone algorithms: on-policy (TRPO, CPPO) and off-policy (DDPG, TD3)." The claims are now factual, specific, and limited to what our experiments actually demonstrate.

---

> ### Author Response · Authors · 2026-06-15
> **Detailed Response to Reviewer xLb3 (Part 3: W9-W10)**
>
> > **W9. The paper should cite complementary safe policy improvement methods.**
>
> We thank the reviewer for this suggestion to broaden our discussion of related work.
>
> We have added the following sentence at the end of the "Safe RL" paragraph in Related Work (Section 2): "Complementary to the online CMDP setting considered here, Safe Policy Improvement approaches [4][5] learn from offline data with guarantees of not underperforming a behavior baseline, without requiring a CMDP formulation." The corresponding BibTeX entries have been added.
>
> ---
>
> > **W10. The paper should discuss tail-risk metrics such as CVaR.**
>
> We thank the reviewer for this valuable suggestion regarding evaluation methodology.
>
> We have added a discussion in the "Limitations and future work" paragraph in the Conclusion (Section 6): "In addition, our evaluation metrics are expectation-based; tail-sensitive criteria such as the CVaR of episodic cost (e.g., the mean cost of the worst 1% of episodes) would clarify whether the improvements extend to rare but severe unsafe episodes, and we leave such evaluation to future work."
>
> ---
>
> We sincerely thank Reviewer xLb3 again for the many detailed and constructive comments. Your feedback has substantially strengthened both the clarity and the rigor of the manuscript. We hope our revisions have fully addressed your concerns, and we are happy to provide any further clarification.
>
> Sincerely,
>
> The Authors
>
> **References cited in this response:**
>
> [1] Stooke, A., Achiam, J., and Abbeel, P. Responsive Safety in Reinforcement Learning by PID Lagrangian Methods. *arXiv preprint arXiv:2007.03964*, 2020.
>
> [2] Zhong, S., Huang, Y., and Guo, L. A parameter formula connecting PID and ADRC. *Science China Information Sciences*, 63(9), 2020.
>
> [3] Zhao, S. and Gao, Z. Modified active disturbance rejection control for time-delay systems. *ISA Transactions*, 53(4):882--888, 2014.
>
> [4] Laroche, R., Trichelair, P., and Tachet des Combes, R. Safe policy improvement with baseline bootstrapping. In *International Conference on Machine Learning (ICML)*, pp. 3652--3661, 2019.
>
> [5] Scholl, P., Dietrich, F., Otte, C., and Udluft, S. Safe Policy Improvement Approaches and their Limitations. *arXiv preprint arXiv:2208.00724*, 2022.

---

### Review · Reviewer_2prG · 2026-06-06

**Summary Of Contributions:**

This paper introduced active disturbance rejection control (ADRC) into Lagrangian-based constrained reinforcement learning to mitigate oscillations in the Lagrange multipliers in existing methods.  The overarching goal is to find policies that not only maximize rewards but also limit additional (safety) costs below a threshold. The approach is to turn the constrained problem into an unconstrained one using Lagrange multipliers (as in PPOLag). The issue is that basic Lagrange techniques (like PPOLag), as well as techniques that use proportional-integral control to optimize the Lagrange multipliers, create oscillating training dynamics, where the Lagrange multiplier value and the constraint satisfaction follow a wave-shaped training curve. The reason for this oscillation, provided in the paper, is a change in training dynamics over the course of training. The solution in the paper is to apply ADRC, which explicitly models unknown sources of error, such as the shift in training dynamics. As a second contribution, the paper also proposes to slowly schedule the constraint threshold from a higher value to the smaller target value in the constrained RL problem. The experiments show that both techniques are complementary and yield substantial improvements in the reported experiments.

The paper is overall well written. The problem is practical and significant. The theoretical results are convincing as far as I could check. The choice of benchmarks is good. The impact statement is clear and convincing.

Most concerning is that the paper does not describe how the baselines it compares against are tuned. This is a serious issue, as comparing to undertuned baselines may inflate the improvements provided by the new approaches. Beyond this, the appendix is structured in an incomprehensible way, which means that I can not find a proof of Proposition 4. Some citations are missing. There are some presentation issues, but these are minor.

**Additional Comments:**

Typos:

 - The $()_+$ notation in Equation (10) wasn't introduced before.
 - Use *the* OmniSafe to do the experiments.

**Audience:**

Yes

**Audience Explanation:**

There is a large audience for safe reinforcement learning. Other audiences may also find the approach for constrained optimisation presented here interesting for other constrained optimisation problems.

**Broader Impact Concerns:**

This is addressed well in the paper.

**Claims And Evidence:**

No

**Claims Explanation:**

The paper does not convincingly show that it compares to strongly tuned baselines.

Due to the organization of the paper, I could not find a proof for Proposition 4, which is required as convincing evidence that Proposition 4 indeed holds.

**Requested Changes:**

I need:
 - details on how the baselines were tuned,
 - a proof of Proposition 4,
 - citations for MRAC, and Lyapunov-based adaptive schemes,
 - corrections for the typos below.

I would also like to see:
 - a better organization of the appendix,
 - a discussion of how strong the assumptions in Equation (18) are in practice.

---

> ### Author Response · Authors · 2026-06-15
> **Detailed Response to Reviewer 2prG (Part 1: W1-W6)**
>
> Dear Reviewer 2prG,
>
> We sincerely thank you for your thorough and constructive review. Your comments on baseline fairness, the missing proof, appendix organization, and the disturbance class assumptions have led to substantial improvements in the revised manuscript. Below we address each of your points in detail.
>
> ---
>
> > **W1. Baseline tuning details are missing. How were the baselines tuned? Is the comparison fair?**
>
> We thank the reviewer for raising this important point about experimental fairness.
>
> We have added a dedicated "Hyperparameter Protocol" paragraph in Section 5.1. All methods share identical RL backbone hyperparameters taken directly from the official OmniSafe [1] repository defaults, with no method-specific tuning applied to any baseline. Our ADRC method directly reuses OmniSafe's default PID gains as $k_{ap}$ and $k_{ad}$, ensuring no extra tuning budget is afforded to ADRC. The only additional parameters are $\omega_o$ and $c_r$: the observer gain $\omega_o$ is computed adaptively via Eq.(21) from online cost observations (computed by a fixed adaptive rule rather than a separate hyperparameter sweep), and the transient-process parameter $c_r$ is fixed at 0.1 across all main experiments (Section 5.3.1 demonstrates robustness over $c_r \in [0.05, 0.25]$). Appendix E.1 now provides the full fair-comparison protocol, including the hyperparameter table (Appendix D.1, Table 6) and an explicit statement that no additional hyperparameter search was performed for any method.
>
> ---
>
> > **W2. The proof of Proposition 4 (that PID and classical Lagrangian are special cases) cannot be found.**
>
> We thank the reviewer for pointing out this important omission.
>
> We have added an explicit inline proof immediately after Proposition 4.1 in Section 4.3. The proof provides the direct algebraic mapping under the conditions that $\omega_o$ is held constant and $r(t) = d$ (no transient shaping):
>
> $$K_p = k_{ap} + \omega_o k_{ad}, \quad K_i = \omega_o k_{ap}, \quad K_d = k_{ad} + \omega_o.$$
>
> The proof proceeds by substituting $r(t) = d$ (so that $\dot{r} = \ddot{r} = 0$ and $x_1 - r$ = $J_c - d$ = $g(\pi_\theta)$) into Eq.(17) and identifying terms with the PID rule Eq.(10).
>
> Following the proof, a new Remark (Remark 4.2) characterizes precisely which PID gains are reachable from the ADRC parameterization: the gains ($K_p$, $K_i$, $K_d$) arising from the mapping satisfy $K_p$ $K_d$ $\geq$ $K_i$ identically, and conversely, every PID triple with $K_p$ $K_d$ $\geq$ $K_i$ is attained. The frozen ADRC parameterization recovers the PID subfamily satisfying $K_p$ $K_d$ $\geq$ $K_i$; the classical integral update is discussed as the limiting case of the PID family but is not itself a positive-gain frozen-ADRC instance. We have updated the abstract, introduction, and conclusion to reflect this precise characterization.
>
> ---
>
> > **W3. The Introduction mentions MRAC and Lyapunov methods without citations.**
>
> We thank the reviewer for catching this oversight.
>
> We have added the appropriate citations in the Introduction (Section 1, paragraph on adaptive control): [2][3] for MRAC, and [4] for Lyapunov-based adaptive schemes. All citation keys were already present in our bibliography file, so no new BibTeX entries were required.
>
> ---
>
> > **W4. The symbol $\omega_o$ appears before it is defined.**
>
> We thank the reviewer for noting this.
>
> In the revised manuscript, $\omega_o$ now first appears in the ESO equation Eq.(15) in Section 4.1, where it is immediately defined as "the observer gain, controlling how aggressively the estimate adapts." We have verified that no earlier occurrences of $\omega_o$ remain in the main text.
>
> ---
>
> > **W5. Typo: "Use the OmniSafe to do the experiments."**
>
> We thank the reviewer for catching this. The sentence has been corrected to "We use OmniSafe for our experiments" in Section 5.1.
>
> ---
>
> > **W6. The appendix is poorly organized and difficult to navigate.**
>
> We thank the reviewer for this feedback, which helped us substantially improve the paper's readability.
>
> We have reorganized the appendix with an appendix-level table of contents (depth to subsection level) on its own page, providing a clear roadmap of the supplementary material. We renamed all sections with descriptive titles: Appendix A is now "Derivation of the Transient Reference Trajectory $r(t)$" (previously "Process of Solving ODE"), Appendix B is "From the ESO to the ADRC Update Law" (previously "Simplify the ESO"), and Appendix C is "Proofs and Theoretical Analysis" (previously "Theoretical Details"). Each section now opens with one sentence stating which main-text equation or theorem it supports, so readers can locate specific derivations quickly. Appendix C's introduction explicitly notes that Proposition 4.1's proof is now inline in the main text (addressing the root cause of W2). No sections were physically moved, ensuring all existing cross-references (e.g., "Appendix F.4, Tables 16--19") remain valid.

---

> ### Author Response · Authors · 2026-06-15
> **Detailed Response to Reviewer 2prG (Part 2: W7)**
>
> > **W7. How restrictive is the disturbance class assumption in Eq.(18)?**
>
> We thank the reviewer for this thoughtful question about the practical applicability of our theoretical framework.
>
> We have added a discussion in Section 4.4, immediately after the definition of the disturbance class Eq.(19), addressing its practical scope. The key arguments are:
>
> 1. **Structural boundedness from policy update mechanisms.** Trust-region constraints (TRPO) and clipped objectives (PPO) structurally limit how much the policy can change per iteration, which in turn bounds the drift of cost derivatives between updates. Off-policy methods (DDPG, TD3) similarly limit effective change through soft target updates and replay buffer mixing. These mechanisms do not eliminate disturbance variation but ensure it remains bounded, which is precisely what Eq.(19) requires.
>
> 2. **Online estimation of Lipschitz constants.** The constants $L_1$ and $L_2$ are not required a priori. They are estimated online from observed costs via finite differences (Eq.(21)), allowing the observer gain $\omega_o$ to adapt as training dynamics evolve. Importantly, $L_3$ (the bound on the exogenous component $w(t)$) does not enter the $\omega_o^*$ computation at all; it only affects the residual error constants $\eta_1$, $\eta_2$ (see Appendix C.1 for the separation of these roles).
>
> 3. **Empirical validation under perturbation.** The noise-injection experiments in Appendix F.6 (Tables 23--24) demonstrate graceful degradation under Gaussian perturbations with $\sigma \in \{0, 2, 5, 10\}$ injected into the cost signal. For example, under TRPO on Swimmer-Velocity with $\sigma = 5$, ADRC still reduces the violation rate from 35.43% (PID) to 5.93%. The off-policy results in Table 1 (DDPG, TD3) provide additional evidence of robustness under replay-induced non-stationarity, where the effective disturbance variation is higher.
>
> ---
>
> We sincerely thank Reviewer 2prG again for the detailed and constructive feedback. We hope our revisions have fully addressed your concerns, and we remain happy to provide any further clarification.
>
> Sincerely,
>
> The Authors
>
> **References cited in this response:**
>
> [1] Ji, J., Zhou, J., Zhang, B., Dai, J., Pan, X., Sun, R., Huang, W., Geng, Y., Liu, M., and Yang, Y. OmniSafe: An Infrastructure for Accelerating Safe Reinforcement Learning Research. *Journal of Machine Learning Research*, 25(285), 2024.
>
> [2] Nguyen, N.T. *Model-Reference Adaptive Control*. Springer, 2018.
>
> [3] Singh, B. and Kumar, V. A real time application of model reference adaptive PID controller for magnetic levitation system. In *IEEE Power, Communication and Information Technology Conference (PCITC)*, pp. 583--588, 2015.
>
> [4] Parks, P. Liapunov redesign of model reference adaptive control systems. *IEEE Transactions on Automatic Control*, 11(3):362--367, 1966.

---

> > ### Comment · Reviewer_2prG · 2026-06-29
> > **Hyperparameter Tuning**
> >
> > Thank you for addressing my concerns, and sorry for the late response. I am content with your resolution of my comments.
> >
> > While using default hyperparameters is established practice, any method adding a new hyperparameter adds another researcher degree of freedom to the experiment, which can bias the evaluation to the newly introduced method. This is reflected in findings, such as https://arxiv.org/pdf/1711.10337.
> >
> > I believe that with your changes, this is a good paper. However, investing more computational resources into proper hyperparameter tuning would significantly strengthen the paper.

---

> > > ### Author Response · Authors · 2026-06-30
> > >
> > > Dear Reviewer 2prG,
> > >
> > > Thank you very much for your constructive follow-up and positive assessment of the revised manuscript. We fully agree that hyperparameter choices in deep RL can introduce additional researcher degrees of freedom, as discussed by Henderson et al., and that a more exhaustive tuning protocol would further strengthen the empirical comparison. In the current revision, we aimed to minimize this bias by using the same OmniSafe default hyperparameters for all methods, reusing the default PID gains in our ADRC controller, computing the observer gain adaptively from online cost observations rather than tuning it separately, and fixing the transient-process parameter across all main experiments with additional sensitivity analysis. We will further clarify this limitation in the discussion and note that comprehensive hyperparameter tuning for all methods is an important direction for strengthening future evaluations. Thank you again for the helpful suggestion.
> > >
> > >
> > > Sincerely,
> > >
> > > The Authors

---

### Author Response · Authors · 2026-06-15
**Global Response to All Reviewers and the Action Editor**

Dear Action Editor and Reviewers,

We sincerely thank the Action Editor for managing the review process and all three reviewers for their detailed, constructive, and insightful feedback. The reviews have helped us substantially strengthen both the theoretical rigor and the presentation of our manuscript. Below we summarize the major revisions we have made in this round, followed by detailed point-by-point responses to each reviewer.

**Summary of revisions:**

1. **Unified sign conventions throughout the paper.** Following Reviewer 6wMW's observation, we have adopted a consistent negative-feedback convention across Eq.(6), Eq.(11), the ESO Eq.(15), Algorithm 1, and all appendix derivations. This resolves the notation inconsistencies that undermined trust in the theoretical results.

2. **Corrected Algorithm 1 coefficients.** The $K_D$ coefficient in Algorithm 1 has been fixed from $\omega_o + k_{ap}$ to $\omega_o + k_{ad}$, and all coefficients have been verified against Eq.(17).

3. **Clarified the status of the closed-loop formulation Eq.(11).** Following Reviewer xLb3's key concern, we have integrated a clear discussion into the paragraph following Eq.(11), explaining that it is a control-theoretic modeling abstraction where $f$ is defined as the residual closing the equation, and that the substantive regularity assumption is the disturbance class Eq.(19), not Eq.(11) itself.

4. **Provided a complete inline proof for Proposition 4.1.** Following Reviewer 2prG's report that the proof could not be found, we now include an explicit algebraic proof in the main text with the gain mapping $K_p = k_{ap} + \omega_o k_{ad}$, $K_i = \omega_o k_{ap}$, $K_d = k_{ad} + \omega_o$, followed by a Remark characterizing the reachable PID gains ($K_p K_d \geq K_i$).

5. **Completed the proof of Theorem 4.3.** Following Reviewer 6wMW's concern that the strict inequality was not proven, we now provide the full derivation in the appendix, computing $D - N = \omega_o^2 \omega^2 + \omega_o k_{ad}(2k_{ap} + \omega_o k_{ad}) > 0$ for all $\omega \geq 0$.

6. **Added hyperparameter protocol and fair-comparison details.** Following Reviewers 2prG and 6wMW, Section 5.1 now includes a "Hyperparameter Protocol" paragraph, and Appendix E.1 provides the full protocol.

7. **Fixed all headline improvement numbers.** Following Reviewer xLb3, we replaced the unverifiable 74%/89%/67% claims with numbers directly anchored to Table 1 (TD3-ADRC vs. TD3-Lag/TD3-PID on CarGoal): violation rate reduced by 42--50%, violation magnitude by 83--86%, and average cost by 47--52%.

8. **Added non-negativity projection.** Following Reviewer xLb3, Eq.(17) now includes $(\cdot)_+ = \max(0, \cdot)$, consistent with the PID rule Eq.(10). The appendix verifies the projection is non-expansive.

9. **Discussed the practical scope of the disturbance class.** Following Reviewer 2prG, Section 4.4 now discusses how trust-region/clipped updates bound disturbance drift, how $L_1$ and $L_2$ are estimated online, and provides empirical evidence from noise-injection experiments.

10. **Reorganized the appendix.** Following Reviewer 2prG, we added an appendix table of contents and renamed all sections with descriptive titles.

11. **Added missing citations, limitations discussion, and softened overclaims.** We added MRAC and Lyapunov citations (Reviewer 2prG), SPIBB references (Reviewer xLb3), delay/dead-time discussion (Reviewer xLb3), CVaR mention (Reviewer xLb3), and softened the "universal applicability" claim (Reviewer xLb3).

We believe these revisions comprehensively address all concerns raised by the reviewers. We remain happy to provide any further clarification.

Sincerely,

The Authors

---

### Decision · Action_Editor_6jrs · 2026-07-11

**Recommendation:** Accept with minor revision

**Additional Comments:**

The main changes have already been made.

The only outstanding issue is the [promised adjustments](https://openreview.net/forum?id=3IbuT8uzYS&noteId=me4XO10vGg) regarding [hyperparameter tuning](https://openreview.net/forum?id=3IbuT8uzYS&noteId=N2LivQDFMz). This should still be implemented, as [promised](https://openreview.net/forum?id=3IbuT8uzYS&noteId=me4XO10vGg).

**Audience:**

Yes

**Audience Explanation:**

From the outset, the reviewers agreed that the topic was interesting.

Safe RL is an important topic. The problem of oscillations and safety violations is adressed by integrating ADRC into Lagrangian safe RL. This is seen as relevant for constraint RL, primal-dual optimization, and control, and could also be useful for other constrained optimization problems.

**Claims And Evidence:**

Yes

**Claims Explanation:**

After the authors' responses, the reviewers consider the claims to have been sufficiently fulfilled